# How Do Large Language Models Understand Graph Patterns? A Benchmark for Graph Pattern Comprehension

**Xinnan Dai**[1], **Haohao Qu**[2], **Yifei Shen**[3], **Bohang Zhang**[4], **Qihao Wen**[1],
**Wenqi Fan**[2], **Dongsheng Li**[3], **Jiliang Tang**[1], **Caihua Shan**[3]
[1]Michigan State University, [2]The Hong Kong Polytechnic University,
[3]Microsoft Research, [4]Independent Researcher
{daixinna, wenqihao, tangjili}@msu.edu, haohao.qu@connect.polyu.hk,
{zhangbohang.pku, wenqifan03}@gmail.com,
{yifeishen, dongsheng.li, caihuashan}@microsoft.com

## Abstract

Benchmarking the capabilities and limitations of large language models (LLMs) in graph-related tasks is becoming an increasingly popular and crucial area of research. Recent studies have shown that LLMs exhibit a preliminary ability to understand graph structures and node features. However, the potential of LLMs in graph pattern mining remains largely unexplored. This is a key component in fields such as computational chemistry, biology, and social network analysis. To bridge this gap, this work introduces a comprehensive benchmark to assess LLMs' capabilities in graph pattern tasks. We have developed a benchmark that evaluates whether LLMs can understand graph patterns based on either terminological or topological descriptions. Additionally, our benchmark tests the LLMs' capacity to autonomously discover graph patterns from data. The benchmark encompasses both synthetic and real datasets, and a variety of models, with a total of 11 tasks and 7 models. Our experimental framework is designed for easy expansion to accommodate new models and datasets. Our findings reveal that: (1) LLMs have preliminary abilities to understand graph patterns, with O1-mini outperforming in the majority of tasks; (2) Formatting input graph data to align with the knowledge acquired during pretraining can enhance performance; (3) LLMs employ diverse potential algorithms to solve one task, with performance varying based on their execution capabilities. Our dataset and implementations are available at `https://github.com/DDigimon/GraphPattern`.

## 1 Introduction

Originally trained on textual data, LLMs have demonstrated remarkable success in various tasks, such as reading comprehension and text reasoning (Achiam et al., 2023; Touvron et al., 2023). To evaluate whether LLMs can adapt the text understanding ability across graphs, several studies have investigated this at both the feature and structural levels (Zhao et al., 2023; Chai et al., 2023). Specifically, LLMs have been shown to enhance node features on graphs (Ye et al., 2023; Huang et al., 2024; Chen et al., 2024c). Additionally, graph structure understanding tasks, such as shortest path and connectivity, have also been evaluated (Guo et al., 2023; Wang et al., 2024).

Graph patterns, a key aspect of graphs, have yet to be thoroughly explored. Mining graph patterns play a critical role in numerous real-world applications. For instance, they aid in uncovering new insights within biological protein-protein interaction networks (Hu et al., 2005; Tran et al., 2013), identifying key molecular structures in chemistry (Murray & Rees, 2009), and detecting fraudulent activities in transaction networks (Cheng et al., 2020), and so on. In addition, these patterns represent fundamental transferable structures among graphs, such as community groups for friendship recommendations in social networks (Wu et al., 2022) and functional groups for molecular property

---

Correspondence to: Caihua Shan (caihuashan@microsoft.com); Jiliang Tang (tangjili@msu.edu).

prediction (Agarwal et al., 2023). Therefore, it is of great interest to investigate whether LLMs possess the capability to comprehend these patterns and effectively apply them to a variety of graph mining and learning tasks.

In this work, we begin by exploring how LLMs handle various types of graph patterns and how they can be applied to real-world applications. Specifically, we categorize the descriptions of graph patterns into three types: **terminology-based**, **topology-based**, and **data-driven**, as shown in Table 1. Based on these descriptions, we progressively challenge LLMs' abilities in graph pattern tasks:

Table 1: Three types of descriptions. 'Square' is defined by terminology and topology to outline its structure. In a data-driven description, 'Square' frequently appears as a common pattern in the data.

| Terminology-based | Topology-based | Data-driven |
|---|---|---|
| The pattern, named Square, is a 4-node cycle with each node connected to exactly two others. | The pattern, defined as G, is an undirected graph with four Node 0, 1, 2, 3. Node 0 is connected to Nodes 1 and 2. Node 1 is connected to Nodes 2 and 3. |  |

**Can LLMs understand graph patterns based on terminology-based descriptions? (Section 3)** Given that LLMs are pre-trained on extensive internet datasets, which include passages relevant to graph terminologies, it is plausible to hypothesize that they possess a rudimentary understanding of simple graph patterns as described in terminology, such as 'triangle' or 'diamond'. To evaluate this, we have devised three subtasks aimed at progressively testing LLMs' comprehension of terminologically defined patterns. In detail, we first examine the alignment between LLMs' and human understanding of the same pattern, and then assess whether LLMs can follow human instructions to modify and detect these patterns.

**Can LLMs understand graph patterns based on topology-based descriptions? (Section 4)** While certain graph patterns can be succinctly described using natural language, more complex patterns often necessitate representation through adjacency lists or edge lists. As LLMs are not trained for permutation invariance, the same pattern, when described with different node ID sequences, may result in varying understanding by LLMs. Thus, to examine LLMs' consistency in recognizing identical patterns, we first require LLMs to perform pattern mapping through isomorphic identification. Following this, we assess their ability to edit and extract topology-based patterns, similar to their handling of terminology-based patterns.

**Can LLMs automatically discover graph patterns from data? (Section 5)** The benchmarks previously discussed assess LLMs' comprehension of rule-based predefined graph patterns. These patterns, initially identified by human experts, represent a foundational level of understanding. To further probe the depth of LLMs' capability, we check their ability to independently mine graph patterns from a provided dataset. We explicitly prompt certain key information to constrain and guide the data-driven patterns, such as detecting dense substructures like k-core, finding the frequent subgraphs, or identifying the distinctive patterns based on the labels of input graphs.

**Can LLMs deal with graph patterns in real-world applications? (Section 6)** Compared with synthetic datasets, real-world datasets contain node and edge attributes and inevitable noise within graphs, increasing the difficulty for LLMs to comprehend. We gather ten real-world datasets across diverse domains, including molecules, social networks, bioinformatics, and computer vision, and perform similar tasks such as pattern detection, to assess their ability.

Based on the extensive experimental results from various tasks, we provide theoretical intuitions and summarize three key empirical insights on the capacity of LLMs for graph patterns in Section 7.

In conclusion, this paper provides an initial yet comprehensive study of LLMs' understanding on graph patterns, aiming to further improve performance in graph learning and mining tasks, and enhance graph reasoning skills.

## 2 BENCHMARK SETTINGS

**Primitive graph patterns.** We select 9 primitive graph patterns with varying numbers of nodes, edges and connection types, commonly used in network analysis (Cheng et al., 2008; Ying et al.,

2019; Li et al., 2021). Specifically, we include 5 undirected graph patterns: triangle, tailed-triangle (T-triangle), square, diamond, and house. For directed graph patterns, we have V-structure (V-S), feedforward loop (FFL), feedback loop (FBL), and directed-diamond (D-diamond). The detailed definitions are illustrated in Appendix Table 11. These graph patterns serve as key examples in constructing datasets and testing LLMs' ability in terminology-based and topology-based tasks.

**Datasets.** For comprehensive benchmarking, we consider multiple factors to construct diverse datasets. We design the terminology-based and topology-based descriptions for patterns. The topology of patterns can be represented using either the adjacency list (A.L.) or edge list (E.L.) format. Furthermore, certain tasks, such as modification and detection, require input graphs beyond the patterns themselves. We randomly generate varying input graph sizes: small (S) with 5–15 nodes, medium (M) with 15–25 nodes, and large (L) with 25–35 nodes, to control the task difficulty. These input graphs are also described in either either adjacency list or edge list format. In addition to the synthetic datasets, we also utilize real-world application attributed graphs from various domains, including molecules, information networks, social networks, and computer visions. We summarize the datasets used and the prompts designed for each task in Appendix Table 16.

**Large Language Models.** Since different LLMs may have varying task performance, we evaluate several widely used models, including GPT-4 (0125-Preview), GPT-4o (2024-05-13), Mixtral (open-mixtral-8x22b), Llama (llama3.1-401B-Instruct), Gemini (gemini-1.5-pro), Claude (claude-3-opus-20240229) and O1-mini (o1-mini-2024-09-12). We set temperatures to 0 for all the models.

## 3 TERMINOLOGY-BASED PATTERNS

In this section, we first introduce a pattern translation task to assess the alignment between LLMs' comprehension and human understanding of terminology-based graph patterns. Then, we design graph modification and pattern detection tasks to evaluate LLMs' ability to either modify the input graph to include specific patterns, or identify all patterns present in the input graph.

### 3.1 PATTERN TRANSLATION

We study whether LLMs have an accurate understanding of terminology-based patterns. Given the terminology of a primitive graph pattern, we prompt LLMs to create a graph including this pattern. Besides, we also require that the resulting graph is constrained by a specified number of nodes and pattern occurrences. One prompt example is "Generate a graph that includes only one triangle, with a total of 20 nodes. Each node should be connected." For evaluation, we use accuracy (ACC) to assess whether the output meets the requirement. In addition, to evaluate the creativity of LLMs, we measure diversity of generated graphs by comparing each pair to ascertain their uniqueness. The diversity score (DIV) is defined as $\text{DIV} = \frac{\#\text{Different pairs}}{\#\text{All pairs}}$.

Table 2: The accuracy and diversity score for pattern translation

| | Undirected pattern | | | | | Directed pattern | | | | | |
| | Triangle | T-Triangle | Square | Diamond | House | V-S | FBL | FFL | D-Diamond | | |
| | ACC DIV | ACC DIV | ACC DIV | ACC DIV | ACC DIV | ACC DIV | ACC DIV | ACC DIV | ACC DIV | Avg. A | Avg. D |
|---|---|---|---|---|---|---|---|---|---|---|---|
| Mixtral | **1.00** 0.00 | 0.00 0.00 | 0.00 0.00 | **1.00** 0.25 | 0.00 0.00 | 0.00 0.00 | **1.00** 0.11 | 0.02 0.00 | **1.00** 0.04 | 0.45 | 0.04 |
| Gemini | 0.52 **0.82** | 0.46 0.32 | 0.24 0.90 | 0.84 **0.88** | 0.50 0.23 | 0.78 0.76 | 0.90 0.89 | 0.90 0.73 | 0.94 **0.91** | 0.67 | 0.71 |
| Claude | **1.00** 0.61 | 0.02 0.00 | 0.24 0.67 | 0.78 0.58 | **0.88** 0.62 | 0.20 0.53 | **1.00** 0.29 | 0.90 0.42 | 0.84 0.42 | 0.65 | 0.46 |
| Llama | **1.00** 0.48 | 0.00 0.00 | 0.86 0.75 | **1.00** 0.80 | 0.12 0.80 | 0.00 0.00 | **1.00** 0.90 | **0.98** 0.77 | **1.00** 0.00 | 0.66 | 0.50 |
| GPT-4 | **1.00** 0.12 | 0.35 **0.75** | 0.98 0.33 | 0.92 0.76 | 0.86 0.80 | 0.42 0.80 | **1.00** 0.28 | **0.98** 0.43 | **1.00** 0.54 | 0.83 | 0.53 |
| GPT-4o | 0.98 0.29 | **0.92** 0.46 | 0.74 0.80 | 0.82 0.77 | 0.68 0.41 | **0.88** 0.05 | **1.00** 0.64 | 0.82 **0.81** | 0.68 0.54 | 0.84 | 0.53 |
| O1-mini | **1.00** 0.79 | 0.62 0.35 | **1.00** 0.80 | 0.74 0.87 | 0.66 **0.95** | 0.82 **0.85** | 0.98 0.75 | 0.96 0.80 | 0.98 0.84 | **0.86** | **0.78** |
| AVG. | 0.93 0.44 | 0.34 0.32 | 0.27 0.63 | 0.87 0.69 | 0.53 0.54 | 0.44 0.49 | 0.98 0.55 | 0.79 0.57 | 0.92 0.47 | 0.71 | 0.51 |

We repeat experiments 50 times for each pattern and LLM, and present the results in Table 2. Most LLMs effectively understand primitive graph patterns, demonstrating their ability to translate terminology-based descriptions into graph structures. Among the models tested, O1-mini stands out with the highest ACC and DIV on average. LLMs perform badly in T-triangle and V-S patterns because LLMs prefer to add one extra edge to these patterns to make them into other unintended patterns. In addition, LLMs show biased outputs, resulting in low DIV scores for simple patterns.

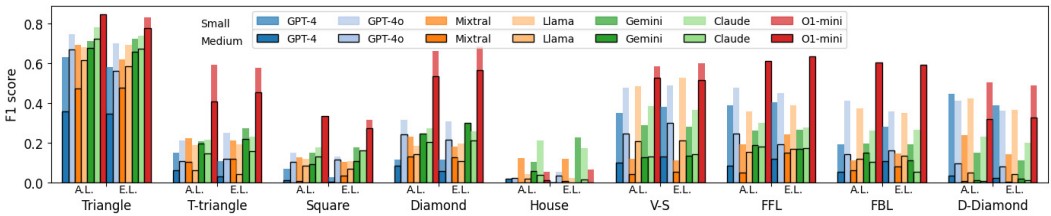

Figure 1: The F1 score for terminology-based pattern detection (small and medium scale)

For instance, when processing triangle patterns, GPT-4o often outputs a simple graph containing a triangle with a chain.

## 3.2 GRAPH MODIFICATION

In graph modification, we construct a dataset of input graphs, each including a specific primitive pattern, and then prompt LLMs to transform one pattern into another. Specifically, we define the following modifications: Square to House (S → H), Square to Diamond (S → D), Diamond to Square (D → S), and FFL to FBL (F → B). They contain edit operations like adding several edges, removing one edge, or changing the direction of an edge. A prompt example is "Modify the input graph to include a house pattern. The input graph G is . . . . The number of modified edges should be minimized." We evaluate if LLMs modify graphs to include the target pattern by the success rate.

The results are presented in Table 3. We observe that O1-mini stands out in its ability to edit diverse terminology-based patterns. Additionally, the scale of the input graphs generally doesn't have a major impact. This is because LLMs generally prioritize high-degree nodes and their neighbors to form the pattern. In larger graphs, LLMs tend to identify more regions for potential edits. These regions are local and invariant to the graph size.

Table 3: The success rate for terminology-based graph modification

| | | Gemini | | | Mixtral | | | Llama | | | Claude | | | GPT-4 | | | GPT-4o | | | O1-mini | |
|---|---|---|---|---|---|---|---|---|---|---|---|---|---|---|---|---|---|---|---|---|---|---|
| | | S. | M. | L. | S. | M. | L. | S. | M. | L. | S. | M. | L. | S. | M. | L. | S. | M. | L. | S. | M. | L. |
| S → H | A.L. | 0.06 | 0.28 | 0.20 | 0.32 | 0.58 | **0.64** | 0.34 | 0.48 | 0.40 | 0.26 | 0.50 | 0.48 | 0.14 | 0.12 | 0.14 | 0.28 | 0.36 | 0.44 | **0.38** | **0.64** | 0.58 |
| | E.L | 0.13 | 0.36 | 0.40 | 0.31 | 0.48 | **0.72** | 0.26 | 0.48 | 0.54 | 0.28 | 0.48 | 0.14 | 0.09 | 0.18 | 0.14 | 0.34 | 0.46 | 0.58 | **0.37** | **0.74** | 0.64 |
| S → D | A.L. | 0.53 | 0.48 | 0.32 | 0.77 | 0.64 | 0.62 | 0.50 | 0.32 | 0.50 | 0.74 | 0.50 | 0.26 | 0.18 | 0.16 | 0.20 | 0.71 | 0.68 | 0.54 | **0.95** | **0.98** | **0.94** |
| | E.L | 0.53 | 0.28 | 0.22 | 0.82 | 0.76 | 0.86 | 0.42 | 0.36 | 0.44 | 0.78 | 0.64 | 0.34 | 0.42 | 0.38 | 0.32 | 0.79 | 0.72 | 0.64 | **0.97** | **0.92** | **0.96** |
| D → S | A.L. | 0.27 | 0.42 | 0.28 | 0.29 | 0.28 | 0.40 | 0.51 | 0.70 | 0.64 | 0.54 | 0.84 | 0.68 | 0.15 | 0.12 | 0.24 | 0.41 | 0.44 | 0.28 | **0.77** | **0.72** | **0.88** |
| | E.L | 0.15 | 0.52 | 0.48 | 0.29 | 0.26 | 0.32 | 0.43 | 0.56 | 0.40 | 0.38 | 0.64 | 0.52 | 0.14 | 0.24 | 0.32 | 0.36 | 0.28 | 0.20 | **0.78** | **0.82** | **0.88** |
| F → B | A.L. | 0.30 | 0.58 | 0.52 | 0.29 | 0.56 | 0.48 | 0.29 | 0.60 | 0.50 | 0.36 | 0.56 | 0.62 | 0.26 | 0.42 | 0.46 | 0.18 | 0.24 | 0.20 | **0.40** | **0.76** | **0.74** |
| | E.L | 0.22 | 0.42 | 0.44 | 0.28 | 0.52 | 0.36 | 0.28 | 0.58 | 0.50 | 0.24 | 0.56 | 0.34 | 0.15 | 0.40 | 0.16 | 0.18 | 0.18 | 0.22 | **0.34** | **0.76** | 0.64 |

## 3.3 PATTERN DETECTION

In the pattern detection task, we prompt LLMs to detect specific primitive graph patterns in the input graph. We randomly generate thousands of non-isomorphic input graphs, with three graph scales: small (5–15 nodes), medium (15–25 nodes), and large (25–35 nodes). The example prompt is "Identify the occurrence of the given pattern in the input graph. The pattern is a diamond, defined as a 4-node motif with five edges. The input graph is . . . ." For each pattern, we calculate the F1 score to evaluate LLM performance in terms of precision and recall.

The results for small and medium scales are shown in Figure 1, with the full results provided in Appendix Table 18. Overall, the performance of most LLMs declines as the pattern becomes more complex and the input graph grows larger. This is because LLMs must examine every pair of nodes to determine whether they align with the pattern. Notably, O1-mini is less affected by the graph scale, as its step-by-step approach may help mitigate this issue.

## 4 TOPOLOGY-BASED PATTERN

In this section, we first perform pattern isomorphic mapping to examine LLMs' understanding of topology-based graph patterns. Subsequently, we conduct graph modification and pattern detection tasks to further evaluate LLMs' capabilities with various patterns and graph scales.

Table 4: The accuracy for isomorphic mapping

|  | GPT-4 | | GPT-4o | | Mixtral | | Llama | | Gemini | | Claude | | O1-mini | |
| --- | --- | --- | --- | --- | --- | --- | --- | --- | --- | --- | --- | --- | --- | --- |
|  | A.L. | E.L | A.L. | E.L | A.L. | E.L | A.L. | E.L | A.L. | E.L | A.L. | E.L | A.L. | E.L |
| Small | 0.30 | 0.83 | 0.30 | 0.94 | 0.20 | 0.93 | 0.42 | **0.96** | 0.22 | 0.39 | 0.41 | 0.94 | **0.77** | 0.66 |
| Medium | 0.08 | 0.34 | 0.00 | 0.82 | 0.00 | 0.64 | 0.24 | 0.86 | 0.00 | 0.14 | 0.18 | **0.92** | 0.32 | 0.28 |
| Large | 0.06 | 0.14 | 0.00 | 0.94 | 0.04 | 0.10 | 0.14 | 0.68 | 0.00 | 0.02 | 0.20 | **1.00** | 0.00 | 0.32 |

## 4.1 GRAPH ISOMORPHIC MAPPING

In a topology-based description, isomorphic graphs can appear different due to the distinct naming of nodes. Thus, LLMs may mistakenly recognize isomorphic graphs as distinct entities. In this task, we test whether LLMs find a one-to-one correspondence (bijection) between the node sets of two isomorphic graphs. We reuse the graph datasets in the terminology-based pattern detection task. For each graph, the original node IDs range from 0 to 35. We shuffle and relabel the node IDs to span from 100 to 135 to create the isomorphic graphs. Each sample is guaranteed to have at least one valid mapping solution. We employ accuracy as the evaluation metric, measuring how effectively LLMs map node IDs correctly between two graphs.

The results, presented in Table 4, show that most LLMs perform well on small datasets, but their performance degrades as the graph scale increases. However, GPT-4o and Claude still maintain high accuracy using the edge list format. Consequently, we select medium- and large-scale graphs for further analysis. Upon manually reviewing the output text from GPT-4o, Claude, and O1-mini, we found that there exist two algorithms to help LLMs reason whether the graphs are isomorphic. One approach involves first calculating node degrees and then mapping nodes with matching degrees, while the other compares the edge list directly. Due to the LLMs' inherent difficulty with counting, results based on degree counting tend to be less reliable. As shown in Figure 2, O1-mini uses the degree counting approach with the proportion 89%. Only about 30% of the outputs from Claude and GPT-4o use degree counting, with the majority relying on edge comparison. As LLMs

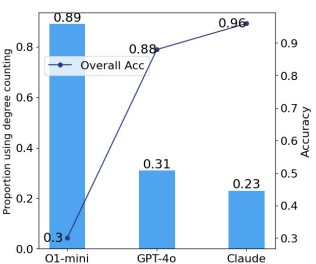

Figure 2: The influence of underlying algorithms used in pattern isomorphic mapping

are not good at counting, the accuracy of Claude and GPT-4o is significantly higher than O1-mini.

## 4.2 GRAPH MODIFICATION

We use the same setting as that in the terminology-based pattern detection, except the descriptions are now replaced with topology-based descriptions. Note that since the graph nodes are represented by numerical IDs, we use uppercase letters for pattern nodes to prevent confusion for LLMs. The results are shown in Table 5. Similar to the terminology-based case, O1-mini still delivers the best performance, and the graph scale has little impact on the outcome. However, the average accuracy for topology-based descriptions is lower than that for terminology-based ones, likely due to increased hallucinations. For instance, when modifying diamond graphs to include square patterns using the edge list format, the presence of squares within diamond structures prevents LLMs from exhibiting any change.

Table 5: The success rate for topology-based graph modification

|  |  | Gemini | | | Mixtral | | | Llama | | | Claude | | | GPT-4 | | | GPT-4o | | | O1-mini | | |
| --- | --- | --- | --- | --- | --- | --- | --- | --- | --- | --- | --- | --- | --- | --- | --- | --- | --- | --- | --- | --- | --- | --- |
|  |  | S. | M. | L. | S. | M. | L. | S. | M. | L. | S. | M. | L. | S. | M. | L. | S. | M. | L. | S. | M. | L. |
| S → H | A.L. | 0.20 | 0.34 | 0.20 | **0.78** | 0.58 | 0.66 | 0.57 | 0.52 | 0.48 | 0.45 | 0.64 | 0.44 | 0.34 | 0.56 | 0.56 | 0.67 | 0.70 | 0.68 | 0.61 | **0.82** | **0.80** |
|  | E.L | 0.22 | 0.42 | 0.26 | **0.66** | **0.68** | 0.54 | 0.45 | 0.50 | 0.48 | 0.38 | 0.54 | 0.42 | 0.24 | 0.60 | 0.52 | 0.36 | 0.52 | 0.60 | 0.46 | **0.68** | **0.74** |
| S → D | A.L. | 0.45 | 0.34 | 0.32 | 0.78 | 0.66 | 0.60 | 0.65 | 0.50 | 0.74 | 0.69 | 0.62 | 0.60 | 0.48 | 0.56 | 0.40 | 0.75 | 0.68 | 0.66 | **0.82** | **0.76** | **0.86** |
|  | E.L | 0.48 | 0.40 | 0.52 | 0.67 | 0.70 | 0.62 | 0.69 | 0.46 | 0.62 | 0.72 | 0.74 | 0.74 | 0.67 | 0.62 | 0.20 | 0.67 | 0.68 | 0.74 | **0.88** | **0.86** | **0.84** |
| D → S | A.L. | 0.12 | 0.24 | 0.12 | **0.58** | 0.42 | 0.60 | 0.29 | 0.26 | 0.40 | 0.12 | 0.26 | 0.36 | 0.16 | 0.48 | 0.40 | 0.41 | 0.40 | 0.48 | 0.55 | **0.56** | **0.64** |
|  | E.L | 0.11 | 0.32 | 0.32 | **0.40** | 0.32 | 0.44 | 0.29 | 0.30 | 0.20 | 0.11 | 0.22 | 0.48 | 0.06 | 0.30 | 0.20 | 0.20 | 0.18 | **0.36** | 0.24 | **0.36** | 0.28 |
| F → B | A.L. | 0.28 | 0.38 | 0.50 | 0.52 | 0.40 | 0.64 | 0.55 | 0.68 | 0.70 | 0.37 | 0.54 | 0.40 | 0.48 | 0.68 | 0.76 | **0.57** | **0.80** | **0.80** | 0.41 | 0.76 | 0.68 |
|  | E.L | 0.20 | 0.24 | 0.32 | 0.34 | 0.52 | 0.40 | 0.41 | 0.50 | 0.44 | 0.29 | 0.44 | 0.34 | 0.27 | 0.46 | 0.48 | **0.44** | 0.62 | 0.52 | 0.32 | **0.66** | **0.66** |

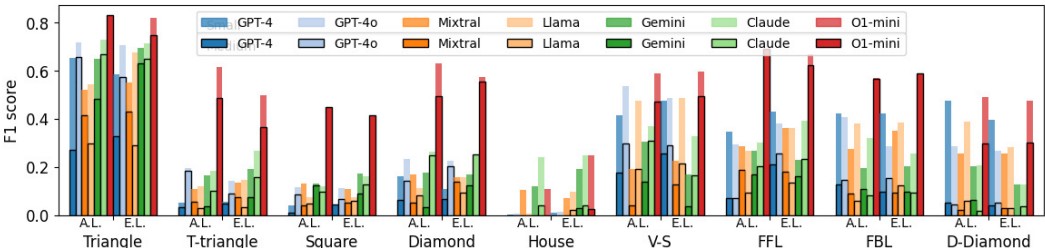

Figure 3: The F1 score of topology-based pattern detection (small and medium scale)

## 4.3 PATTERN DETECTION

We use the same setting as that in the terminology-based pattern detection, except the pattern descriptions are now topology-based descriptions. The results are shown in Figure 3, with full results provided in Appendix Table 19. The performance of LLMs is also influenced by their capabilities, graph scales, and the complexity of patterns. For instance, while O1-mini achieves a high F1 score in detecting tailed-triangle, square and diamond patterns, other models only reach an F1 score of 20%. Moreover, although O1-mini performs well on small-scale graphs, it fails on medium-scale datasets. Among the patterns, the house pattern is particularly difficult to detect, as it is the most complex pattern in our setting.

Table 6: The precision of nodes in k-core detection

|  | GPT-4 | | GPT-4o | | Mixtral | | Llama | | Gemini | | Claude | | O1-mini | |
| --- | --- | --- | --- | --- | --- | --- | --- | --- | --- | --- | --- | --- | --- | --- |
|  | A.L. | E.L | A.L. | E.L | A.L. | E.L | A.L. | E.L | A.L. | E.L | A.L. | E.L | A.L. | E.L |
| Small | **0.63** | 0.62 | 0.61 | 0.62 | 0.62 | 0.61 | **0.63** | **0.63** | 0.61 | 0.60 | **0.63** | **0.63** | **0.63** | **0.63** |
| Medium | 0.73 | 0.62 | **1.00** | 0.85 | 0.84 | 0.80 | 0.78 | 0.84 | 0.80 | 0.80 | 0.88 | **0.88** | 0.88 | **0.88** |
| Large | 0.66 | 0.52 | **1.00** | **1.00** | 0.90 | 0.80 | 0.88 | **1.00** | 0.84 | 0.76 | **1.00** | **1.00** | **1.00** | **1.00** |

## 5 DATA-DRIVEN PATTERNS

In this section, we focus on three classic data-driven patterns that are useful in many areas: densely connected subgraphs, frequent subgraphs within graphs, and discriminative patterns based on labels. By examining densely connected subgraphs, such as k-cores, we can identify communities and make friendship recommendations in social networks. Frequent subgraphs help uncover common structural motifs, which is valuable in chemistry and biology. Discriminative patterns enable us to distinguish between different classes within the data, aiding in classification tasks.

### 5.1 DENSE SUBGRAPH MINING (K-CORE)

K-core is a commonly used densely connected subgraph, defined as a maximal subgraph containing nodes of degree $k$ or more. In this task, we prompt LLMs to leverage the k-core algorithm (Batagelj & Zaversnik, 2003) to identify 3-core in the input graph. We compute Precision as the metric by judging whether the identified nodes belong to a 3-core and present the results in Table 6.

On average, LLMs achieve over 60% precision across various graph scales, demonstrating their ability to execute the algorithm effectively. Notably, LLMs obtain better performance on large-scale graphs than on smaller ones, likely because most nodes in large graphs have degrees greater than 3. LLMs tend to make errors when node degrees are close to 3 but become increasingly accurate as node degrees increase. In Figure 4, we select GPT-4o and O1-mini to analyze the relationship between node degrees and precision scores. Nodes with a degree of 3 have 50% precision, while those with degrees above 5 reach nearly 100% precision.

### 5.2 FREQUENT SUBGRAPH EXTRACTION

Mining frequent subgraphs is an important task on graphs, defined as finding subgraphs that appear frequently in a graph dataset given a frequency threshold. For each pattern, we first generate a graph dataset, ensuring that each graph contains the target pattern. The statistics of datasets are provided in

Table 7: The accuracy of extracted frequent subgraphs

| | | GPT-4 | | | GPT-4o | | | Mixtral | | | Llama | | | Gemini | | | Claude | | | O1-mini | | |
|---|---|---|---|---|---|---|---|---|---|---|---|---|---|---|---|---|---|---|---|---|---|---|
| | | S. | M. | L. | S. | M. | L. | S. | M. | L. | S. | M. | L. | S. | M. | L. | S. | M. | L. | S. | M. | L. |
| Triangle | A.L. | 1.00 | 1.00 | 1.00 | 0.84 | 0.88 | 0.88 | 0.86 | 0.88 | 1.00 | 0.62 | 0.68 | 0.87 | 1.00 | 1.00 | 0.59 | 0.43 | 0.26 | 0.09 | 1.00 | 1.00 | 1.00 |
| | E.L | 1.00 | 1.00 | 1.00 | 0.94 | 1.00 | 0.81 | 1.00 | 1.00 | 1.00 | 0.94 | 0.95 | 1.00 | 1.00 | 0.57 | 0.29 | 0.84 | 0.58 | 0.32 | 1.00 | 1.00 | 1.00 |
| Square | A.L. | 1.00 | 1.00 | 1.00 | 1.00 | 0.86 | 1.00 | 0.69 | 0.95 | 1.00 | 0.61 | 0.94 | 1.00 | 0.75 | 0.47 | 0.40 | 0.30 | 0.37 | 0.07 | 1.00 | 1.00 | 1.00 |
| | E.L | 1.00 | 1.00 | 1.00 | 0.97 | 0.71 | 0.98 | 1.00 | 1.00 | 1.00 | 0.94 | 0.93 | 0.91 | 0.80 | 1.00 | 0.17 | 0.93 | 0.67 | 0.21 | 1.00 | 1.00 | 1.00 |
| Diamond | A.L. | 1.00 | 1.00 | 1.00 | 0.89 | 1.00 | 0.50 | 0.92 | 1.00 | 1.00 | 0.43 | 0.52 | 0.96 | 1.00 | 1.00 | 0.83 | 0.34 | 0.33 | 0.14 | 1.00 | 1.00 | 1.00 |
| | E.L | 1.00 | 1.00 | 1.00 | 1.00 | 0.87 | 0.93 | 1.00 | 1.00 | 1.00 | 0.99 | 0.93 | 0.91 | 1.00 | 0.38 | 0.52 | 0.99 | 0.68 | 0.32 | 1.00 | 1.00 | 1.00 |
| House | A.L. | 1.00 | 1.00 | 1.00 | 1.00 | 1.00 | 0.98 | 0.73 | 1.00 | 1.00 | 0.53 | 0.66 | 0.93 | 1.00 | 0.78 | 0.10 | 0.31 | 0.32 | 0.22 | 1.00 | 1.00 | 1.00 |
| | E.L | 1.00 | 1.00 | 1.00 | 1.00 | 1.00 | 0.76 | 1.00 | 1.00 | 1.00 | 1.00 | 0.74 | 0.95 | 1.00 | 0.33 | 0.18 | 1.00 | 0.55 | 0.44 | 1.00 | 1.00 | 1.00 |
| FFL | A.L. | 1.00 | 1.00 | 1.00 | 0.91 | 0.64 | 1.00 | 0.89 | 1.00 | 0.86 | 0.67 | 0.68 | 0.93 | 0.89 | 0.00 | 1.00 | 0.33 | 0.32 | 0.26 | 1.00 | 1.00 | 1.00 |
| | E.L | 1.00 | 1.00 | 1.00 | 0.96 | 0.67 | 0.92 | 1.00 | 1.00 | 1.00 | 0.92 | 1.00 | 1.00 | 1.00 | 1.00 | 1.00 | 0.68 | 0.52 | 0.34 | 1.00 | 1.00 | 1.00 |
| FBL | A.L. | 1.00 | 1.00 | 1.00 | 0.78 | 0.89 | 0.90 | 1.00 | 1.00 | 0.59 | 0.77 | 0.67 | 0.87 | 1.00 | 1.00 | 1.00 | 0.46 | 0.22 | 0.40 | 1.00 | 1.00 | 0.85 |
| | E.L | 1.00 | 1.00 | 1.00 | 0.98 | 0.75 | 0.83 | 1.00 | 1.00 | 1.00 | 0.96 | 1.00 | 1.00 | 1.00 | 1.00 | 1.00 | 0.84 | 0.45 | 0.31 | 1.00 | 1.00 | 1.00 |
| D-Diamond | A.L. | 1.00 | 1.00 | 1.00 | 0.81 | 0.95 | 1.00 | 1.00 | 1.00 | 0.67 | 0.54 | 0.73 | 0.91 | 0.67 | 0.00 | 1.00 | 0.41 | 0.42 | 0.35 | 1.00 | 1.00 | 1.00 |
| | E.L | 1.00 | 1.00 | 1.00 | 0.85 | 0.88 | 0.66 | 1.00 | 0.60 | 1.00 | 0.78 | 1.00 | 1.00 | 1.00 | 0.75 | 1.00 | 0.61 | 0.46 | 0.28 | 1.00 | 1.00 | 1.00 |

Table 8: Results for discriminative pattern learning by labels

| | GPT-4 | | GPT-4o | | Mixtral | | LLAMA | | Gemini | | Claude | | O1-mini | |
|---|---|---|---|---|---|---|---|---|---|---|---|---|---|---|
| | ACC | D.P. | ACC | D.P. | ACC | D.P. | ACC | D.P. | ACC | D.P. | ACC | D.P. | ACC | D.P. |
| A.L. | 0.82 | 0.66 | 0.63 | 0.25 | 0.52 | 0.33 | 0.80 | 0.43 | **0.91** | **0.70** | 0.79 | 0.68 | 0.77 | 0.31 |
| E.L. | 0.97 | 0.47 | **1.00** | 0.58 | 0.75 | 0.12 | 0.89 | 0.71 | 0.99 | **0.70** | **1.00** | 0.64 | 0.72 | 0.08 |

Table 12. In each turn, we randomly select 10 graphs from the dataset, task LLMs to extract frequent patterns based on these selected graphs, and output patterns in the topology-based description. We repeat this process 100 times and calculate the accuracy as the percentage of cases where the output pattern appears in more than 60% of the selected graphs. It is worth noting that the extracted pattern does not need to exactly match the tested pattern. For example, if the LLMs identify a triangle pattern during testing with a house pattern, we still consider this an accurate outcome. The accuracy and frequency of extracted patterns are summarized in Table 7 and Figure 5, respectively.

In Table 7, LLMs exhibit a strong capability in identifying frequent subgraphs, with GPT-4 and O1-mini showing impressive performance. However, LLMs are prone to detect simpler patterns rather than more complex ones in the dataset. In Figure 5, we illustrate the accuracy when different models extract the defined patterns in the datasets. Although various patterns appear with similar probability across different datasets, LLMs predominantly identify triangles. The house pattern, a combination of a triangle and square, is rarely recognized.

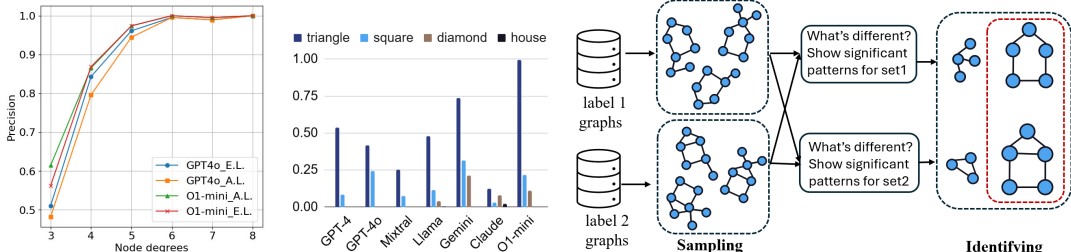

Figure 4: The precision in various node degrees

Figure 5: The frequency of extracted patterns

Figure 6: The pipeline of discriminative pattern learning by labels

## 5.3 DISCRIMINATIVE PATTERN LEARNING

This task is to extract and learn important patterns from input graphs to discriminate labels. The labels, which typically represent categories or outcomes, act as a guide for discovering the most significant patterns for the task. We implement a two-step process to instruct LLMs, as illustrated in Figure 6. We first sample an equal number of graphs from each label to form a dataset. Then, we prompt LLMs to identify discriminative patterns that differentiate one label from another. This process is repeated multiple times, and all extracted patterns are retained. We further filter these patterns: for each pattern, we ensure that over 90% of graphs with the corresponding label contain the identified pattern, while 90% of graphs with other labels do not. The remaining are regarded as discriminative patterns. We use two metrics to evaluate the discriminative patterns. We introduce an extra classification task to measure the effectiveness, where we ask LLMs to predict the labels of new graphs based on the discriminative patterns and compute the prediction accuracy. In addition,

we calculate the discriminative pattern ratio as D.P. $= \frac{\#\text{Discriminative patterns}}{\#\text{Extracted patterns}}$, which assesses the efficiency of the extraction process.

The results in Table 8 demonstrate that most LLMs successfully perform graph classification tasks, particularly when using the edge list format. Interestingly, a high D.P. score does not always linearly correspond to high accuracy. For instance, despite having a D.P. score of 0.08, O1-mini still achieves a classification accuracy of 0.72.

# 6 EVALUATION ON REAL-WORLD GRAPHS

In this section, we gather a range of real-world datasets across diverse domains of graphs, including molecules, social networks, bioinformatics, and computer vision. Different from synthetic dataset, the real-world graphs have node attributes and potential noise in the graph structures, which makes the graph pattern tasks more challenging. More details of datasets and prompts are provided in the appendix B. In this evaluation, we focus on pattern detection and discriminative pattern learning.

**Molecule Pattern Detection.** Followed by Section 3.3 and Section 4.3, we perform pattern detection in molecular graphs, where the goal is to ask LLMs to detect the target pattern in the given molecule. We evaluate three datasets: Benzene, Alkane-Carbonyl (R-CO), and Fluoride-Carbonyl (F-CO), where the target patterns are hexagons, alkane groups, and fluoride groups, respectively. We compare the terminology-based with topology-based patterns, and also assess a combined approach, referred to as Both.

Table 9: The F1 score for pattern detection in molecules

|  |  | Terminology-based | | | Topology-based | | | Both | | |
|---|---|---|---|---|---|---|---|---|---|---|
|  |  | Benzene | R-CO. | F-CO | Benzene | R-CO | F-CO | Benzene | R-CO | F-CO |
| GPT-4 | A.L. | 0.88 | 0.73 | 0.68 | 0.71 | 0.64 | 0.70 | 0.89 | 0.71 | 0.69 |
|  | E.L. | 0.85 | 0.71 | 0.67 | 0.72 | 0.64 | 0.69 | 0.83 | 0.69 | 0.70 |
| GPT-4o | A.L. | 0.88 | 0.66 | 0.67 | 0.76 | 0.66 | 0.69 | 0.87 | 0.68 | 0.67 |
|  | E.L. | 0.90 | 0.66 | 0.67 | 0.78 | 0.67 | 0.69 | 0.86 | 0.65 | 0.67 |

Table 9 illustrates that terminology-based descriptions outperform topology-based ones for both the Benzene and R-CO datasets. However, combining both descriptions does not enhance performance, as the final F1 scores are slightly lower than using terminology alone. This indicates that in real-world applications, LLMs may rely on internal knowledge, such as retrieving the name of patterns, to understand molecular structures.

Table 10: Discriminative patterns and classification result in real-world datasets

|  | Binary classification | | | | | | | | Multi-label classification | | | | | |
|---|---|---|---|---|---|---|---|---|---|---|---|---|---|---|
|  | MUTAG | | OGBG-MOL | | OGBG-BBBP | | IMDB-BINARY | | ENZYYMES | | FINGERPRINT | | IMDB-MULTI | |
|  | A.L. | E.L. | A.L. | E.L. | A.L. | E.L. | A.L. | E.L. | A.L. | E.L. | A.L. | E.L. | A.L. | E.L. |
| GPT-4 | 0.66 | 0.70 | 0.56 | 0.58 | 0.64 | 0.66 | 0.72 | 0.78 | 0.40 | 0.38 | 0.36 | 0.36 | 0.33 | 0.35 |
| GPT-4o | 0.61 | 0.63 | 0.52 | 0.60 | 0.68 | 0.70 | 0.68 | 0.72 | 0.40 | 0.35 | 0.35 | 0.34 | 0.34 | 0.35 |

Discriminative Patterns

**Discriminative Pattern Learning and Classification.** Based on Section 5.3, we conduct the 2-step process of discriminative pattern learning and classification on real-world datasets. We use hundreds of samples for discrimination and test about 50 samples in classification. Further details can be found in Appendix Table 13.

Table 10 presents the results for both binary and multi-label classification accuracy on the molecular (MUTAG, OGB-MOL, OGB-BBBP), social network (IMDB-BINARY, IMDB-MULTI), bioinformation (ENZYYMES), and computer vision (FINGERPRINT) datasets. Notably, we highlight the most discriminative patterns extracted by GPT-4, which provide meaningful insights into the given dataset. For instance, in the MUTAG dataset, NO2 groups are marked as positively influencing mutagenicity (Agarwal et al., 2023). Due to the current high cost of LLMs, we performed limited sampling during the discrimination process. We believe that increasing the sampling size in this stage would improve the overall classification score.

## 7    DISCUSSIONS

In this section, we first provide our theoretical intuitions from the perspective of expressiveness and then summarize key insights from our observations.

### 7.1    THEORETICAL INTUITIONS

Here we discuss the theoretical intuitions of LLMs' capabilities in graph pattern recognition from an expressiveness standpoint, often used in analyzing Transformers (Feng et al., 2024; Li et al., 2024; de Luca & Fountoulakis, 2024). We demonstrate that Transformers, given graph inputs, can be configured to perform graph pattern tasks. The LOCAL framework Angluin (1980) is a distributed computing paradigm, implemented by the message passing among the nodes. Many graph pattern algorithms such as pattern detection can be efficiently implemented by LOCAL (Drucker et al., 2014; Korhonen & Rybicki, 2017). Intuitively if Transformers can simulate any LOCAL algorithms, it has the capability for graph pattern tasks. We build the following theorem for this intuition.

**Theorem 1.** *(Informal) For any LOCAL algorithm A, there exists a Transformer with edge list as input that can simulate A.*

The detailed proof is given in Appendix A. The proof intuition is that the attention mechanism is a message-passing between the tokens and each token is executed in a parallel manner. Consequently, this mechanism enables Transformers to simulate the message passing between nodes. The above theorem suggests the existence of such weights.

### 7.2    EMPIRICAL INSIGHTS

**LLMs use diverse algorithms for one task, and the performance varies due to their execution ability.** We provide two observations: (1) We manually reviewed most of the outputs generated by LLMs in graph mining tasks, and summarized the algorithms used by LLMs in Appendix H. Our analysis reveals that different LLMs utilize diverse algorithms to solve the same problem. For instance, more than eight algorithms are used for pattern detection tasks (Section 3.3). (2) Due to the internal flaws of LLMs, these algorithms, although logically correct, will have different performance. In the graph isomorphic mapping task (Section 4.1), a common algorithm starts by counting node degrees and then mapping nodes. O1-mini uses this approach for 89% of the data but achieves only 30% accuracy due to errors in degree counting. In contrast, Claude applies degree counting to only 23% of the data, relying primarily on a direct edge-matching algorithm for the rest. This alternative strategy enables Claude to achieve an impressive 96% accuracy.

**Input format that aligns with the pretrained knowledge improves the performance.** Our experiments demonstrate that LLMs have basic knowledge about graph patterns and they help LLMs in graph pattern tasks. First, LLMs are pre-trained on extensive internet datasets where graph patterns are often described using specific terminologies. This exposure helps LLMs understand these terms. Comparing the results in Section 3.3 with those in Section 4.3, we observe that terminology-based graph pattern detection generally outperforms topology-based detection. This suggests that LLMs leverage their internal knowledge to enhance performance when provided with terminology as input. Second, the pretrained knowledge will influence the strategies employed by LLMs, and the graph input format that aligns with the strategies will improve the performance. For example, in the case of discriminative pattern learning (Section 5.3), the algorithms used by LLMs often rely on comparing corresponding edges in two graphs. In this scenario, the edge list format typically leads to better performance than the adjacency list format. Conversely, in k-core detection (Section 5.1), the algorithms require counting node degrees, so the edge list is inferior to the adjacency list.

**O1-mini often gives the best results, but not always.** In most tasks, O1-mini has the best performance, especially when the graph size is large. This is not surprising because it has a much longer chain-of-thought. For example, in the pattern detection task, which requires LLMs to search node combinations one by one, O1-mini holds a clear advantage. However, it falls back in the discriminative pattern learning by labels in Section 5.3. The reason is that O1-mini strictly searches for patterns that exactly meet the data requirements, without allowing for fuzzy matching. This results in smaller discriminative patterns being found. Consequently, when O1-mini attempts graph classification, the patterns struggle to generalize, leading to lower classification accuracy.

## 8 RELATED WORK

**Large Language Models for Graphs.** Here, we introduce previous work using LLMs based on the types of graph tasks they focused on. The first category is to solve graph algorithm tasks, such as connectivity, degree counting, cycle check, shortest path, and Hamilton path. NLGraph (Wang et al., 2024) and Talk like a graph (Fatemi et al., 2023) translated graph structures into natural language, and leveraged techniques like few-shot prompting or chain-of-thought (CoT) to feed these descriptions directly into LLMs for inference. Since exact computational algorithms can be employed to obtain an accurate reasoning process and results, GraphInstruct (Luo et al., 2024) built the instruction-tuning dataset and fine-tuned LLMs by Lora. Moreover, Graphwiz (Chen et al., 2024a) extended the scale and diversity of the dataset and applied Direct Preference Optimization (DPO) loss to enhance the performance. Recently, GraphToken (Perozzi et al., 2024) utilized a GNN-like encoder to process the input graph and train this encoder with a frozen LLM.

The second category encompasses graph learning tasks, which typically include node classification, link prediction, and graph classification (Chen et al., 2024c). Unlike graph algorithm tasks relying on deterministic rules for inference, graph learning tasks involve learning the relationship between graphs and labels using provided training examples. Graph-LLM (Chai et al., 2023) and GraphText (Zhao et al., 2023) designed prompts that convert graph structure, node/edge attributes, and label information into natural language, enabling LLMs to make predictions without additional training. In contrast, other approaches incorporate projectors or encoders before LLMs and construct datasets for tuning. LLaGA (Chen et al., 2024b) and MuseGraph (Tan et al., 2024) reorganized the center node and its neighborhood information as a structure-aware textual description, which is then tokenized by a projector and fed into LLMs to predict node labels in citation graphs. GraphGPT (Tang et al., 2024) utilized a GNN encoder to tokenize the input graph before processing it with the LLM to address node classification in text-attributed graphs. MolCA (Liu et al., 2023) and InstructMol (Cao et al., 2023) used similar GNN encoders for molecular structures to predict their properties.

**Graph Pattern Discovery.** Graph patterns (also known as substructures, graphlets, motifs and subgraphs) have been extensively studied before the era of LLMs due to their crucial role in real-world applications, particularly in chemistry (Murray & Rees, 2009), biology (Hu et al., 2005), and social science (Wu et al., 2022). Numerous algorithms have been proposed for pattern discovery. For example, the algorithms designed to identify and count dense pre-defined patterns, such as k-core, k-cycle, and k-clique, were introduced in Milo et al. (2002); Zhang & Parthasarathy (2012). Frequent subgraph mining algorithms were developed to efficiently search for possible patterns (Yan et al., 2008; Elseidy et al., 2014), aiming to reduce the search space as much as possible.

In recent years, deep learning, particularly graph neural networks (GNNs), has become prominent in this area. Many studies (Chen et al., 2020; Zhang et al., 2023; Luo et al., 2022) investigated the theoretical guarantees of specific GNNs capable of identifying certain substructures, and developed subgraph-aware GNNs with strong expressiveness to improve the accuracy of graph learning tasks. More recently, some studies (Zheng et al., 2023) have begun exploring the use of LLMs to identify SMILES patterns in molecular data, opening new avenues for research.

As previously mentioned, few studies have addressed graph pattern discovery using LLMs, despite their importance for many downstream tasks, which still require further exploration.

## 9 CONCLUSION

We investigate how LLMs understand graph patterns. While recent studies demonstrate that LLMs have shown success in various graph-related tasks, the comprehension of graph patterns remains unexplored. To bridge this gap, we propose a benchmark featuring both synthetic and real-world data, spanning 11 subtasks evaluated across 7 LLMs. Using known patterns defined through terminology-based and topology-based descriptions, we conduct tasks including pattern translation, isomorphic mapping, graph modification, and pattern detection. For unknown patterns, we cover classic tasks such as densely connected subgraph detection, frequent pattern mining and discriminative pattern learning using labeled graph data. The real-world datasets are gathered to further assess LLMs' ability to handle the attributes and noise in the pattern. Our results show that LLMs possess a preliminary capability to understand graph patterns and perform tasks related to them effectively.

ACKNOWLEDGMENTS

The work is supported by the Accelerating Foundation Model Research Program (AFMR), Microsoft Research. Xinnan Dai and Jiliang Tang are supported by the National Science Foundation (NSF) under grant numbers CNS2321416, IIS2212032, IIS2212144, IOS2107215, DUE2234015, CNS2246050, DRL2405483 and IOS2035472, the Army Research Office (ARO) under grant number W911NF-21-1-0198, Amazon Faculty Award, Meta, Microsoft and SNAP.

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

## A  PROOF FOR THEOREM 1

We first define the edge list format.

$$\underbrace{u_1\ v_1\ a_{u_1\leftarrow v_1}\ u_1\ v_2\ a_{u_1\leftarrow v_1}\ \ldots\ u_n\ v_n\ a_{u_n\leftarrow v_n}}_{\text{edge list}}\ \underbrace{u_1\ a_1\ \ldots\ u_n\ a_n}_{\text{initial states}} \quad (1)$$

where $u_1, v_1$ are the node id, $a_i$ is the node feature and $a_{u_1\leftarrow v_1}$ is the edge weights. The definition of message-passing graph neural networks.

**Theorem 2** (Formal version of Theorem 1). *Assume the input format is given in equation 1. Let $T_\ell(G_a)$ be the state $(x_1^{(\ell)}, \ldots, x_n^{(\ell)})$ of a Transformer network $T$ with edge list input in 1. For any LOCAL algorithm $N$ and small number $\epsilon$, there exists $T$ such that*

$$\|T_{O(\ell)}(G_a) - N_\ell(G_a)\|_\infty \leq \epsilon \quad \text{for every layer } \ell \text{ and } G_a \in \mathcal{G}_a,$$

*where $\mathcal{G}_a$ is the set of all attributed graphs.*

*Proof.* This theorem is a combination between Lemma 1 and Lemma 2. □

**Lemma 1** (Equivalence between MPGNNs and LOCAL). *Let $N_\ell(G_a)$ be the state $(x_1^{(\ell)}, \ldots, x_n^{(\ell)})$ of a MPGNN network $N$ and $A_\ell(G_a) = (s_1^{(\ell)}, \ldots, s_n^{(\ell)})$ that of a LOCAL algorithm $A$. For any algorithm $A$ there exists $N$ (resp. for any $N$ there exists $A$) such that*

$$A_\ell(G_a) = N_\ell(G_a) \quad \text{for every layer } \ell \text{ and } G_a \in \mathcal{G}_a.$$

---

**Algorithm 1** Message passing graph neural network Loukas (2019)

---

**Initialization:** Set $x_i^{(0)} = a_i$ for all $v_i \in \mathcal{V}$.
**for** layer $\ell = 1, \ldots, d$ **do**
  **for** every edge $e_{i\leftarrow j} \in \mathcal{E}^*$ (in parallel) **do**

$$m_{i\leftarrow j}^{(\ell)} = \text{MSG}_\ell\left(x_i^{(\ell-1)}, x_j^{(\ell-1)}, v_i, v_j, a_{i\leftarrow j}\right)$$

  **for** every node $v_i \in \mathcal{V}$ (in parallel) **do**

$$x_i^{(\ell)} = \text{UP}_\ell\Big(\sum_{v_j \in \mathcal{N}_i^*} m_{i\leftarrow j}^{(\ell)}\Big)$$

Set $x_i = x_i^{(d)}$.
**return** $x_i$ for every $v_i \in \mathcal{V}$.

---

**Lemma 2** (Representing MPGNNs by Transformers). *Assume the input format is given in equation 1 and MSG and UP in Algorithm 1 are implemented by MLPs. Let $T_\ell(G_a)$ be the state $(x_1^{(\ell)}, \ldots, x_n^{(\ell)})$ of a Transformer network $T$ with edge list input in 1. For any MPGNN $N$ and small number $\epsilon$, there exists $T$ such that*

$$\|T_{O(\ell)}(G_a) - N_\ell(G_a)\|_\infty \leq \epsilon \quad \text{for every layer } \ell \text{ and } G_a \in \mathcal{G}_a.$$

*Proof.* The proof is heavily built on the proof of Theorem 1 in Wu et al. (2024).

**Token Embedding and Positional Embedding:** The token embedding includes the token type $e^{\text{type1}}$ (0 for the node feature; 1 for node id; 2 for edge feature), refined token type $e^{\text{type2}}$ (0 for node feature, 1 for the node id tokens in initial state sentences, 2 for the target node tokens in the edge list, 3 for the source node tokens in the edge list, 4 for edge feature tokens), and the token id $e^{\text{token}}$ (from 0 to $|V| - 1$). The two-dimensional positional embedding includes the embedding for initial state tokens $e^{\text{pos1}}$ (0 for edge list tokens, 1 for the first two elements of initial state sentences, 2 for the second two elements of initial state sentences, etc.), embedding for edge list tokens $e^{\text{pos2}}$ (0 for initial state sentence tokens, 1 for the first three elements of the edge list, 2 for the second three elements of the edge list, etc.). There are also placeholders to put the intermediate states of MPGNNs.

**Block 1 - Node feature Preparation:** The goal of the first block is to broadcast the node feature $a_i$ from the initial state sentence tokens to node tokens in edge list. (1) Use MLPs to recover the digits of the node features $a_i$ and put them in the first placeholder if $e_k^{type} == 0$; (2) Copy the first placeholder from initial state sentence token to its previous node token by using **COPY** in Lemma 3 and setting $\mathcal{S}_k = \{j|(e_k^{\text{pos1}} - e_j^{\text{pos1}})^2 < \delta\}$; (3) Broadcast the first placeholder with **MEAN** in Lemma 3 and setting $\mathcal{S}_k = \{j|(e_k^{\text{type1}} - e_j^{\text{type1}})^2 + (e_k^{\text{token}} - e_j^{\text{token}})^2 < \delta\}$. Now the state for every node token $u_i$ is $[e^{\text{type1}}, e^{\text{type2}}, e^{\text{token}}, e^{\text{pos1}}, e^{\text{pos2}}, a_{u_i}]$; (4) Use MLP to put the token id into the second placeholder if $e^{\text{type1}} == 2$ or $e^{\text{type1}} == 3$. Now the state for every node token $u_i$ is $[e^{\text{type1}}, e^{\text{type2}}, e^{\text{token}}, e^{\text{pos1}}, e^{\text{pos2}}, a_{u_i}, u_i]$

**Block 2 - Edge feature Preparation:** The goal of the second block is to copy the prepare the edge features for the target node token, which will be used for message passing. (1) Use MLPs to recover the digits of the edge feature tokens and put them in the third placeholder if $e^{\text{type1}} == 2$; (2) Copy the third placeholder from the edge feature token to the node tokens by using **SUM** in Lemma 3 and setting $\mathcal{S}_k = \{j|(e_k^{\text{pos2}} - e_j^{\text{pos2}})^2 < \delta\}$. Now the state for every target node token $u_i$ of the $i$-th edge is $[e^{\text{type1}}, e^{\text{type2}}, e^{\text{token}}, e^{\text{pos1}}, e^{\text{pos2}}, a_{u_i}, u_i, a_{u_i \leftarrow v_i}]$; (3) Use MLPs to put the node feature in the fourth placeholder if $e^{\text{type1}} == 3$; (4) Copy the fourth placeholder from the source node token to the edge feature token and target node token by using **SUM** in Lemma 3 and setting $\mathcal{S}_k = \{j|(e_k^{\text{pos2}} - e_j^{\text{pos2}})^2 < \delta\}$. Now the state for every target node token $u_i$ of the $i$-th edge is $[e^{\text{type1}}, e^{\text{type2}}, e^{\text{token}}, e^{\text{pos1}}, e^{\text{pos2}}, a_{u_i}, u_i, a_{u_i \leftarrow v_i}, a_{v_i}]$; (5) Put node id $u_i$ into the fifth placeholder if $e^{\text{type1}} == 2$; (6) Copy the fifth placeholder from the source node token to the edge feature token and target node token by using **SUM** in Lemma 3 and setting $\mathcal{S}_k = \{j|(e_k^{\text{pos2}} - e_j^{\text{pos2}})^2 < \delta\}$. Now the state for every target node token $u_i$ of the $i$-th edge is $[e^{\text{type1}}, e^{\text{type2}}, e^{\text{token}}, e^{\text{pos1}}, e^{\text{pos2}}, a_{u_i}, u_i, a_{u_i \leftarrow v_i}, a_{v_i}, v_i]$

**Block 3 - Message Preparation:** The goal of the third block is to compute $m_{u_i \leftarrow v_i}$. (1) Use MLPs to compute $\text{MSG}_\ell$ and place the results in the sixth placeholder. Now the state for every target node token $u_i$ of the $i$-th edge is $[e^{\text{type1}}, e^{\text{type2}}, e^{\text{token}}, e^{\text{pos1}}, e^{\text{pos2}}, a_{u_i}, a_{u_i \leftarrow v_i}, a_{v_i}, \text{MSG}_\ell(a_{u_i}, u_i, a_{u_i \leftarrow v_i}, a_{v_i}, v_i)]$; (2) Use MLPs to clean up the remaining placeholders. Now the state for every node token $u_i$ is $[e^{\text{type1}}, e^{\text{type2}}, e^{\text{token}}, e^{\text{pos1}}, e^{\text{pos1}}, \text{MSG}_\ell(a_{u_i}, u_i, a_{u_i \leftarrow v_i}, a_{v_i}, v_i)]$.

**Block 4 - Message Passing:** The goal of the fourth block is to compute $x_i^{(1)}$. (1) Use two attention heads to perform the sum operation in aggregation. This is achieved by using **SUM** in Lemma 3 for the first placeholder and setting $\mathcal{S}_k = \{j|(e_k^{\text{token}} - e_j^{\text{token}})^2 + (e_k^{\text{type2}} - e_j^{\text{type2}})^2 < \delta\}$. Now the state for every node token $u_i$ is $[e^{\text{type1}}, e^{\text{type2}}, e^{\text{token}}, e^{\text{pos1}}, e^{\text{pos2}}, \sum_{v_j \in \mathcal{N}_i^*} \text{MSG}_\ell(a_{u_i}, u_i, a_{u_i \leftarrow v_i}, a_{v_i}, v_i)]$;

(2) Use MLPs to compute $\text{UP}_\ell$ and obtain $x_i^{(1)}$

After four blocks, the final state for every node token $u_i$ is given by $[e^{\text{type1}}, e^{\text{type2}}, e^{\text{token}}, e^{\text{pos1}}, e^{\text{pos2}}, x_i^{(1)}]$. $x_i^{(l)}$ can be obtained by repeating the above four blocks $k$ times. □

**Lemma 3.** *Feng et al. (2024) Let $n \in \mathbb{N}$ be an integer and $\boldsymbol{x}_1, \cdots, \boldsymbol{x}_n$ be a sequence of vectors where $\boldsymbol{x}_i = (\tilde{\boldsymbol{x}}_i, r_i, 1) \in [-M, M]^{d+2}$ where $M$ is a large constant. Let $\boldsymbol{K}, \boldsymbol{Q}, \boldsymbol{V} \in \mathbb{R}^{d' \times (d+2)}$ be any matrices with $\|\boldsymbol{V}\|_\infty \leq 1$ and let $0 < \rho, \delta < M$ be any real numbers. Denote $\boldsymbol{q}_i = \boldsymbol{Q}\boldsymbol{x}_i, \boldsymbol{k}_j = \boldsymbol{K}\boldsymbol{x}_i, \boldsymbol{v}_j = \boldsymbol{V}\boldsymbol{x}_j$. Define a matching set $\mathcal{S} = \{j||\boldsymbol{q}_i^T \boldsymbol{k}_j| \leq \rho\}$. Define two following operations*

- **COPY**: *The output is a sequence of vectors $\boldsymbol{u}_1, \cdots, \boldsymbol{u}_n$ with $\boldsymbol{u}_i = \boldsymbol{v}_{pos(i)}$, where $pos(i) = \arg\max_{j \in \mathcal{S}_i} r_j$.*

- **MEAN, MAX, SUM**: *The output is a sequence of vectors $\boldsymbol{u}_1, \cdots, \boldsymbol{u}_n$, where $\boldsymbol{u}_i = \square_{j \in \mathcal{S}_i} \boldsymbol{v}_j$ and $\square$ is min or max or sum or mean.*

*Specifically, for any sequence of vectors $\boldsymbol{x}_1, \boldsymbol{x}_2, \cdots, \boldsymbol{x}_n$, denote the corresponding output of the attention layer as $\boldsymbol{o}_1, \boldsymbol{o}_2, \cdots, \boldsymbol{o}_n$. Then, we have $\|\boldsymbol{u}_i - \boldsymbol{o}_i\|_\infty \leq \epsilon$ for all $i \in [n]$ and $\mathcal{S} \neq \emptyset$.*

# B DATASET

## B.1 PRIMITIVE GRAPH PATTERNS

We select primitive graph patterns with varying node counts and edge numbers. First, 3-node patterns are the simplest structures in graphs, so we begin with patterns like the triangle, V-structure (V-S), feedforward loop (FFL), and feedback loop (FBL). In directed graphs, the V-S has two edges, while both the FFL and FBL have three. However, the FFL and FBL differ in the direction of one edge. For 4-node patterns, we select the tailed-triangle (T-triangle), square, and diamond for undirected graphs, and the directed-diamond for directed graphs. Both the T-triangle and square have four edges but differ in their connectivity, while the diamond includes five edges. Finally, we introduce the house pattern, a 5-node structure combining a triangle and a square. The summarization of patterns shown in Table 11.

Table 11: The selected primitive graph patterns (5 undirected and 4 directed patterns)

| Undirected Pattern | | | Directed Pattern | | |
|---|---|---|---|---|---|
| Name | Terminology-based Description | Structure | Name | Terminology-based Description | Structure |
| Triangle | A motif consisting of three nodes where each node is connected to the other two, forming a triangle | | V-structure (V-S) | two nodes have directed edges pointing toward a common target node | |
| Tailed-triangle (T-triangle) | A triangle with an additional node connected to one of the vertices of the triangle | | Feedforward loop (FFL) | A 3-node directed motif in which one source node influences a target node through two distinct pathways | |
| Square | A 4-node cycle where each node is connected to exactly two other nodes | | Feedback loop (FBL) | A 3-node directed cycle where the nodes form a loop | |
| Diamond | A 4-node motif with five edges | | Directed-diamond (D-diamond) | A 4-node motif in a directed graph where one node has directed edges to two intermediate nodes, and both of those intermediate nodes have directed edges to a common target node. | |
| House | A motif resembling the shape of a house with 5 nodes and 6 edges. The vertices and edges are arranged such that there is a triangular "roof" on top of a square or rectangular "base." | | | | |

## B.2 SYNTHETIC DATASET DETAILS

We generate different datasets for various tasks. In the pattern detection task, we randomly generate graphs on small (5-15 nodes), medium (15-25 nodes), and large (25-35 nodes) scales. Besides, to keep the numbers of different patterns reasonable, we ensure the average density of graphs is 0.5 for the undirected graph and 0.25 for the directed graph. We have 1893 undirected graphs and 1313 directed graphs in total. In the modification task, we randomly generate the graph with constraints. For example, we assume that LLMs need to add at most two nodes to achieve the goal, as in the Square $\rightarrow$ House sets. This means the original graphs contain at least one square pattern but lack a house pattern, requiring LLMs to modify the graph to create the house pattern. Similarly, Square $\rightarrow$

Diamond needs to add one edge while Diamond → Square needs to minus one edge. FFL → FBL needs to change one edge direction. In the frequent subgraph task, we restrict the generated graphs to include at least one given pattern, specifically a triangle. In the discriminative pattern learning task, we employ the BA-2motif dataset (Luo et al., 2020). We split it into 900 graphs to evaluate the discriminative pattern learning of LLMs and 100 for classification testing. For the evaluation, we will randomly select 250 instances from the small-scale dataset, 50 instances from the medium-scale dataset, and 50 instances from the large-scale dataset to evaluate the metrics due to the cost of APIs.

Table 12: Details of synthetic datasets

| Task | Dataset type | | difficulty | Num | AVG. node | AVG. edge | AVG. density |
|---|---|---|---|---|---|---|---|
| Pattern detection | Undirected graph | Evaluation | Small | 250 | 9.50 | 22.80 | 0.52 |
| | | | Medium | 250 | 19.50 | 96.20 | 0.52 |
| | | | Large | 250 | 29.50 | 247.96 | 0.58 |
| | Directed graph | Evaluation | Small | 250 | 9.50 | 23.44 | 0.26 |
| | | | Medium | 250 | 19.50 | 96.98 | 0.26 |
| | | | Large | 250 | 29.50 | 223.58 | 0.26 |
| Modification | Undirected graph | Square → House | Small | 166 | 9.71 | 25.07 | 0.56 |
| | | | Medium | 347 | 14.67 | 22.09 | 0.33 |
| | | | Large | 476 | 18.54 | 24.89 | 0.26 |
| | | Square → Diamond | Small | 144 | 9.91 | 10.96 | 0.27 |
| | | | Medium | 332 | 15.32 | 17.95 | 0.19 |
| | | | Large | 484 | 19.54 | 23.45 | 0.16 |
| | | Diamond → Square | Small | 111 | 8.95 | 10.98 | 0.34 |
| | | | Medium | 180 | 12.59 | 13.92 | 0.26 |
| | | | Large | 205 | 14.52 | 16.03 | 0.24 |
| | Directed graph | FFL → FBL | Small | 227 | 9.63 | 14.64 | 0.18 |
| | | | Medium | 396 | 13.69 | 19.08 | 0.13 |
| | | | Large | 493 | 16.60 | 23.48 | 0.12 |
| Frequent subgraph | Undirected graph | Triangle | Small | 231 | 9.87 | 17.61 | 0.39 |
| | | | Medium | 248 | 19.46 | 56.04 | 0.31 |
| | | | Large | 247 | 29.46 | 149.27 | 0.35 |
| | | Square | Small | 217 | 10.14 | 19.35 | 0.40 |
| | | | Medium | 249 | 19.49 | 56.32 | 0.31 |
| | | | Large | 249 | 29.49 | 152.85 | 0.35 |
| | | Diamond | Small | 214 | 10.19 | 20.12 | 0.42 |
| | | | Medium | 244 | 19.44 | 63.30 | 0.35 |
| | | | Large | 246 | 29.45 | 168.59 | 0.39 |
| | | House | Small | 205 | 10.37 | 20.50 | 0.41 |
| | | | Medium | 250 | 19.50 | 60.47 | 0.33 |
| | | | Large | 247 | 29.46 | 156.12 | 0.37 |
| | Directed graph | FFL | Small | 238 | 9.71 | 17.93 | 0.20 |
| | | | Medium | 248 | 19.48 | 59.90 | 0.17 |
| | | | Large | 250 | 29.50 | 154.67 | 0.18 |
| | | FBL | Small | 208 | 10.20 | 20.79 | 0.21 |
| | | | Medium | 244 | 19.41 | 64.15 | 0.18 |
| | | | Large | 248 | 29.48 | 156.56 | 0.18 |
| | | D-Diamond | Small | 187 | 10.60 | 22.33 | 0.21 |
| | | | Medium | 248 | 19.48 | 62.01 | 0.17 |
| | | | Large | 247 | 29.47 | 150.57 | 0.18 |
| Discriminative pattern learning | Discrimination | - | - | 900 | 25 | 25.5 | 0.09 |
| | Classification | - | - | 100 | 25 | 25.5 | 0.09 |

## B.3 REAL-WORLD DATASET DETAILS

To assess the effectiveness of our approach in practical scenarios, we utilize various classical real-world datasets with known ground-truth labels. These datasets encompass six molecule datasets: MUTAG, ogbg-molhiv, BBBP, Benzenes, Alkane-Carbonyl, and Fluoride-Carbonyl; one bioinformatics datasets: ENZYMES; one computer vision dataset: Fingerprint; and two social network datasets: IMDB-BINARY and IMDB-MULTI. Detailed information regarding datasets is delineated in Table 13.

**MUTAG** (Debnath et al., 1991). The MUTAG dataset is a collection of nitroaromatic compounds and its goal is to predict their mutagenicity on Salmonella typhimurium. In our evaluation, we use

the PyGeometric [1] version of the dataset, which comprises 188 molecular graphs, for conducting the binary classification task.

**OGBG-HIV** (Hu et al., 2020; Wu et al., 2018). This dataset encompasses 41,127 graphs, with each graph representing a molecule where nodes denote atoms and edges represent chemical bonds. The primary objective is to predict whether molecules inhibit HIV, constituting a binary classification task. For pattern discrimination, we sample 200 molecular graphs fairly, comprising 100 positive instances (i.e., inhibit HIV) and 100 negative instances. Subsequently, we select another 40 test graphs for pattern-based classification purposes.

**OGBG-BBBP** (Hu et al., 2020). The Blood–brain barrier penetration (BBBP) dataset originates from a study (Martins et al., 2012) focusing on modeling and predicting barrier permeability. As a membrane separating circulating blood and brain extracellular fluid, the blood–brain barrier blocks most drugs, hormones and neurotransmitters. This dataset includes binary labels for 2,050 compounds on their permeability properties. In alignment with our experimental conditions, we randomly select 500 compounds for pattern discrimination and 50 compounds for pattern-based binary classification, ensuring an equitable distribution of positive and negative samples.

**IMDB-BINARY** (Yanardag & Vishwanathan, 2015). It is a movie collaboration dataset that consists of the ego networks of 1,000 actors/actresses who have shared roles in movies listed on IMDB, originating from the Action and Romance genres. In each graph, the nodes symbolize actors/actresses, with edges connecting them if they have appeared together in a movie. For our experiments, we partition 80% of the 1,000 ego networks for pattern discrimination purposes, reserving the remaining networks for a binary classification task, i.e., aiming to predict whether a movie graph is an action or romance network.

**IMDB-MULTI** (Yanardag & Vishwanathan, 2015). IMDB-MULTI is a multi-class extension of IMDB-BINARY, comprising a balanced collection of ego-networks (1,500 graphs) sourced from Comedy, Romance, and Sci-Fi genres. In our study, a subset of 100 graphs from each genre is designated for extracting notable patterns, while a total of 60 graphs are reserved as test samples for a multi-classification objective. This task involves predicting whether a movie graph corresponds to a Comedy, Romance, or Sci-Fi network.

**Fingerprint** (Morris et al., 2020). The Fingerprint dataset is a multi-classification dataset obtained from fingerprint images, where 2,149 fingerprints are transformed into graphs through image filtering and region extraction processes to isolate relevant areas. In our research, we engage in a 3-class classification endeavor utilizing this dataset. We extract patterns by sampling 100 fingerprint graphs from each of the three distinct classes and reserve 20 graphs from each class for the classification task.

**ENZYMES** (Borgwardt et al., 2005). The ENZYMES dataset comprises 600 protein tertiary structures sourced from the BRENDA enzyme database, featuring six distinct enzymes. In our study, we focus on three out of the six enzymes to perform a 3-class classification task using LLMs. Each class is represented by 80 graphs utilized for pattern discrimination, while the remaining graphs are earmarked for evaluation purposes.

**Benzene** (Sanchez-Lengeling et al., 2020). Benzene consists 12,000 molecular graphs from the ZINC15 (Sterling & Irwin, 2015) database, which can be classified into two classes. The main goal is to determine if a Benzene ring is existed in each molecule. In our settings, 200 graphs (1:1 for positive and negative) are sampled uniformly for LLM-based pattern detection, and a hexagon made up of carbon atoms is used as the target pattern in LLM prompts.

**Alkane-Carbonyl** (Sanchez-Lengeling et al., 2020). The Alkane-Carbonyl dataset comprises 4,326 molecule graphs categorized into two distinct classes based on the presence of specific functional groups. Positive samples correspond to molecules containing both alkane and carbonyl (C=O) functional groups. To analyze patterns, we select 100 molecules from each class, aiming to identify whether a molecule includes both alkane and carbonyl functional groups.

**Fluoride-Carbonyl** (Sanchez-Lengeling et al., 2020). The Fluoride-Carbonyl dataset has 8,671 molecular graphs and its ground-truth explanation is based on the particular combination of fluoride (F-) atoms and carbonyl (C=O) functional groups present in each molecule. For the pattern detection

---

[1] https://www.pyg.org/

task, we select 100 molecules from each class, aiming to identify whether a molecule includes both fluoride atoms and carbonyl functional groups.

Table 13: Statistics of real-world datasets. Alkane* and Fluoride* represent the Alkane-Carbonyl and Fluoride-Carbonyl datasets, respectively.

| Task | Domain | Name | Progress | Num | AVG. node | AVG. edge | AVG. density |
|---|---|---|---|---|---|---|---|
| Bi-Class. | Molecule | MUTAG | discrimination | 150 | 15.67 | 16.79 | 0.0725 |
| | | | classification | 38 | 15.68 | 16.76 | 0.0723 |
| | | | overall | 188 | 15.67 | 16.78 | 0.0725 |
| | | OGBG-HIV | discrimination | 200 | 31.77 | 34.82 | 0.0445 |
| | | | classification | 40 | 30.50 | 33.45 | 0.0467 |
| | | | overall | 240 | 31.65 | 34.69 | 0.0447 |
| | | OGBG-BBBP | discrimination | 500 | 21.99 | 23.40 | 0.0595 |
| | | | classification | 50 | 28.24 | 30.64 | 0.0429 |
| | | | overall | 550 | 22.56 | 24.06 | 0.0580 |
| | Social Network | IMDB-BINARY | discrimination | 500 | 19.56 | 96.18 | 0.2457 |
| | | | classification | 50 | 19.73 | 98.43 | 0.2466 |
| | | | overall | 550 | 19.57 | 96.39 | 0.2458 |
| Pattern Detection | Chemicals | Benzene | overall | 200 | 20.49 | 21.75 | 0.0547 |
| | | Alkane* | overall | 200 | 41.54 | 42.72 | 0.0259 |
| | | Fluoride* | overall | 200 | 21.46 | 22.65 | 0.0508 |
| Multi-Class. | Bioinformatics | ENZYMES | discrimination | 240 | 33.40 | 63.91 | 0.0731 |
| | | | classification | 60 | 31.93 | 62.78 | 0.0774 |
| | | | overall | 300 | 33.15 | 63.72 | 0.0738 |
| | Computer Vision | Fingerprint | discrimination | 300 | 2.92 | 2.13 | 0.2428 |
| | | | classification | 60 | 2.93 | 2.20 | 0.2560 |
| | | | overall | 360 | 2.92 | 2.14 | 0.2450 |
| | Social Network | IMDB-MULTI | discrimination | 300 | 12.95 | 67.21 | 0.3503 |
| | | | classification | 60 | 12.62 | 52.90 | 0.3279 |
| | | | overall | 360 | 12.89 | 64.83 | 0.3466 |

## B.4 PROMPT

We collect the prompt for different tasks in Table 16. Further, we provide the molecule description prompts in Table 14. To enhance LLM understanding, we use "both" to combine these two descriptions. The detailed prompt is: "In the context of molecular biology, you have been provided with a pattern motif to compare against a test molecule graph. The pattern is a Terminology-based description, which also can be represented as Topology-based description. ... Test-Molecular ... Now, please determine whether the pattern motif exists in the molecule graph by selecting either "The pattern does exist" or "The pattern does not exist"."

Table 14: The descriptions of functional groups

| Function group | Terminology-based | Topology-based |
|---|---|---|
| Benzene (Cn) | benzene ring | (Node 0 Atom C, Node 1 Atom C), (Node 1 Atom C, Node 2 Atom C), (Node 2 Atom C, Node 3 Atom C), (Node 3 Atom C, Node 4 Atom C), (Node 4 Atom C, Node 5 Atom C) |
| Alkane (C2nH2n+2) | Alkane Carbonyl which contains an unbranched alkane and a carbonyl functional group | (Node 0 Atom C, Node 1 Atom H), (Node 0 Atom C, Node 2 Atom H), (Node 0 Atom C, Node 3 Atom H), (Node 0 Atom C, Node 4 Atom H) |
| Fluoride (COF2) | Fluoride Carbonyl which contains a fluoride and a carbonyl functional group | (Node 0 Atom C, Node 1 Atom O), (Node 0 Atom C, Node 2 Atom F), (Node 0 Atom C, Node 3 Atom F) |

Further, for the molecular graphs, we employ two different methods for graph description: adjacency list (A.L.) and edge list (E.L.). The conversion process involves three steps: 1. Using the function Chem.MolFromSmiles from the Chem library in Python, we extract the atoms and adjacency matrix

of a given molecule from its SMILES representation; 2. The atom and adjacency matrix information is used to construct an undirected graph with the Python tool networkx.Graph; 3. The graph is then described using node and edge information in either adjacency list (A.L.) or edge list (E.L.) format.

Taking a molecular graph with the SMILES of "C(C(=O)[O-])NC(=[NH2+])N" as an example, the molecular graph can be converted to textual format as expressed in the following Table 15:

Table 15: A.L. and E.L. on the molecular graph

| A.L | E.L. |
|---|---|
| G describes an undirected graph among 0, 1, 2, 3, 4, 5, 6, and 7. In this graph:\nNode 0 (atom: C) is connected to nodes 1 (atom: C), 4 (atom: N).\nNode 1 (atom: C) is connected to nodes 0 (atom: C), 2 (atom: O), 3 (atom: O).\nNode 2 (atom: O) is connected to nodes 1 (atom: C).\nNode 3 (atom: O) is connected to nodes 1 (atom: C).\nNode 4 (atom: N) is connected to nodes 0 (atom: C), 5 (atom: C).\nNode 5 (atom: C) is connected to nodes 4 (atom: N), 6 (atom: N), 7 (atom: N).\nNode 6 (atom: N) is connected to nodes 5 (atom: C).\nNode 7 (atom: N) is connected to nodes 5 (atom: C). | G describes an undirected graph among node 0, 1, 2, 3, 4, 5, 6, and 7.\nNode 0 (atom: C) is connected to Node 1 (atom: C).\nNode 0 (atom: C) is connected to Node 4 (atom: N).\nNode 1 (atom: C) is connected to Node 2 (atom: O).\nNode 1 (atom: C) is connected to Node 3 (atom: O).\nNode 4 (atom: N) is connected to Node 5 (atom: C).\nNode 5 (atom: C) is connected to Node 6 (atom: N).\nNode 5 (atom: C) is connected to Node 7 (atom: N). |

Table 16: Summarization of prompts

| Task | Prompt |
|---|---|
| Pattern translation | Generate a graph that includes only one {terminology-based pattern description}, the node number is 20. Each node at least has one edge. |
| Pattern detection | Identify the occurrence patterns of the given motif in the graph. The given pattern is {terminology-based (topology-based) pattern description }. The graph is... |
| Pattern modification | Modify the graph to include the given pattern {terminology-based (topology-based) pattern description }. The pattern is ... The graph is... |
| Pattern isomorphic mapping | Given a pair of isomorphic graphs, determine the node correspondence between the two graphs. The first graph is... The second graph is... |
| K-core detection | Determine the 3-core subgraphs in the graph. The graph is ... |
| Frequent subgraph extraction | Consider the following graphs and summarize the common patterns in them. No. 1. The graph is ... No. 2. The graph is... |
| Discriminative pattern learning | You are provided two sets of graphs. The first set is: No. 1 The graph is... No. t. The graph is... The second set is: No. 1. The graph is... No. 2. The graph is...What are the differences between the two sets? Show the special pattern in Set1 (Set2) |
| Classification | You are an expert at classifying different types of graphs based on whether they contain specific patterns. The first type of graph includes patterns such as: No.1 the pattern is... No.2 The pattern is...The second type includes patterns like: No.1 the pattern is...No.2 The pattern is... Now, please identify which type the given graph is most likely to belong to. The graph is... |

## C  REAL-WORLD APPLICATIONS

### C.1  EXPERIMENTAL SETTINGS

The evaluation setup involves comparing GPT-4, GPT-4o, and O1mini models in real-world applications, following a pipeline similar to that proposed for synthetic datasets to prompt Large Language Models (LLMs) to comprehend graph patterns. Details are depicted in Table 17. In classification tasks, we sample five graphs from each class for comparison within an turns, requesting LLMs to discriminate significant graph patterns within a designated target set. All classes within the dataset are utilized as the target set for the pattern discrimination process, and the discrimination process is performed twice for complementary insights. After filtering out discriminate patterns, we prompt LLMs to classify test graphs individually. An effective LLM-based graph reasoning technique is characterized by the ability to discriminate more patterns and achieve satisfactory performance in classifications, with accuracy serving as the key metric for quantitative assessment. Notably, these settings here apply to both binary and multi-label classification tasks.

Table 17: Hyper parameters for real-world tasks.

| Param | Binary Classification | | | | Multi-label Classification | | |
|---|---|---|---|---|---|---|---|
| | MUTAG | OGBG-HIV | OGBG-BBBP | IMDB-BINARY | ENZYMES | Fingerprint | IMDB-MULTI |
| # Sampling | 10 | 5 | 5 | 5 | 5 | 5 | 5 |
| # Turns | 16 | 40 | 100 | 100 | 32 | 40 | 40 |

## D  THE FULL RESULTS FOR PATTERN DETECTION

The whole results of terminology-based and topology-based pattern detection are shown in Table 18 and Table 19, respectively.

Table 18: The F1 score for terminology-based graph detection

| Scale | Models | Undirected patterns | | | | | | | | | | Dndirected patterns | | | | | | | |
|---|---|---|---|---|---|---|---|---|---|---|---|---|---|---|---|---|---|---|---|
| | | Triangle | | T-triangle | | Square | | Diamond | | House | | V-S | | FFL | | FBL | | D-Diamond | |
| | | A.L. | E.L | A.L. | E.L | A.L. | E.L | A.L. | E.L | A.L. | E.L | A.L. | E.L | A.L. | E.L | A.L. | E.L | A.L. | E.L |
| Small | GPT-4 | .632 | .581 | .151 | .107 | .069 | .026 | .113 | .113 | .006 | .003 | .352 | .380 | .389 | .406 | .191 | .279 | .448 | .396 |
| | GPT-4o | .748 | .702 | .210 | .250 | .149 | .132 | .317 | .309 | .008 | .053 | .477 | .490 | .478 | .450 | .410 | .357 | .411 | .360 |
| | Mixtral | .694 | .622 | .224 | .211 | .128 | .102 | .232 | .181 | .121 | .118 | .117 | .110 | .191 | .241 | .110 | .145 | .238 | .140 |
| | Llama | .681 | .693 | .189 | .193 | .118 | .106 | .185 | .195 | .042 | .021 | .486 | .527 | .358 | .388 | .373 | .349 | .425 | .367 |
| | Gemini | .712 | .725 | .207 | .274 | .150 | .176 | .230 | .262 | .104 | .225 | .287 | .281 | .263 | .267 | .194 | .193 | .149 | .111 |
| | Claude | .782 | .740 | .217 | .229 | .178 | .149 | .273 | .259 | .210 | .171 | .385 | .365 | .300 | .277 | .262 | .265 | .229 | .201 |
| | O1-mini | .828 | .832 | .593 | .578 | .335 | .316 | .663 | .684 | .054 | .066 | .584 | .600 | .605 | .602 | .567 | .584 | .504 | .488 |
| Medium | GPT-4 | .356 | .345 | .061 | .031 | .010 | .004 | .083 | .057 | .017 | .000 | .101 | .130 | .085 | .117 | .054 | .108 | .035 | .023 |
| | GPT-4o | .671 | .563 | .108 | .120 | .102 | .114 | .242 | .215 | .024 | .034 | .247 | .300 | .246 | .191 | .141 | .163 | .097 | .081 |
| | Mixtral | .474 | .478 | .104 | .118 | .006 | .032 | .132 | .125 | .000 | .007 | .040 | .053 | .049 | .149 | .060 | .081 | .007 | .003 |
| | Llama | .618 | .584 | .059 | .043 | .082 | .070 | .140 | .108 | .020 | .000 | .207 | .210 | .153 | .167 | .119 | .134 | .051 | .043 |
| | Gemini | .678 | .658 | .197 | .218 | .091 | .107 | .247 | .299 | .056 | .000 | .125 | .133 | .188 | .167 | .148 | .111 | .012 | .018 |
| | Claude | .725 | .673 | .144 | .157 | .130 | .161 | .205 | .210 | .038 | .016 | .130 | .141 | .181 | .173 | .104 | .052 | .006 | .009 |
| | O1-mini | .848 | .777 | .409 | .453 | .335 | .274 | .535 | .567 | .009 | .000 | .527 | .516 | .611 | .637 | .603 | .594 | .318 | .327 |
| Large | GPT-4 | .057 | .086 | .003 | .005 | .003 | .000 | .001 | .002 | .000 | .000 | .067 | .061 | .066 | .064 | .031 | .059 | .029 | .016 |
| | GPT-4o | .404 | .317 | .016 | .045 | .011 | .012 | .063 | .041 | .000 | .000 | .156 | .226 | .151 | .140 | .111 | .066 | .091 | .058 |
| | Mixtral | .319 | .233 | .016 | .014 | .005 | .006 | .064 | .056 | .000 | .022 | .016 | .030 | .046 | .155 | .034 | .055 | .008 | .012 |
| | Llama | .361 | .384 | .012 | .013 | .018 | .011 | .025 | .030 | .020 | .020 | .111 | .123 | .133 | .167 | .054 | .073 | .022 | .014 |
| | Gemini | .600 | .496 | .035 | .014 | .043 | .011 | .099 | .063 | .038 | .047 | .101 | .103 | .174 | .166 | .110 | .137 | .001 | .000 |
| | Claude | .320 | .278 | .013 | .020 | .020 | .023 | .051 | .044 | .020 | .020 | .074 | .054 | .147 | .138 | .066 | .078 | .004 | .007 |
| | O1-mini | .636 | .428 | .065 | .072 | .039 | .026 | .103 | .025 | .000 | .000 | .533 | .560 | .635 | .637 | .568 | .592 | .363 | .367 |

The figures are shown as terminology-based pattern detection for all scales in Figure 7 and topology-based pattern detection for all scales in Figure 8

Table 19: The F1 score for topology-based graph detection

| Scale | Models | Undirected patterns | | | | | | | | | | Dndirected patterns | | | | | | | |
|---|---|---|---|---|---|---|---|---|---|---|---|---|---|---|---|---|---|---|---|
| | | Triangle | | T-triangle | | Square | | Diamond | | House | | V-S | | FFL | | FBL | | D-Diamond | |
| | | A.L. | E.L | A.L. | E.L | A.L. | E.L | A.L. | E.L | A.L. | E.L | A.L. | E.L | A.L. | E.L | A.L. | E.L | A.L. | E.L |
| Small | GPT-4 | .653 | .584 | .051 | .056 | .041 | .039 | .162 | .110 | .002 | .010 | .415 | .475 | .346 | .429 | .423 | .422 | .477 | .397 |
| | GPT-4o | .717 | .706 | .194 | .143 | .118 | .111 | .233 | .227 | .005 | .015 | .537 | .489 | .296 | .383 | .407 | .287 | .285 | .269 |
| | Mixtral | .520 | .550 | .108 | .134 | .132 | .107 | .170 | .157 | .105 | .070 | .191 | .227 | .287 | .364 | .276 | .352 | .255 | .257 |
| | Llama | .545 | .678 | .122 | .145 | .076 | .074 | .112 | .157 | .005 | .097 | .476 | .488 | .266 | .361 | .382 | .386 | .388 | .283 |
| | Gemini | .651 | .696 | .166 | .192 | .133 | .175 | .177 | .171 | .122 | .192 | .304 | .171 | .266 | .230 | .194 | .205 | .203 | .126 |
| | Claude | .730 | .713 | .186 | .269 | .122 | .160 | .263 | .250 | .241 | .249 | .370 | .329 | .302 | .393 | .322 | .257 | .206 | .128 |
| | O1-mini | .832 | .821 | .617 | .499 | .365 | .364 | .633 | .574 | .107 | .249 | .588 | .599 | .678 | .670 | .572 | .566 | .492 | .477 |
| Medium | GPT-4 | .273 | .329 | .032 | .047 | .012 | .045 | .064 | .068 | .000 | .000 | .177 | .256 | .072 | .210 | .129 | .099 | .051 | .041 |
| | GPT-4o | .657 | .575 | .183 | .089 | .086 | .066 | .143 | .205 | .000 | .000 | .298 | .289 | .072 | .258 | .146 | .154 | .046 | .053 |
| | Mixtral | .414 | .430 | .057 | .074 | .039 | .051 | .051 | .139 | .000 | .000 | .040 | .127 | .188 | .180 | .089 | .095 | .021 | .029 |
| | Llama | .299 | .290 | .029 | .032 | .050 | .061 | .081 | .094 | .000 | .020 | .193 | .213 | .094 | .134 | .059 | .124 | .061 | .035 |
| | Gemini | .484 | .632 | .037 | .075 | .123 | .088 | .033 | .124 | .000 | .030 | .140 | .037 | .170 | .162 | .109 | .098 | .062 | .000 |
| | Claude | .671 | .651 | .100 | .159 | .097 | .127 | .247 | .253 | .040 | .040 | .310 | .166 | .204 | .233 | .084 | .092 | .019 | .035 |
| | O1-mini | .833 | .749 | .488 | .367 | .449 | .417 | .494 | .557 | .000 | .024 | .471 | .496 | .690 | .625 | .567 | .591 | .298 | .301 |
| Large | GPT-4 | .067 | .113 | .004 | .002 | .000 | .002 | .003 | .005 | .000 | .000 | .194 | .230 | .068 | .133 | .108 | .085 | .007 | .010 |
| | GPT-4o | .327 | .257 | .018 | .015 | .003 | .008 | .027 | .055 | .000 | .000 | .240 | .158 | .065 | .180 | .119 | .096 | .055 | .014 |
| | Mixtral | .177 | .262 | .006 | .010 | .006 | .009 | .046 | .025 | .000 | .000 | .043 | .086 | .144 | .184 | .021 | .037 | .006 | .013 |
| | Llama | .085 | .068 | .012 | .004 | .012 | .001 | .047 | .008 | .020 | .020 | .042 | .094 | .106 | .070 | .020 | .134 | .003 | .013 |
| | Gemini | .186 | .535 | .001 | .000 | .003 | .061 | .002 | .005 | .000 | .027 | .064 | .025 | .122 | .152 | .098 | .074 | .016 | .000 |
| | Claude | .454 | .286 | .011 | .015 | .010 | .015 | .048 | .072 | .000 | .040 | .188 | .162 | .151 | .170 | .107 | .080 | .011 | .040 |
| | O1-mini | .511 | .514 | .045 | .049 | .044 | .050 | .134 | .114 | .000 | .000 | .518 | .488 | .608 | .625 | .591 | .595 | .340 | .343 |

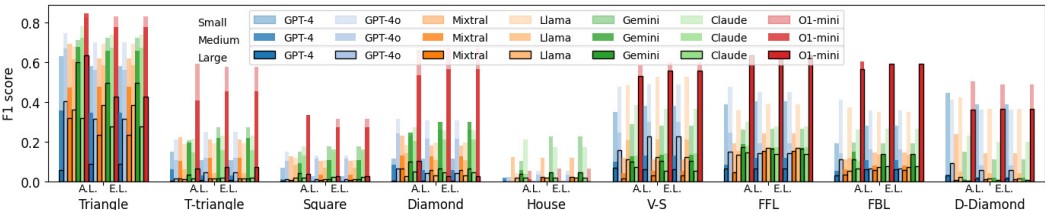

Figure 7: The F1 score of topology-based pattern detection (small and medium scale)

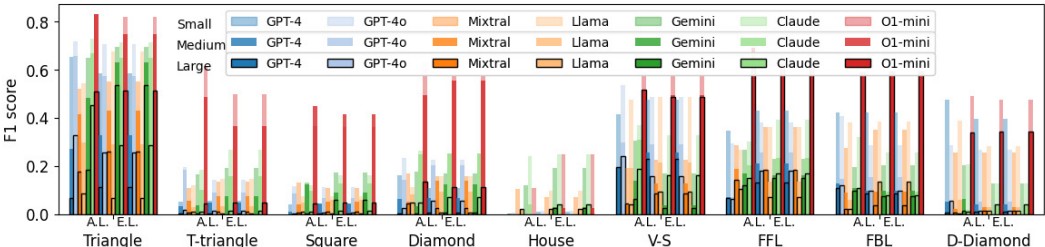

Figure 8: The F1 score of topology-based pattern detection (small and medium scale)

# E  PSEUDO-CODES

## E.1  FREQUENT SUBGRAPH EXTRACTION

The pseudo-code of the algorithm is shown in Algorithm 2.

## E.2  DISCRIMINATIVE PATTERN LEARNING

The pseudo-code of the algorithm is shown in Algorithm 3.

# F  BIRD-VIEW

To show the model's ability across various tasks, we provide a bird-view of the models.

For each LLM, we select the best performance from either edge list or adjacency list graph descriptions and then calculate the models' average scores across small, medium, and large-scale datasets.

---

**Algorithm 2** Frequent Subgraph Extraction

---

1: **Input:** A graph dataset $G = \{g_1, g_2, \ldots, g_n\}$, frequency threshold $f_{\text{thres}}$
2: **Output:** Frequent patterns and accuracy
3: **for** iteration $i = 1$ to 100 **do**
4:     Randomly select 10 graphs from $G$ to form a subset $G_p$
5:     Prompt LLMs to extract the set of frequent patterns $P$ based on $G_p$
6:     Initialize $\text{Acc}_i = 0$
7:     **for** each pattern $p \in P$ **do**
8:         **if** $p$ appears in more than $f_{\text{thres}}$ of graphs in $G_p$ **then**
9:             Increment $\text{Acc}_i$
10:     Compute $\text{Acc}_i = \frac{\text{Acc}_i}{\text{\# of patterns in } P}$
11: Compute overall accuracy $\text{Acc} = \frac{\sum \text{Acc}_i}{\text{\# of iterations}}$
12: **Return:** Extracted frequent patterns and accuracy

---

---

**Algorithm 3** Discriminative Pattern Extraction and Evaluation

---

1: **Input:** Two graph dataset $G^1 = \{g_1^1, g_2^1, \ldots, g_{n_1}^1\}$ with label $L_1$ and $G^2 = \{g_1^2, g_2^2, \ldots, g_{n_2}^2\}$ with label $L_2$
2: **Output:** Discriminative patterns and Metrics
3: **Step 1: Pattern Extraction**
4: **for** each iteration $i$ **do**
5:     Sample an equal number of graphs from $G^1$ and $G^2$ to form a balanced dataset $G_i$
6:     Prompt LLMs to identify discriminative patterns from $G_i$
7:     Add the extracted patterns into the set $P$
8: **Step 2: Pattern Filtering**
9: **for** each pattern $p \in P$ **do**
10:     Compute the occurrence of $p$ in $G^1$ and $G^2$
11:     **if** $\big(\text{occurrence}(p, G^1) \geq 90\%$ **and** $\text{occurrence}(p, G^2) < 10\%\big)$ **OR** $\big(\text{occurrence}(p, G^2) \geq 90\%$ **and** $\text{occurrence}(p, G^1) < 10\%\big)$ **then**
12:         Retain $p$ as a discriminative pattern
13: Obtain final discriminative pattern set $P_{\text{final}}$
14: **Step 3: D.P. Computation**
15: Compute the discriminative pattern ratio as:

$$\text{D.P.} = \frac{\text{\#Discriminative patterns in } P_{\text{final}}}{\text{\#Extracted patterns in } P}$$

16: **Step 4: Classification Accuracy Computation**
17: **for** each new graph $g$ in the test set $G_{\text{test}}$ **do**
18:     Prompt LLMs to predict the label of $g$ based on $P_{\text{final}}$
19: Compute the prediction accuracy $\text{Acc}$ as the proportion of correctly predicted labels
20: **Return:** $P_{\text{final}}$, $\text{Acc}$, and D.P.

---

Furthermore, we average the scores across different graph patterns. Finally, we rank the models for each task and provide an overall ranking.

Table 20: Model rank across various tasks

| | Terminology-based patterns | | | Topology-based patterns | | | Data-driven patterns | | | AVG. rank |
|---|---|---|---|---|---|---|---|---|---|---|
| | Pattern translation | Graph modification | pattern detection | Isomophic mapping | Graph modification | pattern detection | K-core | Frequent subgraph extraction | Discriminative pattern learning | |
| GPT-4 | 3 | 6 | 7 | 6 | 5 | 5 | 7 | 1 | 4 | 4.9 |
| GPT-4o | 2 | 3 | 2 | 2 | 2 | 3 | 1 | 4 | 1 | 2.2 |
| Mixtral | 7 | 4 | 6 | 4 | 3 | 7 | 5 | 1 | 7 | 4.9 |
| Llama | 5 | 2 | 5 | 3 | 4 | 6 | 4 | 5 | 5 | 4.3 |
| Gemini | 4 | 7 | 3 | 7 | 7 | 4 | 6 | 6 | 3 | 5.2 |
| Claude | 6 | 5 | 4 | 1 | 6 | 2 | 2 | 7 | 1 | 3.8 |
| O1-mini | 1 | 1 | 1 | 5 | 1 | 1 | 2 | 1 | 6 | 2.1 |

In Table 20, O1-mini achieves an average rank of 2.1, outperforming other models in most cases while still facing challenges in isomorphic mapping and discriminative pattern learning tasks. In-

terestingly, GPT-4o demonstrates balanced performance across all tasks. Overall, we recommend using O1-mini, GPT-4o, and Claude for solving graph pattern tasks.

# G   ALGORITHM SUMMARY

To ensure a comprehensive evaluation that encompasses both top-performing results and identified mistakes from under-performing models, we meticulously examine 10% of the samples across all LLMs utilized in diverse tasks. The descriptions of specific procedures and their respective occurrence rates in varied scenarios are detailed in Table 21 and Table 22.

# H   EXPERIMENTS ANALYSIS

## H.1   UNINTENDED MODIFICATION IN PATTERN TRANSLATION

LLMs generally follow the logic of first creating the target pattern and then adding extra edges to expand it into a larger graph, as shown in Appendix G. To illustrate this process, we analyze output from GPT-4 as an example of how LLMs perform graph translation in tailed-triangle generation:

Here's how the graph can be formulated: - **Tailed Triangle Edges**: (1, 2), (2, 3), (3, 1), (1, 4) - This forms the tailed triangle. - **Connecting Remaining Nodes**: To ensure each of the remaining nodes (5 to 20) has at least one edge and to avoid forming any additional tailed triangles, we can simply connect each new node to the next one in sequence. This will form a chain that can be connected back to one of the nodes in the tailed triangle to ensure the graph is undirected and connected. Here's the complete edge list: 1. (1, 2) 2. (2, 3) 3. (3, 1) 4. (1, 4) 5. (4, 5) 6. (5, 6) 7. (6, 7) 8. (7, 8) 9. (8, 9) 10. (9, 10) 11. (10, 11) 12. (11, 12) 13. (12, 13) 14. (13, 14) 15. (14, 15) 16. (15, 16) 17. (16, 17) 18. (17, 18) 19. (18, 19) 20. (19, 20) 21. (20, 3) - This last edge connects the chain back to the triangle, ensuring the graph is undirected and connected.

The connection (20, 3) is an unintended modification, as the graph is already connected without it. Another example is from Gemini while generating the VS pattern. The example is as follows:

Here's a directed graph with 20 nodes and a single V-structure, along with the edge list representation: **V-Structure:** * Node 1 –> Node 20 * Node 2 –> Node 20 **Remaining Graph Structure (ensuring each node has at least one edge):** We'll create a cycle involving the remaining nodes and then add a few extra edges for variety: * 3 –> 4 –> 5 –> 6 –> 7 –> 8 –> 9 –> 10 –> 11 –> 12 –> 13 –> 14 –> 15 –> 16 –> 17 –> 18 –> 19 –> 3 (This forms the cycle) * 1 –> 5 (Additional edge) * 10 –> 18 (Additional edge)

Nodes 1, 2, and 20 already form a VS structure, but Gemini creates a cycle and adds two additional edges to connect all the nodes in the graph. This results in the formation of another VS structure involving nodes (1, 4, 5).

## H.2   LLMs' BEHAVIORS IN THE MODIFICATION CASE

According to the algorithms summarized in Appendix G, we find that most responses follow a strategy where they first select a subset of nodes equal in number to the target pattern and then apply modifications to match the given pattern. Second, we calculated the average degree of the nodes selected by the LLMs and summarized this information in Table 23.

We find that the nodes selected by LLMs consistently have higher degrees than the average node degree of the graph, particularly in Medium and Large scales. This suggests that LLMs are more likely to select higher-degree nodes for editing.

## H.3   HALLUCINATION IN PATTERN DETECTION

The hallucination happened in graph understanding, which means adding or ignoring edges on the graphs. We analyze the hallucinations that occurred during terminology-based and topology-based pattern detections. After analyzing in Appendix G, we find the behavior of LLMs on pattern detections are: (1) LLMs often provide a solution without actually executing the algorithm. This leads to failures, such as Gemini and GPT-4 in terminology-based pattern detection and GPT-4o in topology-based pattern detection. (2) In terminology-based pattern detection tasks, LLMs are more flexible

Table 21: Overview of algorithms employed by LLMs for graph pattern comprehension tasks

| Method | Task | Item | Description |
|---|---|---|---|
| Terminology -based | Pattern Translation | A | Use external tools e.g. NetworkX |
| | | B | Directly give an answer |
| | | C | 1. Construct a structure based on the given patterns.; 2. Add random paths or trees among other nodes. |
| | Pattern Modification | A | 1. Select a set of nodes.2. Modify the subset to match the target pattern. |
| | | B | A special algorithm on house 1. Identify a triangle; 2. Modify a square based on the triangle. |
| | | C | A special algorithm on house 1. Identify a square; 2. Modify a triangle based on the square. |
| | | D | Assume the graph already meets the requirements and avoid making any modifications. |
| | Pattern Detection | A | Directly give an answer |
| | | B | Use external tools e.g. NetworkX |
| | | C | Draw a figure of graph and give answer |
| | | D | Traverse every node, and check whether this node and its neighbors can form the pattern. |
| | | E | Generate all possible node combinations and verify one by one. |
| | | F | Traverse all possible edge combinations, and verify if they form the pattern. |
| | | G | A special algorithm on house: Identify triangles as the roof first and check if the triangle has a square as its base. |
| | | H | A special algorithm on the house: Identify squares as the base first and check if the square has a triangle as its roof. |
| | | I | Only give the process but no answers |
| Topology -based | Isomorphic Mapping | A | 1. Count the degrees of nodes in both graphs. 2. Identify nodes with matching degrees and assign them as mapping pairs. |
| | | B | Directly map the edge connections. |
| | | C | Provide example code (networkx package) only, without displaying the results. |
| | | D | Using the VF2 algorithm. |
| | Pattern Modification | A | 1. Select a set of nodes. 2. Adjust the subset to match the target pattern. |
| | Pattern Detection | A | Directly give an answer |
| | | B | Traverse every node, and check whether this node and its neighbors can form the target pattern. |
| | | C | Generate all combinations with the specified number of nodes and select those that meet the pattern definition. |
| | | D | Traverse all edges, and determine if they form a pattern based on their common nodes. |
| | | E | Only give the process but no answers |
| | | F | Using external tools e.g. networkx |
| Data-driven | Dense Subgraph Mining (K-core) | A | Using external tools e.g. networkx |
| | | B | 1. Count the degrees. 2. Modify the graph. 3. Repeat the steps iteratively. |
| | | C | Assume every node meets the degree requirement and take no further action. |
| | | D | Collect nodes with a degree greater than 3. |
| | Frequency Subgraph Extraction | A | Directly give an answer |
| | | B | 1.List common graph patterns, such as triangles, stars, and loops; 2. Verify whether these patterns exist in the given graphs. |
| | | C | 1. Check the nodes by their IDs; 2. Examine the neighbors of each node to determine if their combinations appear in other graphs. |
| | | D | Check the patterns by node IDs. For example. 1.Nodes connected to others with smaller IDs. 2. Nodes connected to only one other node. |
| | | E | No solution |
| | Discriminative Pattern Learning | A | 1. Assume the patterns are common structures; 2. Identify which nodes exhibit these patterns; 3. Compare the differences in their connections |
| | | B | Check the edge connection patterns to determine if they are identical. |

Table 22: Percentage of occurrence for each algorithm during LLM-based graph pattern comprehension

| Method | Task | Item | Llama | Gemini | Mixstral | GPT-4 | GPT-4o | Claude | O1-mini |
|---|---|---|---|---|---|---|---|---|---|
| Terminology-based | Pattern Translation | A | 100.00% | 0.00% | 0.00% | 0.00% | 0.00% | 0.00% | 0.00% |
| | | B | 0.00% | 0.00% | 0.00% | 0.00% | 0.00% | 100.00% | 0.00% |
| | | C | 0.00% | 100.00% | 100.00% | 100.00% | 100.00% | 0.00% | 100.00% |
| | Pattern Modification | A | 100.00% | 85.00% | 80.00% | 75.00% | 95.00% | 80.00% | 100.00% |
| | | B | 0.00% | 10.00% | 20.00% | 0.00% | 5.00% | 20.00% | 0.00% |
| | | C | 0.00% | 0.00% | 0.00% | 25.00% | 0.00% | 0.00% | 0.00% |
| | | D | 0.00% | 15.00% | 0.00% | 0.00% | 0.00% | 0.00% | 0.00% |
| | Pattern Detection | A | 0.00% | 34.00% | 24.00% | 4.00% | 0.00% | 24.00% | 18.00% |
| | | B | 6.00% | 0.00% | 4.00% | 0.00% | 6.00% | 0.00% | 20.00% |
| | | C | 0.00% | 4.00% | 0.00% | 0.00% | 4.00% | 0.00% | 0.00% |
| | | D | 0.00% | 4.00% | 24.00% | 24.00% | 38.00% | 12.00% | 4.00% |
| | | E | 54.00% | 12.00% | 20.00% | 16.00% | 38.00% | 60.00% | 28.00% |
| | | F | 40.00% | 8.00% | 14.00% | 16.00% | 0.00% | 4.00% | 18.00% |
| | | G | 0.00% | 0.00% | 8.00% | 2.00% | 12.00% | 0.00% | 2.00% |
| | | H | 0.00% | 2.00% | 2.00% | 0.00% | 0.00% | 0.00% | 10.00% |
| | | I | 0.00% | 36.00% | 4.00% | 38.00% | 2.00% | 0.00% | 0.00% |
| Topology-based | Isomorphic Mapping | A | 0.00% | 20.00% | 60.00% | 70.00% | 50.00% | 0.00% | 100.00% |
| | | B | 100.00% | 10.00% | 20.00% | 0.00% | 50.00% | 100.00% | 0.00% |
| | | C | 0.00% | 70.00% | 0.00% | 30.00% | 0.00% | 0.00% | 0.00% |
| | | D | 0.00% | 0.00% | 20.00% | 0.00% | 0.00% | 0.00% | 0.00% |
| | Pattern Modification | A | 100.00% | 100.00% | 100.00% | 100.00% | 100.00% | 100.00% | 100.00% |
| | Pattern Detection | A | 0.00% | 43.33% | 10.00% | 0.00% | 0.00% | 53.33% | 10.00% |
| | | B | 0.00% | 36.67% | 16.67% | 33.33% | 36.67% | 36.67% | 36.67% |
| | | C | 66.67% | 16.67% | 33.33% | 33.33% | 43.33% | 10.00% | 50.00% |
| | | D | 33.33% | 3.33% | 40.00% | 33.33% | 0.00% | 0.00% | 0.00% |
| | | E | 0.00% | 0.00% | 0.00% | 0.00% | 20.00% | 0.00% | 0.00% |
| | | F | 0.00% | 0.00% | 0.00% | 0.00% | 0.00% | 0.00% | 3.33% |
| Data-driven | Dense Subgraph Mining (K-core) | A | 0.00% | 0.00% | 0.00% | 20.00% | 0.00% | 0.00% | 0.00% |
| | | B | 0.00% | 100.00% | 0.00% | 80.00% | 100.00% | 80.00% | 40.00% |
| | | C | 0.00% | 0.00% | 0.00% | 0.00% | 0.00% | 20.00% | 60.00% |
| | | D | 100.00% | 0.00% | 100.00% | 0.00% | 0.00% | 0.00% | 0.00% |
| | Frequency Subgraph Extraction | A | 0.00% | 0.00% | 0.00% | 0.00% | 0.00% | 100.00% | 0.00% |
| | | B | 100.00% | 30.00% | 10.00% | 0.00% | 40.00% | 0.00% | 100.00% |
| | | C | 0.00% | 0.00% | 0.00% | 100.00% | 60.00% | 0.00% | 0.00% |
| | | D | 0.00% | 0.00% | 90.00% | 0.00% | 0.00% | 0.00% | 0.00% |
| | | E | 0.00% | 70.00% | 0.00% | 0.00% | 0.00% | 0.00% | 0.00% |
| | Discriminative Pattern Learning | A | 0.00% | 10.00% | 0.00% | 0.00% | 0.00% | 0.00% | 20.00% |
| | | B | 100.00% | 90.00% | 100.00% | 0.00% | 0.00% | 100.00% | 80.00% |

Table 23: Degree analysis in pattern modification tasks

| Scale | AVG. degree | Llama | Gemini | Mixtral | GPT-4 | GPT-4o | Claude | O1-mini |
|---|---|---|---|---|---|---|---|---|
| Small | 3.32 | 3.41 | 2.60 | 2.64 | 3.66 | 3.61 | 3.75 | 3.65 |
| Medium | 2.15 | 2.30 | 2.98 | 2.69 | 2.39 | 2.78 | 2.95 | 2.95 |
| Large | 2.36 | 2.80 | 2.89 | 3.10 | 2.38 | 3.03 | 3.39 | 3.15 |

to utilize different algorithms. For instance, LLMs can decompose a house pattern into separate triangle and square detections, transferring the problem into simpler tasks. (3) We observe that most LLMs prefer to list all possible combinations first and then check whether they match the target pattern. However, their accuracy varies significantly. To explore underlying failure reasons, we further calculate the precision and recall of detected patterns. These two metrics provide insight into the type of hallucinations that occur when LLMs perform pattern detection. A low precision suggests that LLMs hallucinate extra edges in the extracted patterns, whereas a low recall indicates that some edges in the input graph were overlooked by LLMs.

As shown in the Table 24, we find that LLMs achieve higher precision than recall. This indicates that most errors come from the overlooked edges. Furthermore, most LLMs show performance drops when transitioning from terminology-based to topology-based detection. The terminology helps reduce the hallucination.

Table 24: Precision and recall on the large scale of triangle dataset

|  |  | Llama | Gemini | Mixstral | GPT-4 | GPT-4o | Claude | O1-mini |
|---|---|---|---|---|---|---|---|---|
| Terminology-based | Precision | 0.599 | 0.622 | 0.543 | 0.449 | 0.609 | 0.763 | 0.776 |
|  | Recall | 0.294 | 0.493 | 0.224 | 0.039 | 0.256 | 0.304 | 0.416 |
| Topology-based | Precision | 0.190 | 0.484 | 0.507 | 0.409 | 0.586 | 0.764 | 0.765 |
|  | Recall | 0.052 | 0.308 | 0.142 | 0.050 | 0.195 | 0.249 | 0.387 |
| Decrease | Precision | -0.409 | -0.138 | -0.036 | -0.039 | -0.023 | 0.001 | -0.011 |
|  | Recall | -0.242 | -0.185 | -0.082 | 0.012 | -0.062 | -0.056 | -0.029 |

### H.4 DEGREES ANALYSIS IN K-CORE

We provide an example to explain why the 3 is important for LLMs in k-core solution. Here is an output of O1-mini:

After analyzing the given graph, we aim to identify the **3-core** subgraphs. A **k-core** of a graph is a maximal subgraph in which every vertex has at least degree **k**. ### Step-by-Step Analysis: 1. **Initial Degrees:** - **Nodes with degree $\geq$ 3:** 0, 2, 3, 8, 9, 12, 13, 15, 19 - **Nodes with degree $<$ 3:** 1, 4, 5, 6, 7, 10, 11, 14, 16, 17, 18, 20, 21
2. **Iterative Pruning:** - **First Removal:** Eliminate nodes with degree 3. This affects the remaining nodes' degrees. - **Recalculation:** After removal, the degrees of $>$ the remaining nodes drop below 3: - For example, node 0 initially has degree 3 but loses connections as other nodes are pruned. - **Subsequent Removals:** Continually removing nodes with degrees dropping below 3 leads to the eventual elimination of all nodes.
3. **Final Outcome:** - No subset of nodes remains where each node has at least degree 3. - Therefore, there are **no 3-core subgraphs** in the given graph.

We observe that LLMs do not calculate the exact node degree but instead evaluate whether it is greater or less than 3. Therefore, the critical factor for LLMs to make accurate predictions occurs when the node's degree is around 3.

## I ANALYSIS ON CHAIN-OF-THOUGHT PROMPTING

We have conducted several experiments to illustrate the effect of Chain-of-Thought prompting on both terminology-based and topology-based pattern detection tasks using edge list descriptions. Specifically, we utilize 3 cases with the reasoning process as demonstrations to require LLMs to detect triangle and house patterns in small-scale graphs and triangle patterns in medium-scale graphs. The results are summarized in Table 25.

Table 25: Performance comparison between zero-shot and CoT promptings in pattern detection.

| Method | Model | Zero-shot | | | CoT | | | Avg. Increase |
|---|---|---|---|---|---|---|---|---|
|  |  | triangle(S) | house(S) | triangle(M) | triangle(S) | house(S) | triangle(M) |  |
| Terminology-based | Gemini | 0.725 | 0.225 | 0.218 | 0.822 | 0.103 | 0.513 | 0.090 |
|  | O1-mini | 0.832 | 0.066 | 0.409 | 0.811 | 0.011 | 0.727 | 0.081 |
| Topology-based | Gemini | 0.651 | 0.122 | 0.484 | 0.767 | 0.263 | 0.596 | 0.123 |
|  | O1-mini | 0.832 | 0.000 | 0.833 | 0.736 | 0.075 | 0.756 | -0.033 |

Overall, these results indicate that CoT prompting generally enhances pattern detection performance, particularly in terminology-based tasks. However, the effect of CoT is limited when the models already acheive high scores in the zero-shot setting. This aligns with previous studies that in-context learning does not always enhance the ability of LLMs to understand graph structuresWang et al. (2024); Fatemi et al. (2023).

## J    SPLIT TABLES

Here we split the tables to make them clearer to read.

Table 26: Terminology-based graph modification in A.L. description

| A.L. | Gemini S. | Gemini M. | Gemini L. | Mixtral S. | Mixtral M. | Mixtral L. | Llama S. | Llama M. | Llama L. | Claude S. | Claude M. | Claude L. | GPT-4 S. | GPT-4 M. | GPT-4 L. | GPT-4o S. | GPT-4o M. | GPT-4o L. | O1-mini S. | O1-mini M. | O1-mini L. |
|---|---|---|---|---|---|---|---|---|---|---|---|---|---|---|---|---|---|---|---|---|---|
| S → H | 0.06 | 0.28 | 0.20 | 0.32 | 0.58 | **0.64** | 0.34 | 0.48 | 0.40 | 0.26 | 0.50 | 0.48 | 0.14 | 0.12 | 0.14 | 0.28 | 0.36 | 0.44 | **0.38** | **0.64** | 0.58 |
| S → D | 0.53 | 0.48 | 0.32 | 0.77 | 0.64 | 0.62 | 0.50 | 0.32 | 0.50 | 0.74 | 0.50 | 0.26 | 0.18 | 0.16 | 0.20 | 0.71 | 0.68 | 0.54 | **0.95** | **0.98** | **0.94** |
| D → S | 0.27 | 0.42 | 0.28 | 0.29 | 0.28 | 0.40 | 0.51 | 0.70 | 0.64 | 0.54 | 0.84 | 0.68 | 0.15 | 0.12 | 0.24 | 0.41 | 0.44 | 0.28 | **0.77** | **0.72** | **0.88** |
| F → B | 0.30 | 0.58 | 0.52 | 0.29 | 0.56 | 0.48 | 0.29 | 0.60 | 0.50 | 0.36 | 0.56 | 0.62 | 0.26 | 0.42 | 0.46 | 0.18 | 0.24 | 0.20 | **0.40** | **0.76** | **0.74** |

Table 27: Terminology-based graph modification in E.L. description

| E.L. | Gemini S. | Gemini M. | Gemini L. | Mixtral S. | Mixtral M. | Mixtral L. | Llama S. | Llama M. | Llama L. | Claude S. | Claude M. | Claude L. | GPT-4 S. | GPT-4 M. | GPT-4 L. | GPT-4o S. | GPT-4o M. | GPT-4o L. | O1-mini S. | O1-mini M. | O1-mini L. |
|---|---|---|---|---|---|---|---|---|---|---|---|---|---|---|---|---|---|---|---|---|---|
| S → H | 0.13 | 0.36 | 0.40 | 0.31 | 0.48 | **0.72** | 0.26 | 0.48 | 0.54 | 0.28 | 0.48 | 0.14 | 0.09 | 0.18 | 0.14 | 0.34 | 0.46 | 0.58 | **0.37** | **0.74** | 0.64 |
| S → D | 0.53 | 0.28 | 0.22 | 0.82 | 0.76 | 0.86 | 0.42 | 0.36 | 0.44 | 0.78 | 0.64 | 0.34 | 0.42 | 0.38 | 0.32 | 0.79 | 0.72 | 0.64 | **0.97** | **0.92** | **0.96** |
| D → S | 0.15 | 0.52 | 0.48 | 0.29 | 0.26 | 0.32 | 0.43 | 0.56 | 0.40 | 0.38 | 0.64 | 0.52 | 0.14 | 0.24 | 0.32 | 0.36 | 0.28 | 0.20 | **0.78** | **0.82** | **0.88** |
| F → B | 0.22 | 0.42 | 0.44 | 0.28 | 0.52 | 0.36 | 0.28 | 0.58 | 0.50 | 0.24 | 0.56 | 0.34 | 0.15 | 0.40 | 0.16 | 0.18 | 0.18 | 0.22 | **0.34** | **0.76** | **0.64** |

Table 28: Terminology-based graph detection in A.L. description

| Scale | Model | Undirected patterns | | | | | Dndirected patterns | | | |
|---|---|---|---|---|---|---|---|---|---|---|
| | | Triangle | T-triangle | Square | Diamond | House | V-S | FFL | FBL | D-Diamond |
| Small | GPT-4 | .632 | .151 | .069 | .113 | .006 | .352 | .389 | .191 | .448 |
| | GPT-4o | .748 | .210 | .149 | .317 | .008 | .477 | .478 | .410 | .411 |
| | Mixtral | .694 | .224 | .128 | .232 | .121 | .117 | .191 | .110 | .238 |
| | Llama | .681 | .189 | .118 | .185 | .042 | .486 | .358 | .373 | .425 |
| | Gemini | .712 | .207 | .150 | .230 | .104 | .287 | .263 | .194 | .149 |
| | Claude | .782 | .217 | .178 | .273 | .210 | .385 | .300 | .262 | .229 |
| | O1-mini | .828 | .593 | .335 | .663 | .054 | .584 | .605 | .567 | .504 |
| Medium | GPT-4 | .356 | .061 | .010 | .083 | .017 | .101 | .085 | .054 | .035 |
| | GPT-4o | .671 | .108 | .102 | .242 | .024 | .247 | .246 | .141 | .097 |
| | Mixtral | .474 | .104 | .006 | .132 | .000 | .040 | .049 | .060 | .007 |
| | Llama | .618 | .059 | .082 | .140 | .020 | .207 | .153 | .119 | .051 |
| | Gemini | .678 | .197 | .091 | .247 | .056 | .125 | .188 | .148 | .012 |
| | Claude | .725 | .144 | .130 | .205 | .038 | .130 | .181 | .104 | .006 |
| | O1-mini | .848 | .409 | .335 | .535 | .009 | .527 | .611 | .603 | .318 |
| Large | GPT-4 | .057 | .003 | .003 | .001 | .000 | .067 | .066 | .031 | .029 |
| | GPT-4o | .404 | .016 | .011 | .063 | .000 | .156 | .151 | .111 | .091 |
| | Mixtral | .319 | .016 | .005 | .064 | .000 | .016 | .046 | .034 | .008 |
| | Llama | .361 | .012 | .018 | .025 | .020 | .111 | .133 | .054 | .022 |
| | Gemini | .600 | .035 | .043 | .099 | .038 | .101 | .174 | .110 | .001 |
| | Claude | .320 | .013 | .020 | .051 | .020 | .074 | .147 | .066 | .004 |
| | O1-mini | .636 | .065 | .039 | .103 | .000 | .533 | .635 | .568 | .363 |

Table 29: Terminology-based graph detection in E.L. description

| Scale | Model | Undirected patterns | | | | | Dndirected patterns | | | |
|---|---|---|---|---|---|---|---|---|---|---|
| | | Triangle | T-triangle | Square | Diamond | House | V-S | FFL | FBL | D-Diamond |
| Small | GPT-4 | .581 | .107 | .026 | .113 | .003 | .380 | .406 | .279 | .396 |
| | GPT-4o | .702 | .250 | .132 | .309 | .053 | .490 | .450 | .357 | .360 |
| | Mixtral | .622 | .211 | .102 | .181 | .118 | .241 | .241 | .145 | .140 |
| | Llama | .693 | .193 | .106 | .195 | .021 | .527 | .388 | .349 | .367 |
| | Gemini | .725 | .274 | .176 | .262 | .225 | .281 | .267 | .193 | .111 |
| | Claude | .740 | .229 | .149 | .259 | .171 | .365 | .277 | .265 | .201 |
| | O1-mini | .832 | .578 | .316 | .684 | .066 | .600 | .602 | .584 | .488 |
| Medium | GPT-4 | .345 | .031 | .004 | .057 | .000 | .130 | .117 | .108 | .023 |
| | GPT-4o | .563 | .120 | .114 | .215 | .034 | .300 | .191 | .163 | .081 |
| | Mixtral | .478 | .118 | .032 | .125 | .007 | .053 | .149 | .081 | .003 |
| | Llama | .584 | .043 | .070 | .108 | .000 | .210 | .167 | .134 | .043 |
| | Gemini | .658 | .218 | .107 | .299 | .000 | .133 | .167 | .111 | .018 |
| | Claude | .673 | .157 | .161 | .210 | .016 | .141 | .173 | .052 | .009 |
| | O1-mini | .777 | .453 | .274 | .567 | .000 | .516 | .637 | .594 | .327 |
| Large | GPT-4 | .086 | .005 | .000 | .002 | .000 | .061 | .064 | .059 | .016 |
| | GPT-4o | .317 | .045 | .012 | .041 | .000 | .226 | .140 | .066 | .058 |
| | Mixtral | .233 | .014 | .006 | .056 | .022 | .030 | .155 | .055 | .012 |
| | Llama | .384 | .013 | .011 | .030 | .020 | .123 | .167 | .073 | .014 |
| | Gemini | .496 | .014 | .011 | .063 | .047 | .103 | .166 | .137 | .000 |
| | Claude | .278 | .020 | .023 | .044 | .020 | .054 | .138 | .078 | .007 |
| | O1-mini | .428 | .072 | .026 | .025 | .000 | .560 | .637 | .592 | .367 |

Table 30: Topology-based graph modification in A.L. description

| A.L. | Gemini | | | Mixtral | | | Llama | | | Claude | | | GPT-4 | | | GPT-4o | | | O1-mini | | |
|---|---|---|---|---|---|---|---|---|---|---|---|---|---|---|---|---|---|---|---|---|---|
| | S. | M. | L. | S. | M. | L. | S. | M. | L. | S. | M. | L. | S. | M. | L. | S. | M. | L. | S. | M. | L. |
| S → H | 0.20 | 0.34 | 0.20 | **0.78** | 0.58 | 0.66 | 0.57 | 0.52 | 0.48 | 0.45 | 0.64 | 0.44 | 0.34 | 0.56 | 0.56 | 0.67 | 0.70 | 0.68 | 0.61 | **0.82** | **0.80** |
| S → D | 0.45 | 0.34 | 0.32 | 0.78 | 0.66 | 0.60 | 0.65 | 0.50 | 0.74 | 0.69 | 0.62 | 0.60 | 0.48 | 0.56 | 0.40 | 0.75 | 0.68 | 0.66 | **0.82** | **0.76** | **0.86** |
| D → S | 0.12 | 0.24 | 0.12 | **0.58** | 0.42 | 0.60 | 0.29 | 0.26 | 0.40 | 0.12 | 0.26 | 0.36 | 0.16 | 0.48 | 0.40 | 0.41 | 0.40 | 0.48 | 0.55 | **0.56** | **0.64** |
| F → B | 0.28 | 0.38 | 0.50 | 0.52 | 0.40 | 0.64 | 0.55 | 0.68 | 0.70 | 0.37 | 0.54 | 0.40 | 0.48 | 0.68 | 0.76 | **0.57** | **0.80** | **0.80** | 0.41 | 0.76 | 0.68 |

Table 31: Topology-based graph modification in E.L. description

| E.L. | Gemini | | | Mixtral | | | Llama | | | Claude | | | GPT-4 | | | GPT-4o | | | O1-mini | | |
|---|---|---|---|---|---|---|---|---|---|---|---|---|---|---|---|---|---|---|---|---|---|
| | S. | M. | L. | S. | M. | L. | S. | M. | L. | S. | M. | L. | S. | M. | L. | S. | M. | L. | S. | M. | L. |
| S → H | 0.22 | 0.42 | 0.26 | **0.66** | **0.68** | 0.54 | 0.45 | 0.50 | 0.48 | 0.38 | 0.54 | 0.42 | 0.24 | 0.60 | 0.52 | 0.36 | 0.52 | 0.60 | 0.46 | **0.68** | **0.74** |
| S → D | 0.48 | 0.40 | 0.52 | 0.67 | 0.70 | 0.62 | 0.69 | 0.46 | 0.62 | 0.72 | 0.74 | 0.74 | 0.67 | 0.62 | 0.20 | 0.67 | 0.68 | 0.74 | **0.88** | **0.86** | **0.84** |
| D → S | 0.11 | 0.32 | 0.32 | **0.40** | 0.32 | 0.44 | 0.29 | 0.30 | 0.20 | 0.11 | 0.22 | 0.48 | 0.06 | 0.30 | 0.20 | 0.20 | 0.18 | **0.36** | 0.24 | **0.36** | 0.28 |
| F → B | 0.20 | 0.24 | 0.32 | 0.34 | 0.52 | 0.40 | 0.41 | 0.50 | 0.44 | 0.29 | 0.44 | 0.34 | 0.27 | 0.46 | 0.48 | **0.44** | 0.62 | 0.52 | 0.32 | **0.66** | **0.66** |

Table 32: Topology-based graph pattern detection in A.L. description

| Scale | Model | Undirected patterns | | | | | Dndirected patterns | | | |
|---|---|---|---|---|---|---|---|---|---|---|
| | | Triangle | T-triangle | Square | Diamond | House | V-S | FFL | FBL | D-Diamond |
| Small | GPT-4 | .653 | .051 | .041 | .162 | .002 | .415 | .346 | .423 | .477 |
| | GPT-4o | .717 | .194 | .118 | .233 | .005 | .537 | .296 | .407 | .285 |
| | Mixtral | .520 | .108 | .132 | .170 | .105 | .191 | .287 | .276 | .255 |
| | Llama | .545 | .122 | .076 | .112 | .005 | .476 | .266 | .382 | .388 |
| | Gemini | .651 | .166 | .133 | .177 | .122 | .304 | .266 | .194 | .203 |
| | Claude | .730 | .186 | .122 | .263 | .241 | .370 | .302 | .322 | .206 |
| | O1-mini | .832 | .617 | .365 | .633 | .107 | .588 | .678 | .572 | .492 |
| Medium | GPT-4 | .273 | .032 | .012 | .064 | .000 | .177 | .072 | .129 | .051 |
| | GPT-4o | .657 | .183 | .086 | .143 | .000 | .298 | .072 | .146 | .046 |
| | Mixtral | .414 | .057 | .039 | .051 | .000 | .040 | .188 | .089 | .021 |
| | Llama | .299 | .029 | .050 | .081 | .000 | .193 | .094 | .059 | .061 |
| | Gemini | .484 | .037 | .123 | .033 | .000 | .140 | .170 | .109 | .062 |
| | Claude | .671 | .100 | .097 | .247 | .040 | .310 | .204 | .084 | .019 |
| | O1-mini | .833 | .488 | .449 | .494 | .000 | .471 | .690 | .567 | .298 |
| Large | GPT-4 | .067 | .004 | .000 | .003 | .000 | .194 | .068 | .108 | .007 |
| | GPT-4o | .327 | .018 | .003 | .027 | .000 | .240 | .065 | .119 | .055 |
| | Mixtral | .177 | .006 | .006 | .046 | .000 | .043 | .144 | .021 | .006 |
| | Llama | .085 | .012 | .012 | .047 | .020 | .042 | .106 | .020 | .003 |
| | Gemini | .186 | .001 | .003 | .002 | .000 | .064 | .122 | .098 | .016 |
| | Claude | .454 | .011 | .010 | .048 | .000 | .188 | .151 | .107 | .011 |
| | O1-mini | .511 | .045 | .044 | .134 | .000 | .518 | .608 | .591 | .340 |

Table 33: Topology-based graph pattern detection in E.L. description

| Scale | Model | Undirected patterns | | | | | Dndirected patterns | | | |
|---|---|---|---|---|---|---|---|---|---|---|
| | | Triangle | T-triangle | Square | Diamond | House | V-S | FFL | FBL | D-Diamond |
| Small | GPT-4 | .584 | .056 | .039 | .110 | .010 | .475 | .429 | .422 | .397 |
| | GPT-4o | .706 | .143 | .111 | .227 | .015 | .489 | .383 | .287 | .269 |
| | Mixtral | .550 | .134 | .107 | .157 | .070 | .227 | .364 | .352 | .257 |
| | Llama | .678 | .145 | .074 | .157 | .097 | .488 | .361 | .386 | .283 |
| | Gemini | .696 | .192 | .175 | .171 | .192 | .171 | .230 | .205 | .126 |
| | Claude | .713 | .269 | .160 | .250 | .249 | .329 | .393 | .257 | .128 |
| | O1-mini | .821 | .499 | .364 | .574 | .249 | .599 | .670 | .566 | .477 |
| Medium | GPT-4 | .329 | .047 | .045 | .068 | .000 | .256 | .210 | .099 | .041 |
| | GPT-4o | .575 | .089 | .066 | .205 | .000 | .289 | .258 | .154 | .053 |
| | Mixtral | .430 | .074 | .051 | .139 | .000 | .127 | .180 | .095 | .029 |
| | Llama | .290 | .032 | .061 | .094 | .020 | .213 | .134 | .124 | .035 |
| | Gemini | .632 | .075 | .088 | .124 | .030 | .037 | .162 | .098 | .000 |
| | Claude | .651 | .159 | .127 | .253 | .040 | .166 | .233 | .092 | .035 |
| | O1-mini | .749 | .367 | .417 | .557 | .024 | .496 | .625 | .591 | .301 |
| Large | GPT-4 | .113 | .002 | .002 | .005 | .000 | .230 | .133 | .085 | .010 |
| | GPT-4o | .257 | .015 | .008 | .055 | .000 | .158 | .180 | .096 | .014 |
| | Mixtral | .262 | .010 | .009 | .025 | .000 | .086 | .184 | .037 | .013 |
| | Llama | .068 | .004 | .001 | .008 | .020 | .094 | .070 | .134 | .013 |
| | Gemini | .535 | .000 | .061 | .005 | .027 | .025 | .152 | .074 | .000 |
| | Claude | .286 | .015 | .015 | .072 | .040 | .162 | .170 | .080 | .040 |
| | O1-mini | .514 | .049 | .050 | .114 | .000 | .488 | .625 | .595 | .343 |

Table 34: Frequency subgraph extraction in A.L. description

| A.L. | GPT-4 | | | GPT-4o | | | Mixtral | | | Llama | | | Gemini | | | Claude | | | O1-mini | | |
|---|---|---|---|---|---|---|---|---|---|---|---|---|---|---|---|---|---|---|---|---|---|
| | S. | M. | L. | S. | M. | L. | S. | M. | L. | S. | M. | L. | S. | M. | L. | S. | M. | L. | S. | M. | L. |
| Triangle | 1.00 | 1.00 | 1.00 | 0.84 | 0.88 | 0.88 | 0.86 | 0.88 | 1.00 | 0.62 | 0.68 | 0.87 | 1.00 | 1.00 | 0.59 | 0.43 | 0.26 | 0.09 | 1.00 | 1.00 | 1.00 |
| Square | 1.00 | 1.00 | 1.00 | 1.00 | 0.86 | 1.00 | 0.69 | 0.95 | 1.00 | 0.61 | 0.94 | 1.00 | 0.75 | 0.47 | 0.40 | 0.30 | 0.37 | 0.07 | 1.00 | 1.00 | 1.00 |
| Diamond | 1.00 | 1.00 | 1.00 | 0.89 | 1.00 | 0.50 | 0.92 | 1.00 | 1.00 | 0.43 | 0.52 | 0.96 | 1.00 | 1.00 | 0.83 | 0.34 | 0.33 | 0.14 | 1.00 | 1.00 | 1.00 |
| House | 1.00 | 1.00 | 1.00 | 1.00 | 1.00 | 0.98 | 0.73 | 1.00 | 1.00 | 0.53 | 0.66 | 0.93 | 1.00 | 0.78 | 0.10 | 0.31 | 0.32 | 0.22 | 1.00 | 1.00 | 1.00 |
| FFL | 1.00 | 1.00 | 1.00 | 0.91 | 0.64 | 1.00 | 0.89 | 1.00 | 0.86 | 0.67 | 0.68 | 0.93 | 0.89 | 0.00 | 1.00 | 0.33 | 0.32 | 0.26 | 1.00 | 1.00 | 1.00 |
| FBL | 1.00 | 1.00 | 1.00 | 0.78 | 0.89 | 0.90 | 1.00 | 1.00 | 0.59 | 0.77 | 0.67 | 0.87 | 1.00 | 1.00 | 1.00 | 0.46 | 0.22 | 0.40 | 1.00 | 1.00 | 0.85 |
| D-Diamond | 1.00 | 1.00 | 1.00 | 0.81 | 0.95 | 1.00 | 1.00 | 1.00 | 0.67 | 0.54 | 0.73 | 0.91 | 0.67 | 0.00 | 1.00 | 0.41 | 0.42 | 0.35 | 1.00 | 1.00 | 1.00 |

Table 35: Frequency subgraph extraction in E.L. description

| E.L. | GPT-4 | | | GPT-4o | | | Mixtral | | | Llama | | | Gemini | | | Claude | | | O1-mini | | |
|---|---|---|---|---|---|---|---|---|---|---|---|---|---|---|---|---|---|---|---|---|---|
| | S. | M. | L. | S. | M. | L. | S. | M. | L. | S. | M. | L. | S. | M. | L. | S. | M. | L. | S. | M. | L. |
| Triangle | 1.00 | 1.00 | 1.00 | 0.94 | 1.00 | 0.81 | 1.00 | 1.00 | 1.00 | 0.94 | 0.95 | 1.00 | 1.00 | 0.57 | 0.29 | 0.84 | 0.58 | 0.32 | 1.00 | 1.00 | 1.00 |
| Square | 1.00 | 1.00 | 1.00 | 0.97 | 0.71 | 0.98 | 1.00 | 1.00 | 1.00 | 0.94 | 0.93 | 0.91 | 0.80 | 1.00 | 0.17 | 0.93 | 0.67 | 0.21 | 1.00 | 1.00 | 1.00 |
| Diamond | 1.00 | 1.00 | 1.00 | 1.00 | 0.87 | 0.93 | 1.00 | 1.00 | 1.00 | 0.99 | 0.93 | 0.91 | 1.00 | 0.38 | 0.52 | 0.99 | 0.68 | 0.32 | 1.00 | 1.00 | 1.00 |
| House | 1.00 | 1.00 | 1.00 | 1.00 | 1.00 | 0.76 | 1.00 | 1.00 | 1.00 | 1.00 | 0.74 | 0.95 | 1.00 | 0.33 | 0.18 | 1.00 | 0.55 | 0.44 | 1.00 | 1.00 | 1.00 |
| FFL | 1.00 | 1.00 | 1.00 | 0.96 | 0.67 | 0.92 | 1.00 | 1.00 | 1.00 | 0.92 | 1.00 | 1.00 | 1.00 | 1.00 | 1.00 | 0.68 | 0.52 | 0.34 | 1.00 | 1.00 | 1.00 |
| FBL | 1.00 | 1.00 | 1.00 | 0.98 | 0.75 | 0.83 | 1.00 | 1.00 | 1.00 | 0.96 | 1.00 | 1.00 | 1.00 | 1.00 | 1.00 | 0.84 | 0.45 | 0.31 | 1.00 | 1.00 | 1.00 |
| D-Diamond | 1.00 | 1.00 | 1.00 | 0.85 | 0.88 | 0.66 | 1.00 | 0.60 | 1.00 | 0.78 | 1.00 | 1.00 | 1.00 | 0.75 | 1.00 | 0.61 | 0.46 | 0.28 | 1.00 | 1.00 | 1.00 |

