# OpenReview forum: "How Do Large Language Models Understand Graph Patterns? A Benchmark for Graph Pattern Comprehension"
_ICLR.cc/2025/Conference — ICLR 2025 Poster_

### Official Review · Reviewer_jJDa · 2024-10-29

**Soundness:** 3
**Presentation:** 4
**Contribution:** 1
**Rating:** 6
**Confidence:** 4

**Summary:**

The paper avaluates large language models (LLMs) on graph pattern comprehension. It explores three types of descriptions: terminology-based, topology-based, and data-drive and evaluates model performance across synthetic and real-world datasets. The study highlights LLMs' limitations and potential for recognizing graph patterns, especially with the inclusion of recent models like O1-mini.

**Strengths:**

1. The paper includes O1, a very recent model specifically designed for reasoning, highlighting its capabilities and limitations for graph tasks and revealing that there is still room for improvement in how LLMs handle graphs.
2. The paper is well-written, with good clarity and well-organized explanations, making it accessible for readers. The consideration of multiple input formats serves as a useful starting point for anyone new to the field of graph tasks using LLMs.
3. The range of tasks, spanning both synthetic and real-world datasets, provides a comprehensive evaluation of LLM performance on graph-related tasks.

**Weaknesses:**

1. Lack of Novelty:
The paper’s findings align with existing research, notably with studies like [1], which already demonstrate that LLMs have limited graph understanding. Although the inclusion of O1 is new and valuable, most results are expected and reflect known limitations of other LLMs in graph comprehension.

2. Predictable Results:
The finding that “formatting input data to align with pretraining knowledge can enhance performance” is elementary and expected in LLM research. This does not offer a significant new insight and detracts from the paper's contribution.

3. Limitations of Terminology-Based Approaches:
While terminology-based descriptions can be effective for small, simple graphs, they become impractical for larger, denser graphs with multiple cycles, squares, and complex structures. The approach lacks scalability, which is a significant drawback in the context of graph pattern tasks.

4. Overlap with Existing Work:
Prior studies, such as [2] have already evaluated multiple topology-based prompts across diverse tasks, revealing similar findings. This paper’s contribution is limited since it doesn’t introduce substantial new insights beyond these previous efforts.


[1] Wang, Heng, et al. "Can language models solve graph problems in natural language?." Advances in Neural Information Processing Systems 36 (2024).

[2] Fatemi, Bahare, Jonathan Halcrow, and Bryan Perozzi. "Talk like a graph: Encoding graphs for large language models." arXiv preprint arXiv:2310.04560 (2023).

**Questions:**

1. In Section 3.1, what is the temperature used for evaluating diversity? Would increasing the temperature lead to more diversity?
2. In Section 5.1, the paper mentions that “LLMs tend to make errors when node degrees are close to 3.” Could the authors clarify why this happens?
3. In Section 6, is the test set complete, or is only a subset used for testing? Section C.3 does not clearly clarify this.
4. In the data-driven approach, what is the input format? Are these images?
5. Would fine-tuning be a feasible strategy to improve accuracy for real-world graph tasks?

---

> ### Author Response · Authors · 2024-11-23
> **Overview**
>
> Thank you for your valuable suggestions! Before addressing your questions, we would like to emphasize that our paper belongs to the dataset and benchmark track, aiming to provide various synthetic and real-world datasets for evaluating the performance of LLMs in graph mining tasks.
>
> Unlike graph algorithm problems, which focus on solving well-defined tasks such as finding shortest paths or detecting cycles, graph mining involves discovering patterns, insights, or useful information from graph-structured data. The goal is to uncover hidden or implicit patterns, such as frequent subgraphs, community structures, or anomalies. This process is critical in many real-world applications, including finance, chemistry, biology, and social networks.
>
> Previous works have already provided comprehensive benchmarking on graph algorithm problems, as highlighted in the papers you referenced. However, due to the distinct nature and important real-world applications of graph mining tasks, our work introduces a novel benchmarking framework focused specifically on graph pattern discovery.

---

> ### Author Response · Authors · 2024-11-23
> **Weakness 1**
>
> >W1: Lack of Novelty: The paper’s findings align with existing research, notably with studies like [1], which already demonstrate that LLMs have limited graph understanding. Although the inclusion of O1 is new and valuable, most results are expected and reflect known limitations of other LLMs in graph comprehension.
>
> Thanks for your advice. NLGraph [1] studied the graph reasoning ability of LLMs in solving traditional graph algorithm problems, such as connectivity, shortest path and maximum flow. In contrast, we focus on tasks related to graph patterns, including pattern modification and detection, frequent subgraph mining and discriminative pattern learning. These tasks belong to graph mining tasks. These two categories differ significantly in terms of task input formats, the abilities they require from LLMs, and their real-world applications.
>
>
> ### (1) Task Input Formats
> NLGraph focuses on the algorithmic tasks applied to a single graph input. In our study, however, we evaluate LLMs  using tasks that involve multiple graph inputs. For example, we need to prompt multiple graphs in frequent subgraph mining, which requires a more nuanced understanding and comparative analysis across graphs rather than operations within a single graph.
>
> ### (2) Required Abilities
>
> Although they both require LLMs to have a foundational understanding of graph structures, the additional abilities needed are notably different.
> Most graph algorithm problems demand deterministic reasoning, requiring LLMs to perform step-by-step logical problem-solving based on well-defined algorithms. In contrast, graph pattern mining problems are more exploratory. Instead of deterministic calculations, LLMs need to leverage knowledge of concepts of graph patterns, and employ heuristic or probabilistic searches to identify patterns within inputs. This requires common-sense reasoning and the ability to generalize across graphs.
>
> ### (3) Real-World Applications
>
> NLGraph lacks evaluations on real-world datasets. Typically, graph algorithm problems are applied in domains such as transportation scheduling and resource allocation, where precise computations are critical. In contrast, our study involves real-world datasets such as molecular graphs (chemical structure analysis) and social network graphs (community detection), which are often overlooked in previous works.
>
>
> ### Different from [1], we also have many unique insights.  We list a few representative ones below:
>
> (1) While O1 outperforms other LLMs in most tasks as expected, it may underperform in certain tasks, such as the isomorphic mapping and discriminative pattern learning, where it ranks only 5th and 6th, respectively. Interestingly, GPT-4o demonstrates more balanced performance across all tasks compared to O1, ranking at least 4th.
>
> (2) As for the choice of terminology and topology to describe graph patterns, we find that using terminology for well-known patterns is more effective. This approach allows LLMs to leverage internal knowledge, thereby improving their understanding and reasoning capabilities.
>
> (3) Previous work recommended using adjacency list and edge list formats to describe general graphs. Building on this insight, we find that the optimal input graph format depends on the specific task and the algorithm employed by LLMs to solve it. For example, in discriminative pattern learning, the algorithms used by LLMs often rely on edge combinations, making edge comparisons highly relevant. In this scenario, the adjacency list achieves only a 0.9 score with Gemini, while the edge list allows GPT-4o and Claude to reach a perfect score of 1.0. Conversely, in k-core detection, the algorithms require node degree and the edge list is inferior to the adjacency list.

---

> ### Author Response · Authors · 2024-11-23
> **Weakness 2 -  4**
>
> >W2: Predictable Results: The finding that “formatting input data to align with pretraining knowledge can enhance performance” is elementary and expected in LLM research. This does not offer a significant new insight and detracts from the paper's contribution.
>
> Thank you for giving us the opportunity to clarify this. Although this statement is a well-established practice in LLM research, but how to formulate it in different applications is unknown. Our work specifies this principle to graph mining tasks, which differ from graph algorithm problems and have been less extensively studied. This distinction also allows us to provide new insights.
>
> First of all, graph mining tasks require graph pattern inputs, so we explore how to describe the patterns, comparing terminology-based and topology-based approaches. Our findings indicate that terminology-based descriptions are more effective for well-known patterns because these terminologies are more aligned with pretraining knowledge. Moreover, we extend this to real-world applications. Our experiments show that the terminology of functional groups can enhance LLMs' ability to predict the properties of molecules.
>
> Secondly, previous work suggested the adjacency list and edge list formats are better to translate the graph-structure data into textual description. Building on this, we find that the optimal input graph format depends on the specific task and the algorithm employed by LLMs to solve it. For example, in discriminative pattern learning, the algorithms used by LLMs often rely on edge combinations, making edge comparisons highly relevant. In this scenario, the adjacency list achieves only a 0.9 score with Gemini, while the edge list allows GPT-4o and Claude to reach a perfect score of 1.0. Conversely, when tasks require consideration of node information, such as in k-core detection, the adjacency list tends to perform better than the edge list. For instance, in the medium dataset, GPT-4o achieves the highest accuracy of 1.0 using the adjacency list, while Claude and O1-mini achieve only 0.88 with the edge list.
>
> > W3: Limitations of Terminology-Based Approaches: While terminology-based descriptions can be effective for small, simple graphs, they become impractical for larger, denser graphs with multiple cycles, squares, and complex structures. The approach lacks scalability, which is a significant drawback in the context of graph pattern tasks.
>
> We’re glad to have the chance to clarify this for you. Please note that graph pattern tasks have two types of inputs, the target graph pattern and an input graph. In our study, we only use terminology-based approaches to describe well-known graph patterns, not for input graphs. For instance, in molecular property prediction, functional groups are often assigned specific names and formal definitions. We use terminology to describe these functional groups, and use topology to describe the input molecules. The use of terminology enables LLMs to leverage established domain knowledge, enhancing their understanding and reasoning capabilities. Our experiments with real-world datasets demonstrate that these widely-used terminologies significantly improve LLM performance.
>
> For more complex graph patterns, we also employ topology-based approaches. These descriptions are inherently scalable and can describe any defined pattern. By combining terminology-based and topology-based descriptions, we provide a comprehensive solution for diverse graph patterns.
>
> >W4: Overlap with Existing Work: Prior studies, such as [2] have already evaluated multiple topology-based prompts across diverse tasks, revealing similar findings.
>
> Thank you for giving us the space to clarify this point. In [2], the authors utilized several topology-based descriptions to encode input graphs and evaluate LLMs' reasoning ability in many graph tasks, including edge existence, node degree, node count, edge count, connectivity and cycle check. They also demonstrated that the choice of topology-based prompts significantly impacts the results. Based on the average accuracy reported in Table 5 of [2], we selected adjacency lists and edge lists as our topology-based descriptions.
>
> Furthermore, we find that the optimal input graph format depends on the specific task and the algorithm employed by LLMs to solve it. For example, in discriminative pattern learning, the algorithms used by LLMs often rely on edge combinations, making edge comparisons highly relevant. In this scenario, the adjacency list achieves only a 0.9 score with Gemini, while the edge list allows GPT-4o and Claude to reach a perfect score of 1.0. Conversely, when tasks require consideration of node information, such as in k-core detection, the adjacency list tends to perform better than the edge list. For instance, in the medium dataset, GPT-4o achieves the highest accuracy of 1.0 using the adjacency list, while Claude and O1-mini achieve only 0.88 with the edge list.

---

> ### Author Response · Authors · 2024-11-23
> **Q1 - Q4**
>
> > Q1: In Section 3.1, what is the temperature used for evaluating diversity? Would increasing the temperature lead to more diversity?
>
> Thank you for your thoughtful question. LLMs do not always produce consistent results, even when the temperature is set to 0. This is because most LLMs operate in parallel systems, which can involve probabilistic random sampling on GPUs, unpredictable execution order of subsystems, and variations in floating-point arithmetic implementations. Several papers, such as [1][2], examine the impact of GPU-related factors, while [3] investigates the non-deterministic behavior of LLMs, even at a temperature of 0 with a fixed random seed.
>
> [1] Pham H V, Qian S, Wang J, et al. Problems and opportunities in training deep learning software systems: An analysis of variance[C]//Proceedings of the 35th IEEE/ACM international conference on automated software engineering. 2020: 771-783.
>
> [2] Hagmann M, Meier P, Riezler S. Towards Inferential Reproducibility of Machine Learning Research[C]//The Eleventh International Conference on Learning Representations 2023.
>
> [3] Blackwell R E, Barry J, Cohn A G. Towards Reproducible LLM Evaluation: Quantifying Uncertainty in LLM Benchmark Scores[J]. arXiv preprint arXiv:2410.03492, 2024.
>
>
> Temperature significantly affects the diversity of generated graphs. We tested temperatures ranging from 0 to 1 using Llama to generate triangles, and the results are as follows:
>
> |     | t=0  | t=0.5 | t=1.0 |
> |-----|------|-------|-------|
> | ACC |    1.00 |  0.76 |   0.70 |
> | DIV | 0.48 |  0.78 |  0.85 |
>
> The diversity score increases from 0.48 to 0.85, while the accuracy decreases from 1.00 to 0.70.
>
> > Q2: In Section 5.1, the paper mentions that “LLMs tend to make errors when node degrees are close to 3.” Could the authors clarify why this happens?
>
> Thank you for the opportunity to clarify this. LLMs use an estimation method to determine which nodes should be retained. Here is an example of how LLMs handle a k-core algorithm using O1-mini:
> >>After analyzing the given graph, we aim to identify the **3-core** subgraphs. A **k-core** of a graph is a maximal subgraph in which every vertex has at least degree **k**.
> >>
> >>### Step-by-Step Analysis:
> >>
> >>1. **Initial Degrees:**
> >>   - **Nodes with degree ≥ 3:** 0, 2, 3, 8, 9, 12, 13, 15, 19
> >>   - **Nodes with degree < 3:** 1, 4, 5, 6, 7, 10, 11, 14, 16, 17, 18, 20, 21
> >>
> >>2. **Iterative Pruning:**
> >>   - **First Removal:** Eliminate nodes with degree 3. This affects the remaining nodes' degrees.
> >>   - **Recalculation:** After removal, the degrees of >the remaining nodes drop below 3:
>      - For example, node 0 initially has degree 3 but loses connections as other nodes are pruned.
> >>   - **Subsequent Removals:** Continually removing nodes with degrees dropping below 3 leads to the eventual elimination of all nodes.
> >>
> >>3. **Final Outcome:**
> >>   - No subset of nodes remains where each node has at least degree 3.
> >>   - Therefore, there are **no 3-core subgraphs** in the given graph.
>
>
> We observe that LLMs do not calculate the exact node degree but instead evaluate whether it is greater or less than 3. Therefore, the critical factor for LLMs to make accurate predictions occurs when the node's degree is around 3.
>
> We've added this example in Appendix I.4.
>
> >Q3: In Section 6, is the test set complete, or is only a subset used for testing? Section C.3 does not clearly clarify this.
>
> We appreciate the opportunity to clarify this. We have two types of datasets: one for discriminative pattern learning and the other for classification. Both are randomly selected subsets of the original real-world dataset. The primary reason for not using the entire dataset is the high computational cost of testing all samples. We believe this subset is sufficient to represent the original dataset effectively in our experiments.
>
> > Q4: In the data-driven approach, what is the input format? Are these images?
>
> Thank you for your question. We do not use any images in our experiments. In a data-driven approach, LLMs extract undefined patterns from the provided data, and the input formats are textual representations of graphs, not images. For example, in discriminative pattern learning, the prompt is
>
> "You are provided with two sets of graphs.
> The first set is:
> The No.1. graph is [adjacency list description].
> The No.2. graph is [adjacency list description].
>
> The second set is:
> The No.1. graph is [adjacency list description].
> The No.2. graph is [adjacency list description].
>
> What are the differences between these two sets of graphs? Identify the discriminative pattern in set 1."

---

> ### Author Response · Authors · 2024-11-23
> **Q5**
>
> > Q5: Would fine-tuning be a feasible strategy to improve accuracy for real-world graph tasks?
>
> Thanks for your comments. We agree that fine-tuning can be an efficient method to improve the accuracy of real-world graph tasks. However, we lack the resources to fine-tune closed-source LLMs, such as the GPT series, or resource-intensive models like Llama-3.1-405B. To check the potential of fine-tuning, we experimented with a smaller model, Llama-3.2-3B, fine-tuned using LoRA. We evaluated the impact of fine-tuning on pattern detection under topology-based conditions, including Benzene, Alkane-Carbonyl (R-CO), and Fluoride-Carbonyl (F-CO) patterns. The graphs were represented using edge lists, and the model was trained on 1,100 samples (1000 for training and 100 for validation), which were excluded from the test sets.
>
> | F1-Score | R-CO | F-CO  | Benzene |
> |-----|-----|------|-------|
> Finetuned on LLama-3B | 0.56 | 0.73 | 0.57 |
> zero shot on LLama-3B   | 0.49 | 0.67 | 0.60 |
> zero shot on GPT-4   | 0.67 | 0.69 | 0.78 |
>
>
> The results indicate that fine-tuning significantly enhances the capabilities of LLMs. In the F-CO case, a small fine-tuned model even outperformed GPT-4, demonstrating the effectiveness of this approach in boosting LLM performance. This finding offers valuable insights for future research and exploration.

---

> > ### Comment · Reviewer_jJDa · 2024-11-24
> >
> > Thank you for the detailed response. My concerns have been fully addressed. While I still feel that the idea may not be highly novel, it is certainly heading in the right direction. Consequently, I would like to revise my score.

---

> > > ### Author Response · Authors · 2024-11-24
> > >
> > > Thanks for your feedback. We'll include the changes in the rebuttal and highlight the potential directions in the revision.

---

### Official Review · Reviewer_Dygb · 2024-10-30

**Soundness:** 3
**Presentation:** 3
**Contribution:** 3
**Rating:** 8
**Confidence:** 5

**Summary:**

Summary:
The author proposed a new benchmark evaluates LLMs in graph pattern recognition. While LLMs show early abilities in understanding graph structures, their potential in pattern mining is under-explored. This benchmark, covering 11 tasks and 7 models, tests LLMs' capacity to recognize and discover patterns in synthetic and real data. Key findings show O1-mini outperforms in most tasks, aligning data formatting with pretrained knowledge enhances performance, and LLMs use unique strategies distinct from traditional algorithms.

**Strengths:**

Pros:
- The paper provide benchmark for evaluating LLMs’ ability in understanding graph patterns.
- Analysis has been conducted based on the proposed benchmark. Multiple research questions have been studied.

**Weaknesses:**

Cons:
- It would be great if o1-preview result can also be included, if feasible.
- In molecular graphs, how is the molecule features being provided to the LLMs? I am curious about how the molecular graph is being converted to textual format and feed into the LLMs.  More details are encouraged to be included. If edge lists is utilized, then example of the edge list representing molecules are encouraged to be shown.
- For the question, Can LLMs automatically discover graph patterns in real-world applications? A work using LLMs to find patterns in molecular data is encourage to be mentioned. The work has tried to use LLMs to identify key functional groups in molecular data for diverse molecular property prediction tasks[1].   [1] Zheng, Y., Koh, H. Y., Ju, J., Nguyen, A. T., May, L. T., Webb, G. I., & Pan, S. (2023). Large language models for scientific synthesis, inference and explanation. arXiv preprint arXiv:2310.07984.

**Questions:**

Same as Cons

---

> ### Author Response · Authors · 2024-11-23
> **Weakness 1 -  3**
>
> > W1: It would be great if o1-preview result can also be included, if feasible.
>
> We agree that including O1-preview is highly valuable for evaluating the capabilities of LLMs. However, the cost of using O1-preview is prohibitively high. We have conducted the pattern translation task using O1-preview, and the results are as follows.
>
> |            | Triangle |      | T-Triangle |      | Diamond |      | Cost |
> |------------|:--------:|:----:|:----------:|:----:|:-------:|:----:|------|
> |            | ACC      | DIV  | ACC        | DIV  | ACC     | DIV  |      |
> | O1-mini    | 1.00     | 0.79 | 0.62       | 0.70 | 0.74    | 0.87 | 1.5$ |
> | O1-preview |     1.00 | 0.82 |       0.84 | 0.58 |    1.00 | 0.89 | 32$  |
>
> Overall, we believe that O1-preview would yield better results, but it is too costly to use for all experiments.
>
>
> > W2: In molecular graphs, how is the molecule features being provided to the LLMs? I am curious about how the molecular graph is being converted to textual format and feed into the LLMs. More details are encouraged to be included. If edge lists is utilized, then example of the edge list representing molecules are encouraged to be shown.
>
> Thank you for the opportunity to clarify this. Specifically, we employ two different methods for graph description: adjacency list (A.L.) and edge list (E.L.). The conversion process involves three steps:
> * 1. Using the function Chem.MolFromSmiles from the Chem library in Python, we extract the atoms and adjacency matrix of a given molecule from its SMILES representation.
> * 2. The atom and adjacency matrix information is used to construct an undirected graph with the Python tool networkx.Graph.
> * 3. The graph is then described using node and edge information in either adjacency list (A.L.) or edge list (E.L.) format.
>
> Taking a molecular graph with the SMILES of "C(C(=O)[O-])NC(=[NH2+])N" as an example, the molecular graph can be converted to textual format as expressed in the following table:
>
>
> | A.L. | E.L.    |
> |:-----------:|:-----------:|
> | G describes an undirected graph among 0, 1, 2, 3, 4, 5, 6, and 7. In this graph:\nNode 0 (atom: C) is connected to nodes 1 (atom: C), 4 (atom: N).\nNode 1 (atom: C) is connected to nodes 0 (atom: C), 2 (atom: O), 3 (atom: O).\nNode 2 (atom: O) is connected to nodes 1 (atom: C).\nNode 3 (atom: O) is connected to nodes 1 (atom: C).\nNode 4 (atom: N) is connected to nodes 0 (atom: C), 5 (atom: C).\nNode 5 (atom: C) is connected to nodes 4 (atom: N), 6 (atom: N), 7 (atom: N).\nNode 6 (atom: N) is connected to nodes 5 (atom: C).\nNode 7 (atom: N) is connected to nodes 5 (atom: C). | G describes an undirected graph among node 0, 1, 2, 3, 4, 5, 6, and 7.\nNode 0 (atom: C) is connected to Node 1 (atom: C).\nNode 0 (atom: C) is connected to Node 4 (atom: N).\nNode 1 (atom: C) is connected to Node 2 (atom: O).\nNode 1 (atom: C) is connected to Node 3 (atom: O).\nNode 4 (atom: N) is connected to Node 5 (atom: C).\nNode 5 (atom: C) is connected to Node 6 (atom: N).\nNode 5 (atom: C) is connected to Node 7 (atom: N). |
>
> Notably, the atom H is omitted by default in the *Chem* tool for conciseness.
>
> Note that we do not include edge features in our experiments, but it is feasible to incorporate them as part of the description. For example, a double bond between Carbon and Oxygen could be described as: Node 0 (atom: C) is connected to Node 1 (atom: O) via a double bond. This could be an interesting direction for future exploration.
>
> We've added this in the Appendix.C.4.
>
> > W3: For the question, Can LLMs automatically discover graph patterns in real-world applications? A work using LLMs to find patterns in molecular data is encourage to be mentioned. The work has tried to use LLMs to identify key functional groups in molecular data for diverse molecular property prediction tasks[1].[1] Zheng, Y., Koh, H. Y., Ju, J., Nguyen, A. T., May, L. T., Webb, G. I., & Pan, S. (2023). Large language models for scientific synthesis, inference and explanation. arXiv preprint arXiv:2310.07984.
>
> Thank you for your valuable insights. The previous work [1] discusses molecular patterns in SMILES, a molecule-specific format. However, our work focuses on a more general graph format, which can be applied to other real-world domains, such as social networks and computer vision. As a result, we observe that LLMs can predict patterns not only in scientific domains but also in other areas. We cited this paper in the related work section.

---

> > ### Author Response · Authors · 2024-11-24
> >
> > Dear Reviewer Dygb,
> >
> > Thank you for taking the time to review our paper.
> >
> > We hope our responses have addressed your concerns.
> >
> > Since it is approaching the end of the discussion period, if you have any further questions or feedback, please don’t hesitate to let us know!
> >
> > Best regards,
> >
> > Authors

---

> > > ### Author Response · Authors · 2024-11-26
> > >
> > > Dear Reviewer Dygb,
> > >
> > > Thank you for taking the time to review our paper.
> > >
> > > We hope our responses have addressed your concerns.
> > >
> > > Since it is approaching the end of the modifying, if you have any further questions or feedback, please don’t hesitate to let us know!
> > >
> > > Best regards,
> > >
> > > Authors

---

> > > > ### Comment · Reviewer_Dygb · 2024-11-27
> > > >
> > > > Thanks the authors for the detailed response. My concerns have been addressed and thus updated my score.

---

> > > > > ### Author Response · Authors · 2024-11-27
> > > > >
> > > > > Thank you for your valuable feedback! We’re glad our response addressed your concerns, and we will include the changes made to the rebuttal in our revision.

---

### Official Review · Reviewer_3Su5 · 2024-11-04

**Soundness:** 3
**Presentation:** 3
**Contribution:** 3
**Rating:** 6
**Confidence:** 4

**Summary:**

The paper uses 11 experiments to evaluate the abilities of 7 state-of-the-art (SOTA) LLMs to understand graph patterns from synthetic and real data, as well as their abilities to discover these patterns from data. The authors vary the description of these patterns (terminology-based vs topology-based) in order to observe the impact on performance. The authors claim that models may employ strategies different to strategies found in traditional algorithms to solve tasks, and that models tend to perform better when the input description of the pattern follows a terminology-based description rather than a topological-based description.

**Strengths:**

- The paper is well motivated relative to prior work. This work introduces several key graph pattern tasks, and effectively highlights the potential applications of these tasks.
- The experimentation done in the paper are very comprehensive. The authors evaluate a large set of SOTA LLMs on a large breadth of experiments, including both synthetic and real datasets. This breadth in experimentation effectively showcase the current abilities of SOTA LLMs on graph pattern understanding and discovery.

**Weaknesses:**

=== Lack of In-Context Learning ===
- All experiments in the paper are done in a zero-shot setting. Demonstrating the impact of in-context learning, such as CoT prompting, on a subset of the experiments would improve the contribution of the paper.

=== Lack of Clarity in Writing ===
- Section 3.3 lacks clarity on the underlying data. The authors mention that in this task, the LLM takes in an input graph, and is instructed to “detect specific primitive graph patterns” within the input graph. Does every input graph have a graph pattern inside it, or do some input graphs have no graph pattern?
- Following from the above point, in Section 4.1 the authors “reuse the graph datasets in the terminology-based pattern detection task” in order to test the models’ abilities to map between isomorphic graphs. If some of these graphs do not have a graph pattern, then isn’t this experiment testing graph isomorphic mapping and not pattern isomorphic mapping? What is the explicit relevance of this section with respect to graph patterns?
- Sections 5.2, 5.3, and 7.2 are poorly written. In Section 5.2, it is unclear what the accuracies in Table 7 represent and how they differ from the accuracies in Figure 5. It is unclear as to what either are reporting. In Section 5.3, the structure of the experiment is difficult to follow. For both Section 5.2’s and 5.3’s experiment, including an algorithmic description or pseudocode for the experiment would also greatly improve clarity. In Section 7.2 where the authors first discuss the strategies LLMs use, for each task, it would be effective to state what strategies traditional algorithms actually use, and then compare these to the strategies used by LLMs. Also, the second point in this first paragraph states that “the adjacency list is often better than the edge list in experiments”, but it is unclear as to what relevance this has on the central claim of this paragraph. When the authors discuss the impact of input format on performance, they mention that “terminology-based graph pattern detection is usually better than topology-based ones”, but then soon afterwards repeat themselves by mentioning that the “terminology-based description is often better than topology-based”. It is also unclear what this point, as well as the later point that “the adjacency list input is better than the edge list input”, have to do with the pretrained knowledge of the LLMs.

=== Unsubstantiated Claims===
- In Section 3.2, the author state that “the scale of the input graphs generally doesn’t have a major impact...because LLMs generally prioritize high-degree nodes and their neighbors to form the pattern. In larger graphs, LLMs tend to identify more regions for potential edits.” It would be helpful for the authors to empirically validate that the LLMs are in fact prioritizing high-degree nodes when forming the pattern, as this would provide evidence to substantiate this claim.

**Questions:**

- In Section 3.1, how can DIV be more than 0 if the temperature of all models is set to 0?
- In Section 3.3, does every input graph have a graph pattern inside it, or do some input graphs have no graph pattern?
- In Section 6, how does the “Both” description look like? How do the “alkane groups and fluoride groups” target patterns look like?

---

> ### Author Response · Authors · 2024-11-23
>
> > W1: All experiments in the paper are done in a zero-shot setting. Demonstrating the impact of in-context learning, such as CoT prompting, on a subset of the experiments would improve the contribution of the paper.
>
> Thanks for your valuable comments. We have conducted several experiments to illustrate the effect of Chain-of-Thought prompting on both terminology-based and topology-based pattern detection tasks using edge list descriptions. Specifically, we utilize 3 cases with the reasoning process as demonstrations to require LLMs to detect triangle and house patterns in small-scale graphs and triangle patterns in medium-scale graphs. The results are summarized in Table 1 and Table 2, highlighting the comparison between zero-shot and CoT settings.
>
> Table 1. The CoT results for terminology-based pattern detection
>
> |         | Zero-shot     |       |             | CoT      |       |             | Avg. Increase |
> |---------|----------|-------|-------------|----------|-------|-------------|-----------------|
> |         | triangle(S) | house(S) | triangle(M) | triangle(S) | house(S) | triangle(M) |                 |
> | Gemini  |     .725 |  .225 |        .218 |     .822 |  .103 |        .513 |            +.090 |
> | O1-mini |     .832 |  .066 |        .409 |     .811 |  .011 |        .727 |            +.081 |
>
> Table 2. The CoT results for topology-based pattern detection
>
> |        | Zero-shot |       |             | CoT|       |             | Avg. Increase |
> |--------|-----------|-------|-------------|----------|-------|-------------|-----------------|
> |        | triangle(S)  | house(S) | triangle(M) | triangle(S) | house(S) | triangle(M) |                 |
> |Gemini |      .651 |  .122 |        .484 |     .767 |  .263 |        .596 |            +.123 |
> |  O1-mini  |      .832 |  .000 |        .833 |     .736 |  .075 |        .756 |           -.033 |
>
> Overall, these results indicate that CoT prompting generally enhances pattern detection performance, particularly in terminology-based tasks. However, the effect of CoT is limited when the models already acheive high scores in the zero-shot setting. This aligns with previous studies that in-context learning does not always enhance the ability of LLMs to understand graph structures [1][2].
>
> Please note that CoT significantly increases the number of tokens, adding approximately 1,075 tokens per sample in the topology-based house detection process. Thus, considering the time and cost involved, we do not apply CoT in the original manuscript.
>
> We've included the analysis of CoT prompting in Appendix J.
>
> [1] Wang, Heng, et al. "Can language models solve graph problems in natural language?." Advances in Neural Information Processing Systems 36 (2024).
>
> [2] Fatemi, Bahare, Jonathan Halcrow, and Bryan Perozzi. "Talk like a graph: Encoding graphs for large language models." arXiv preprint arXiv:2310.04560 (2023).
>
> > W2 and Q3: Section 3.3 lacks clarity on the underlying data. The authors mention that in this task, the LLM takes in an input graph, and is instructed to “detect specific primitive graph patterns” within the input graph. Does every input graph have a graph pattern inside it, or do some input graphs have no graph pattern?
>
> We appreciate the opportunity to clarify this. We do not guarantee that every graph includes the target pattern. As a result, the input graphs range from containing no patterns to several target patterns. To ensure a fair evaluation, we use the F1 score to assess the patterns extracted by LLMs. If a graph contains multiple patterns, the F1 score reflects how well the extracted patterns match the ground truth.
>
> > W3: Following from the above point, in Section 4.1 the authors “reuse the graph datasets in the terminology-based pattern detection task” in order to test the models’ abilities to map between isomorphic graphs. If some of these graphs do not have a graph pattern, then isn’t this experiment testing graph isomorphic mapping and not pattern isomorphic mapping? What is the explicit relevance of this section with respect to graph patterns?
>
> Thank you for raising this point. The goal of Section 4.1 is to evaluate LLMs' ability to perform graph isomorphic mapping and demonstrate their consistency in recognizing identical graphs. While graph patterns can be considered as small subgraphs, this section is not very related to graph patterns. We appreciate your suggestions and update the title of Section 4.1 in the revised paper.

---

> ### Author Response · Authors · 2024-11-23
>
> > W4: Sections 5.2, 5.3, and 7.2 are poorly written.
>
> Thanks for your valuable suggestions. We've modified the paper to make them clear based on your advice.
>
> We have rewritten Section 5.2. The new version is: Mining frequent subgraphs is an important task on graphs, defined as finding subgraphs that appear frequently in a graph dataset given a frequency threshold. For each pattern, we first generate a graph dataset, ensuring that each graph contains the target pattern. The statistics of the datasets are provided in Table 12. In each turn, we randomly select 10 graphs from the dataset, task LLMs to extract frequent patterns based on these selected graphs, and output patterns in the topology-based description. We repeat this process 100 times and calculate the accuracy as the percentage of cases where the output pattern appears in more than 60\% of the selected graphs. It is worth noting that the extracted pattern does not need to match the target pattern precisely. For example, if the LLMs identify a triangle pattern during testing with a house pattern, we still consider this an accurate outcome. The accuracy and frequency of extracted patterns are summarized in Table 7 and Figure 5, respectively.
>
> Table 7 shows that LLMs can exhibit a strong capability in identifying frequent subgraphs, with GPT-4 and O1-mini showing impressive performance. However, LLMs are prone to detect simpler patterns rather than more complex ones. To further analyze the gap between the LLMs' outcome and target patterns, we include Figure 5. This figure aggregates responses from various LLMs on all datasets and illustrates the frequency of each pattern. We observe that triangles are the easiest patterns for LLMs to identify, while house patterns are significantly more challenging. Among the models tested, only Claude can identify some house patterns.
>
> For Sections 5.2 and 5.3, we have included the pseudo-code in Appendices F.1 and F.2, respectively. The pseudo-code for Section 5.2 is as follows:
>
> >**Input:** A graph dataset $G = ${$g_1, g_2, \dots, g_n$} , frequency threshold $f_{\mathrm{thres}}$
> >
> > **Output:** Frequent patterns and accuracy
> >
> > **For** iteration $i = 1$ to $100$
> >
> >   &nbsp;&nbsp;&nbsp;   Randomly select 10 graphs from $G$ to form a subset $G_p$
> >
> >   &nbsp;&nbsp;&nbsp;  Prompt LLMs to extract the set of frequent patterns $P$ based on $G_{p}$
> >
> >   &nbsp;&nbsp;&nbsp;  Initialize $\mathrm{Acc}_i$ = 0
> >
> > &nbsp;&nbsp;&nbsp;     **For** each pattern $p \in P$
> >
> > &nbsp;&nbsp;&nbsp;&nbsp;&nbsp;&nbsp;         **If** $p$ appears in more than $f_{\mathrm{thres}}$ of graphs in $G_p$
> >
> > &nbsp;&nbsp;&nbsp;  &nbsp;&nbsp;&nbsp;&nbsp;&nbsp;&nbsp;              Increment $\mathrm{Acc}_i$
> >
> >    &nbsp;&nbsp;&nbsp;   Compute  $\mathrm{Acc}_i$ = $\mathrm{Acc}_i$ / (# of patterns in $P$)
> >
> > Compute overall accuracy $\mathrm{Acc}$= ($\sum$ $\mathrm{Acc}_i$)/(\# of iterations)
> >
> >**Return:** Extracted frequent patterns and accuracy
>
> For Section 5.3, the pseudo-code is:
>
> > **Input:** Two graph dataset $G^1 = ${$g^1_1, g^1_2, \dots, g^1_{n_1}$} with label $L_1$ and
>  $G^2 =$ {$g^2_1, g^2_2, \dots, g^2_{n_2}$} with label $L_2$
> >
> > **Output:** Discriminative patterns and Metrics
> >
> > **Step 1: Pattern Extraction**
> >
> >**For** each iteration $i$
> >
> >   &nbsp;&nbsp;&nbsp;  Sample an equal number of graphs from $G^1$ and $G^2$ to form a balanced dataset $G_i$
> >
> >  &nbsp;&nbsp;&nbsp;  Prompt LLMs to identify discriminative patterns from $G_i$
> >
> >   &nbsp;&nbsp;&nbsp;  Add the extracted patterns into the set $P$
> >
> > **Step 2: Pattern Filtering**
> >
> >**For** each pattern $p \in P$
> >
> >&nbsp;&nbsp;&nbsp; Compute the occurrence of $p$ in $G^1$ and  $G^2$
> >
> >  &nbsp;&nbsp;&nbsp;  **If** ($\mathrm{occurrence}(p, G^1) \geq 90\%$ and $\mathrm{occurrence}(p, G^2) < 10\%$) or
>           ($\mathrm{occurrence}(p, G^2) \geq 90\%$ and $\mathrm{occurrence}(p, G^1) < 10\%$)
> >
> >  &nbsp;&nbsp;&nbsp;&nbsp;&nbsp;&nbsp;   Retain $p$ as a discriminative pattern
> >
> > Obtain final discriminative pattern set $P_{\mathrm{final}}$
> >
> > **Step 3: D.P. Computation**
> >
> > Compute the discriminative pattern ratio as:
> >
> > $\mathrm{D.P.}$ = (\# of Discriminative patterns in $P_{\mathrm{final}}$)/(\# of Extracted patterns in $P$)
> >
> > **Step 4: Classification Accuracy Computation**
> >
> >**For** each new graph $g$ in the test set $G_{\mathrm{test}}$
> >
> >&nbsp;&nbsp;&nbsp;     Prompt LLMs to predict the label of $g$ based on $P_{\mathrm{final}}$
> >
> > Compute the prediction accuracy $\mathrm{Acc}$ as the proportion of correctly predicted labels
> >
> > **Return:** $P_{\mathrm{final}}$, $\mathrm{Acc}$, and $\mathrm{D.P.}$

---

> ### Author Response · Authors · 2024-11-23
>
> >W4: Sections 5.2, 5.3, and 7.2 are poorly written. (Continued)
>
> For section 7.2, we reorganized it as:
>
> **LLMs use diverse algorithms for one task, and the performance varies due to their execution ability:** We provide two observations: (1) We manually reviewed most of the outputs generated by LLMs in graph mining tasks, and summarized the algorithms used by LLMs in Appendix H. Our analysis reveals that different LLMs utilize diverse algorithms to solve the same problem. For instance, more than eight algorithms are used for pattern detection tasks (Section 3.3). (2) Due to the internal flaws of LLMs, these algorithms, although logically correct, will have different performance. In the graph isomorphic mapping task (Section 4), a common algorithm starts by counting node degrees and then mapping nodes. O1-mini uses this approach for 89% of the data but achieves only 30% accuracy due to errors in degree counting. In contrast, Claude applies degree counting to only 23% of the data, relying primarily on a direct edge-matching algorithm for the rest. This alternative strategy enables Claude to achieve an impressive 96% accuracy.
>
> **Input format that aligns with the pretrained knowledge improves the performance:** First, LLMs are pre-trained on extensive internet datasets where graph patterns are often described using specific terminologies. This exposure helps LLMs understand these terms. Comparing the results in Section 3.3 and Section 4.3, we observe that terminology-based graph pattern detection generally outperforms topology-based detection. This suggests that LLMs leverage their internal knowledge to enhance performance when provided with terminology as input. Second, the pretrained knowledge will influence the strategies employed by LLMs, and the graph input format that aligns with the strategies will improve the performance. For example, in the case of discriminative pattern learning (Section 5.3), the algorithms used by LLMs often rely on comparing corresponding edges in two graphs. In this scenario, the edge list format typically leads to better performance than the adjacency list format. Conversely, in k-core detection (Section 5.1), the algorithms require counting node degrees and the edge list is inferior to the adjacency list.
>
> > W5: In Section 3.2, the author state that “the scale of the input graphs generally doesn’t have a major impact...because LLMs generally prioritize high-degree nodes and their neighbors to form the pattern...” It would be helpful for the authors to empirically validate that the LLMs are in fact prioritizing high-degree nodes when forming the pattern, as this would provide evidence to substantiate this claim.
>
> We appreciate the opportunity to clarify this question. We manually reviewed 10% of LLM responses for the graph modification task. Table 1 revealed that most responses follow a two-step strategy: selecting a subset of nodes matching the target pattern size and modifying it to align with the pattern.
>
> Table 1. The percentage of potential algorithms used by LLMs in graph modification tasks.
> | Algorithm                                                                                                          | Llama   | Gemini | Mixstral | GPT-4  | GPT-4o | Claude | O1-mini |
> |--------------------------------------------------------------------------------------------------------------------|---------|--------|----------|--------|--------|--------|---------|
> | Select a set of nodes, and then modify this subset to align with the target pattern                                         | 100% | 85% |   80% | 75% | 95% | 80% | 100% |
> | Special algorithm on house patterns: Identify a triangle, then modify a square based on the triangle  |   0% | 10% |  20% |  0% |  5% | 20% |   0% |
> | Special algorithm on house patterns: Identify a square, then modify a triangle based on the square  |   0% |  0% |   0% | 25% |  0% |  0% |   0% |
> | Assume the graph already meets the requirement and avoid modifications                                |   0% | 5% |   0% |  0% |  0% |  0% |   0% |
>
> Second, we calculated the average degree of the nodes selected by LLMs and summarized this information below:
>
> | Scale  | AVG. degree | Llama | Gemini | Mixtral | GPT-4 | GPT-4o | Claude | O1-mini |
> |--------|-------------|-------|--------|---------|-------|--------|--------|---------|
> | Small  |        3.32 |  3.41 |   2.60 |    2.64 |  3.66 |   3.61 |   3.75 |    3.65 |
> | Medium |        2.15 |  2.30 |   2.98 |    2.69 |  2.39 |   2.78 |   2.95 |    2.95 |
> | Large  |        2.36 |  2.80 |   2.89 |    3.10 |  2.38 |   3.03 |   3.39 |    3.15 |
>
> We find that the nodes selected by LLMs consistently have higher degrees than the average node degree of the graph, particularly in Medium and Large scales. This suggests that LLMs are more likely to select higher-degree nodes for editing.
>
> We've added the analysis in Appendix I.2.

---

> ### Author Response · Authors · 2024-11-23
>
> > Q1:In Section 3.1, how can DIV be more than 0 if the temperature of all models is set to 0?
>
> Thank you for your thoughtful question. LLMs do not always produce consistent results, even when the temperature is set to 0. This is because most LLMs operate in parallel systems, which can involve probabilistic random sampling on GPUs, unpredictable execution order of subsystems, and variations in floating-point arithmetic implementations. Several papers, such as [1][2], examine the impact of GPU-related factors, while [3] investigates the non-deterministic behavior of LLMs, even at a temperature of 0 with a fixed random seed.
>
> [1] Pham H V, Qian S, Wang J, et al. Problems and opportunities in training deep learning software systems: An analysis of variance[C]//Proceedings of the 35th IEEE/ACM international conference on automated software engineering. 2020: 771-783.
>
> [2] Hagmann M, Meier P, Riezler S. Towards Inferential Reproducibility of Machine Learning Research[C]//The Eleventh International Conference on Learning Representations 2023.
>
> [3] Blackwell R E, Barry J, Cohn A G. Towards Reproducible LLM Evaluation: Quantifying Uncertainty in LLM Benchmark Scores[J]. arXiv preprint arXiv:2410.03492, 2024.
>
> > Q3: In Section 6, how does the “Both” description look like? How do the “alkane groups and fluoride groups” target patterns look like?
>
> We appreatiate to have the opportunity to clearify this. In our experiments, we use Benzene, Alkane, and Fluoride as examples of functional groups.
>
> For each functional group, we provide two distinct methods for pattern description: terminology-based and topology-based, as summarized in the table below.
>
>
>
> |            | Terminology | Topology    |
> |------------|:-----------:|:-----------:|
> | Benzene (Cn)   | benzene ring | (Node 0 Atom C, Node 1 Atom C), (Node 1 Atom C, Node 2 Atom C), (Node 2 Atom C, Node 3 Atom C), (Node 3 Atom C, Node 4 Atom C), (Node 4 Atom C, Node 5 Atom C) |
> | Alkane (C2nH2n+2)     | Alkane Carbonyl which contains an unbranched alkane and a carbonyl functional group | (Node 0 Atom C, Node 1 Atom H), (Node 0 Atom C, Node 2 Atom H), (Node 0 Atom C, Node 3 Atom H), (Node 0 Atom C, Node 4 Atom H) |
> | Fluoride (COF2)   | Fluoride Carbonyl which contains a fluoride and a carbonyl functional group | (Node 0 Atom C, Node 1 Atom O), (Node 0 Atom C, Node 2 Atom F), (Node 0 Atom C, Node 3 Atom F) |
>
> To enhance LLM understanding, we use "both" to combine these two descriptions. The detailed prompt is:
>
> *In the context of molecular biology, you have been provided with a pattern motif to compare against a test molecule graph. The pattern is a [Terminology-based description], which also can be represented as [Topology-based description]. ... [Test-Molecular] ... Now, please determine whether the pattern motif exists in the molecule graph by selecting either "The pattern does exist" or "The pattern does not exist".*
>
> We've added that information in Appendix. C.4.

---

> > ### Author Response · Authors · 2024-11-24
> >
> > Dear Reviewer 3Su5,
> >
> > Thank you for taking the time to review our paper.
> >
> > We hope our responses have addressed your concerns.
> >
> > Since it is approaching the end of the discussion period, if you have any further questions or feedback, please don’t hesitate to let us know!
> >
> >
> > Best regards,
> >
> > Authors

---

> > > ### Author Response · Authors · 2024-11-26
> > >
> > > Dear Reviewer 3Su5,
> > >
> > > Thank you for taking the time to review our paper.
> > >
> > > We hope our responses have addressed your concerns.
> > >
> > > Since it is approaching the end of the modifying, if you have any further questions or feedback, please don’t hesitate to let us know!
> > >
> > > Best regards,
> > >
> > > Authors

---

> ### Comment · Reviewer_3Su5 · 2024-11-27
> **Responding to Official Comments**
>
> I thank the authors for their detailed responses and additions to the paper. I have a few follow up questions:
>
> 1. I am still confused as to the overall relevance of Section 4.1 to the paper, and I am curious as to why the authors didn't remove this section from the paper entirely. This would be helpful to clarify, otherwise I would suggest removing this section from the final version of the paper.
>
> 2. The evaluation of CoT prompting is a great addition, and it seems like a promising in-context method for pattern mining. However, it would be helpful to know what the <input, chain of thought, output> prompt was (as defined in [1]), and how many examples were used.
>
> 3. Appendix Section H is also very strong, thanks for adding it! You mention that the models prefer "selecting a subset of nodes matching the target pattern size and modifying it to align with the pattern." Can you offer any insights into if the models arbitrarily select *any* subset of nodes matching the target pattern size, or if the models are approaching this selection in another way?
>
> In all, I believe that the paper is in a stronger position as a result, and have increased my scores.
>
> [1] Wei, Jason, et al. "Chain-of-thought prompting elicits reasoning in large language models." Advances in neural information processing systems 35 (2022): 24824-24837.

---

> ### Author Response · Authors · 2024-11-27
>
> Thank you for your valuable feedback! We’re glad that our previous response addressed most of your concerns. We apologize for any unclear points in our earlier reply and sincerely appreciate your patience. Below, we provide a new response to your further questions.
>
> >I am still confused as to the overall relevance of Section 4.1 to the paper, and I am curious as to why the authors didn't remove this section from the paper entirely. This would be helpful to clarify, otherwise I would suggest removing this section from the final version of the paper.
>
> Thanks for your thoughtful suggestion! The overall goal of the paper is to progressively evaluate and challenge LLMs’ abilities in handling graph pattern tasks.
>
> In Section 4, we investigate LLMs' recognition ability for graph patterns using topology-based descriptions. A key property of such description is permutation invariance, which requires recognizing that different representations of the same graph convey identical information. Thus, in Section 4.1, we conduct experiments using isomorphic mappings of graphs, randomly including zero to multiple specific graph patterns. This test provides valuable insights into the broader recognition capabilities of LLMs when working with topology-based descriptions. We then revisit their ability to align with human instructions for modifying and detecting graph patterns using topology-based descriptions in Sections 4.2 and 4.3.
>
>
> We hope this explanation clarifies the relevance and importance of Section 4.1. We appreciate your suggestion, and we will consider revisiting the section's presentation to ensure its importance is communicated more clearly. Thank you again for your valuable feedback!
>
> >The evaluation of CoT prompting is a great addition, and it seems like a promising in-context method for pattern mining. However, it would be helpful to know what the <input, chain of thought, output> prompt was (as defined in [1]), and how many examples were used.
>
>
>
> Thank you for your valuable suggestions! We use three examples with detailed reasoning processes as demonstrations in the CoT prompt.
>
> The input format is structured as <Example 1 with its reasoning process and answer, Example 2 with its reasoning process and answer, Example 3 with its reasoning process and answer, Question>.
>
> For instance, an example with its reasoning process and answer in a topology-based house detection task looks like this:
>
>
> >>Q: Identify the occurrence patterns of the given motif in the graph.
> >>
> >>The pattern is:
> >>G describes an undirected graph among node A, B, C, D, and E.
> >>Node A is connected to Node B.
> >>Node A is connected to Node D.
> >>Node B is connected to Node C.
> >>Node C is connected to Node D.
> >>Node C is connected to Node E.
> >>Node D is connected to Node E.
> >>
> >>The graph is:
> >>G describes an undirected graph among node 0, 1, 2, 3, 4, and 5.
> >>Node 0 is connected to Node 4.
> >>Node 0 is connected to Node 5.
> >>Node 0 is connected to Node 1.
> >>Node 1 is connected to Node 4.
> >>Node 2 is connected to Node 4.
> >>Node 2 is connected to Node 3.
> >>Node 3 is connected to Node 5.
> >>Node 4 is connected to Node 5.
> >>
> >>Please identify the patterns for each node and list all of them as follows: The detected patterns are: [(#1, #2, #3, #4, #5), …].
> >>Let's think step by step.
> >>
> >>A: To identify the occurrence patterns of the given motif in the graph, we need to find all subgraphs that match the structure described in the motif.
> >>
> >>The motif structure is as follows:
> >> - Node A is connected to Node B.
> >> - Node A is connected to Node D.
> >> - Node B is connected to Node C.
> >> - Node C is connected to Node D.
> >> - Node C is connected to Node E.
> >> - Node D is connected to Node E.
> >>
> >>We will look for all sets of 5 nodes in the given graph that match this structure.
> >>
> >>We check all of the subsets of 5 nodes.
> >>The subsets are 0, 1, 2, 3, 4; 0, 2, 3, 4, 5; 0, 1, 3, 4, 5; 0, 1, 2, 4, 5; 0, 1, 2, 3, 5; 1, 2, 3, 4, 5.
> >>
> >>Check subset 0, 1, 2, 3, 4:
> >>They do not match A, B, C, D, E
> >>
> >>Check subset 0, 2, 3, 4, 5:
> >>Node 0 is connected to Node 4, and 5 in the set. The possible matching for C, D, E is (0, 4, 5)
> >>Node 3 is connected to Node 2 and 5 in the set.
> >>Node 2 is connected to Node 3 and 4 in the set.
> >>Therefore, the possible matching is E: 0, C: 4, D: 5; A: 2, B: 3
> >>
> >>Check subset 0, 1, 3, 4, 5:
> >>They do not match A, B, C, D, E
> >>
> >>Check subset 0, 1, 2, 4, 5:
> >>They do not match A, B, C, D, E
> >>
> >>Check subset 0, 1, 2, 3, 5:
> >>They do not match A, B, C, D, E
> >>
> >>Check subset 1, 2, 3, 4, 5:
> >>They do not match A, B, C, D, E
> >>
> >>So the final answer is [(0, 2, 3, 4, 5)].

---

> ### Author Response · Authors · 2024-11-27
>
> >Appendix Section H is also very strong, thanks for adding it! You mention that the models prefer "selecting a subset of nodes matching the target pattern size and modifying it to align with the pattern." Can you offer any insights into if the models arbitrarily select any subset of nodes matching the target pattern size, or if the models are approaching this selection in another way?
>
> Thank you for your valuable insights! Based on the responses generated by LLMs, we can identify the exact nodes they select. However, the LLMs directly provide the selected nodes without explaining the reasoning or the selection process. Therefore, in addition to investigating the influence of node degrees, we also explore whether node IDs have any impact on the selection process.
>
> We computed the average node ID values in the graph datasets, and compared them with the average IDs of nodes selected by each LLM during the graph modification task (transforming a diamond to a square). The results are presented in the table below:
>
> | Scale  | AVG. Node idx | Llama | Gemini | Mixstral | GPT-4 | GPT-4o | Claude | O1-mini |
> |--------|---------------|-------|--------|----------|-------|--------|--------|---------|
> | Small  |          4.29 |  2.43 |   1.36 |     2.54 |  3.47 |   2.97 |   3.09 |    2.99 |
> | Medium |          9.07 |  4.48 |   6.27 |     8.52 |  7.63 |   4.92 |   7.03 |    6.00 |
> | Large  |         14.00 |  6.30 |   8.34 |    12.50 | 10.95 |   6.45 |  10.68 |    9.75 |
>
>
> From these results, we observe that node IDs appear to influence the subset selection process. The LLMs, particularly Llama, Gemini and GPT-4o, tend to select nodes that are mentioned earlier (i.e., nodes with smaller IDs) and have higher degrees in the graph description.
>
> Thanks again for your valuable feedback!

---

### Official Review · Reviewer_FSEL · 2024-11-04

**Soundness:** 4
**Presentation:** 2
**Contribution:** 4
**Rating:** 8
**Confidence:** 4

**Summary:**

This paper presents a benchmark that evaluates SOTA LLM graph pattern understanding and whether any graph reasoning is gleaned from pretraining in graph-based tasks. The authors describe three distinct settings for evaluating graph pattern understanding: terminology-based, topology-based, and data-driven.

The terminology-based evaluation explores whether LLMs can comprehend and reproduce graph patterns from the terminology found in pretraining data. Models are tested by examining their alignment with human understanding of a given pattern and assessing whether LLMs can follow human instructions for pattern-detection.

The topology-based evaluation assesses if LLMs are consistent in their ability to recognize identical patterns in different permutations. Models are evaluated on their ability to perform pattern mapping through isomorphic identification, graph editing, and extracting topology-based patterns.

The last evaluation strategy evaluates LLMs’ ability to independently identify and mine graph patterns within real-world datasets.
The paper evaluates 7 SOTA LLMs on their ability to understand 5 undirected graph patterns and 4 directed graph patterns. The results also include model performances on prompts using adjacency lists and edge lists, which are both popular formats for representing graph in LLM-Graph reasoning.

**Strengths:**

While the novelty of the overall goal is limited, this work offers the LLM-Graph reasoning community valuable, and seemingly reliable insight into how LLMs are able to manipulate and understand graph patterns. The paper also stands out because of its clear experimental design and extensive set of experimental results, which would yield insight into the growing research community focusing on LLMs applied to graph reasoning.

**Weaknesses:**

While the paper is generally well-organized and well-written, the paper suffers from a lack of space. It becomes difficult to parse, given the breadth of experimental results, some of which are not followed by fully satisfactory analyses. There are a total of 9 tasks, each with their own table or figure. Furthermore, several tables include both adjacency list and edge list results, which makes tables very difficult to read. I would suggest splitting the results into separate tables or even moving the lesser results to the appendix, as the impact prompt format is not a central result.

The analysis of each set of experiments is often quite short and focuses mainly on the performance of o1 or generalizes to all LLMs. While it is important to discuss the best performing model, the paper offers little insights into the types of mistakes being made by underperforming models. For instance, section 4.2 simply states that the decreased average performance (as compared to the terminology-based results discussed in 3.2) were “likely due to increased hallucinations”. Is this backed by the experimental results?

Section 4.1 provides an example of the type of analysis the other results analysis sections would benefit from. I also think the paper would benefit from providing a bird’s-eye view of how each model performs across all tasks. This would provide insight into the relative strengths and weaknesses of each model, which are currently difficult to glean.

Minor Issues:
- The paper mentions that LLMs tend to add extra edges to patterns such as T-triangle and V-S, leading to unintended structures. It would be helpful to clarify whether these extra edges result in completely disconnected structures or simply unintended modifications.

- The paper asserts that the adjacency list format is better suited for LLMs, which does seem to be the case for o1; however, the results for other models (e.g., pattern isomorphic mapping) do not seem to be as conclusive. A short analysis of these results would be helpful to the community when deciding which format to use for a given model in future experiments.

- The exclusion of 'large' graph sizes results in many figures limits the reliability of those figures.

**Questions:**

- Has analysis of the errors of each model been done? If so, is it possible to include these analyses in the appendix?
- Could you clarify whether the extra edges added by LLMs in the T-triangle and V-S patterns result in completely disconnected structures or merely unintended modifications?
- What was the reasoning behind not analyzing the impact of parameter size on model performance? This information could provide valuable insights, especially given that certain models outperformed others.
- Could you expand on the analysis of the input prompt formats?
- Can you separate the EL and AL results into separate tables in the final version of the paper? If space constraints prevent both from being included in the main text, I suggest moving the weaker result to the appendix with the results from the above question.
- Was the exclusion of 'large' graphs in many figures done for the purpose of interpretability? If so, please include versions with the full or just the large results in the appendix.

---

> ### Author Response · Authors · 2024-11-23
>
> > Q1. The tables are hard to read due to the mixture of adjacency list and edge list results. (from Weakness 1 and Question 5)
>
> Thank you for your valuable advice! Following your suggestions, we have separated all the tables to improve the paper's readability in Appendix K.
>
> > Q2. The experiment analysis is brief, primarily focusing on O1, with limited analysis of other underperforming LLMs. (from Weakness 2 and Question 1)
>
> Thank you for highlighting this point, and we sincerely appreciate the opportunity to provide further clarification. To analyze the performance of other LLMs, we manually reviewed 10% of the responses from all LLMs and included an analysis section in Appendix H. Specifically, we used terminology-based and topology-based pattern detection tasks as examples to examine the potential algorithms used by LLMs.
>
> Table 1. The percentage of potential algorithms used by LLMs to solve the terminology-based pattern detection task
>
> | Algorithms                                                                                                                        | Llama  | Gemini | Mixstral | GPT-4  | GPT-4o | Claude | O1-mini |
> |-----------------------------------------------------------------------------------------------------------------------------------|--------|--------|----------|--------|--------|--------|---------|
> | Directly give an answer                                                                                                           |  0.00% | 34.00% |   24.00% |  4.00% |  0.00% | 24.00% |  18.00% |
> |  Use external tools (e.g. NetworkX)                                                                                                 |  6.00% |  0.00% |    4.00% |  0.00% |  6.00% |  0.00% |  20.00% |
> | Draw a figure of the graph and give an answer                                                                                         |  0.00% |  4.00% |    0.00% |  0.00% |  4.00% |  0.00% |   0.00% |
> | Traverse every node, and check whether this node and its neighbors can form the  pattern                                    |  0.00% |  4.00% |   24.00% | 24.00% | 38.00% | 12.00% |   4.00% |
> | Generate all possible node combinations and verify one by one    | 54.00% | 12.00% |   20.00% | 16.00% | 38.00% | 60.00% |  28.00% |
> | Traverse all possible edge combinations, and verify if they form the pattern                        | 40.00% |  8.00% |   14.00% | 16.00% |  0.00% |  4.00% |  18.00% |
> | A special algorithm on house pattern:  Identify triangles as the roof first and check if the triangle has a square as its base. |  0.00% |  0.00% |    8.00% |  2.00% | 12.00% |  0.00% |   2.00% |
> | A special algorithm on house pattern: Identify squares as the base first and check if the square has a triangle as its roof.      |  0.00% |  2.00% |    2.00% |  0.00% |  0.00% |  0.00% |  10.00% |
> | Only give the process but no answers                                                                                                |  0.00% | 36.00% |    4.00% | 38.00% |  2.00% |  0.00% |   0.00% |
>
>
> Table 2. The percentage of potential algorithms used by LLMs in the topology-based pattern detection task
>
> | Algorithms                                                                                                      | Llama  | Gemini | Mixstral | GPT-4  | GPT-4o | Claude | O1-mini |
> |-----------------------------------------------------------------------------------------------------------------|--------|--------|----------|--------|--------|--------|---------|
> | Directly give an answer                                                                                         |  0.00% | 43.33% |   10.00% |  0.00% |  0.00% | 53.33% |  10.00% |
> | Using external tools (e.g. NetworkX)                                                                              |  0.00% |  0.00% |    0.00% |  0.00% |  0.00% |  0.00% |   3.33% |
> | Traverse every node, and check whether this node and its neighbors can form the target pattern                  |  0.00% | 36.67% |   16.67% | 33.33% | 36.67% | 36.67% |  36.67% |
> | Generate all combinations with the specified number of nodes and select those that meet the pattern definition. | 66.67% | 16.67% |   33.33% | 33.33% | 43.33% | 10.00% |  50.00% |
> | Traverse all edges, and determine if they form a pattern based on their common nodes.                           | 33.33% |  3.33% |   40.00% | 33.33% |  0.00% |  0.00% |   0.00% |
> | Only give the process but no answers                                                                              |  0.00% |  0.00% |    0.00% |  0.00% | 20.00% |  0.00% |   0.00% |

---

> ### Author Response · Authors · 2024-11-23
>
> The tables show that: (1) LLMs often provide a solution without actually executing the algorithm. This leads to failures, such as Gemini and GPT-4 in terminology-based pattern detection and GPT-4o in topology-based pattern detection.
> (2) In terminology-based pattern detection tasks, LLMs are more flexible to utilize different algorithms. For instance, LLMs can decompose a house pattern into separate triangle and square detections, transferring the problem into simpler tasks.
> (3) We observe that most LLMs prefer to list all possible combinations first and then check whether they match the target pattern. However, their accuracy varies significantly. To explore underlying failure reasons, we further calculate the precision and recall of detected patterns. These two metrics provide insight into the type of hallucinations that occur when LLMs perform pattern detection. A low precision suggests that LLMs hallucinate extra edges in the extracted patterns, whereas a low recall indicates that some edges in the input graph were overlooked by LLMs.
>
> As shown in the following table, we find that LLMs achieve higher precision than recall. This indicates that the most errors come from the overlooked edges. Furthermore, most LLMs show performance drops when transitioning from terminology-based to topology-based detection. The terminology helps reduce the hallucination.
>
> |                   |           | Llama  | Gemini | Mixstral | GPT-4  | GPT-4o | Claude | O1-mini |
> |-------------------|-----------|--------|--------|----------|--------|--------|--------|---------|
> | Terminology-based | Precision |  0.599 |  0.622 |    0.543 |  0.449 |  0.609 |  0.763 |   0.776 |
> |                   | Recall    |  0.294 |  0.493 |    0.224 |  0.039 |  0.256 |  0.304 |   0.416 |
> | Topology-based    | Precision |  0.190 |  0.484 |    0.507 |  0.409 |  0.586 |  0.764 |   0.765 |
> |                   | Recall    |  0.052 |  0.308 |    0.142 |  0.050 |  0.195 |  0.249 |   0.387 |
> | Decrease          | Precision | -0.409 | -0.138 |   -0.036 | -0.039 | -0.023 |  0.001 |  -0.011 |
> |                   | Recall    | -0.242 | -0.185 |   -0.082 |  0.012 | -0.062 | -0.056 |  -0.029 |
>
> In conclusion, we analyzed the potential algorithms employed by LLMs and identified the underlying reasons for their failures. The complete analysis is included in Appendix I.2.

---

> ### Author Response · Authors · 2024-11-23
>
> >Q3. Provide a bird’s-eye view of how each model performs across all tasks. (from Weakness 3)
>
> Thank you for your valuable feedback on including a bird's-eye view of model performance. We have incorporated this table in Appendix G.
>
> For each LLM, we select the best performance from either edge list or adjacency list graph descriptions and then calculate the models' average scores across small, medium, and large-scale datasets. Furthermore, we average the scores across different graph patterns. Finally, we rank the models for each task and provide an overall ranking.
>
> In the table, O1-mini achieves an average rank of 2.1, outperforming other models in most cases while still facing challenges in isomorphic mapping and discriminative pattern learning tasks. Interestingly, GPT-4o demonstrates balanced performance across all tasks. Overall, we recommend using O1-mini, GPT-4o, and Claude for solving graph pattern tasks.
>
> |         | terminology-based patterns |                    |                   | topology-based patterns |                    |                   | data-driven patterns |                              |                                 | AVG. rank |
> |---------|----------------------------|--------------------|-------------------|-------------------------|--------------------|-------------------|----------------------|------------------------------|---------------------------------|-----------|
> |         | pattern translation        | graph modification | pattern detection | isomophic mapping       | graph modification | pattern detection | k-core               | frequent subgraph extraction | discriminative pattern learning |           |
> | GPT-4   |                          3 |                  6 |                 7 |                       6 |                  5 |                 5 |                    7 |                            1 |                               4 |       4.9 |
> | GPT-4o  |                          2 |                  3 |                 2 |                       2 |                  2 |                 3 |                    1 |                            4 |                               1 |       2.2 |
> | Mixtral |                          7 |                  4 |                 6 |                       4 |                  3 |                 7 |                    5 |                            1 |                               7 |       4.9 |
> | Llama   |                          5 |                  2 |                 5 |                       3 |                  4 |                 6 |                    4 |                            5 |                               5 |       4.3 |
> | Gemini  |                          4 |                  7 |                 3 |                       7 |                  7 |                 4 |                    6 |                            6 |                               3 |       5.2 |
> | Claude  |                          6 |                  5 |                 4 |                       1 |                  6 |                 2 |                    2 |                            7 |                               1 |       3.8 |
> | O1-mini |                          1 |                  1 |                 1 |                       5 |                  1 |                 1 |                    2 |                            1 |                               6 |       2.1 |

---

> ### Author Response · Authors · 2024-11-23
>
> > Q4. Disconnected structures or merely unintended modifications in T-triangle and V-S? (from Issue 1 and Question 2)
>
> Thanks for your question! After manually reviewing each generated structure, we did not find any disconnected structures. LLMs generally follow a logic of first creating the target pattern and then adding extra edges to expand it into a larger graph. To illustrate this process, we analyze output from GPT-4 as an example of how LLMs perform graph translation in tailed-triangle generation:
>
>
> >> Here's how the graph can be formulated:
> >>
> >> - **Tailed Triangle Edges**: (1, 2), (2, 3), (3, 1), (1, 4) - This forms the tailed triangle.
> >> - **Connecting Remaining Nodes**: To ensure each of the remaining nodes (5 to 20) has at least one edge and to avoid forming any additional tailed triangles, we can simply connect each new node to the next one in sequence. This will form a chain that can be connected back to one of the nodes in the tailed triangle to ensure the graph is undirected and connected.
> >>
> >> Here's the complete edge list:
> >>
> >> 1. (1, 2)
> >> 2. (2, 3)
> >> 3. (3, 1)
> >> 4. (1, 4)
> >> 5. (4, 5)
> >> 6. (5, 6)
> >> 7. (6, 7)
> >> 8. (7, 8)
> >> 9. (8, 9)
> >> 10. (9, 10)
> >> 11. (10, 11)
> >> 12. (11, 12)
> >> 13. (12, 13)
> >> 14. (13, 14)
> >> 15. (14, 15)
> >> 16. (15, 16)
> >> 17. (16, 17)
> >> 18. (17, 18)
> >> 19. (18, 19)
> >> 20. (19, 20)
> >> 21. (20, 3) - This last edge connects the chain back to the triangle, ensuring the graph is undirected and connected.
>
> The connection (20, 3) is an unintended modification, as the graph is already connected without it. Another example is from Gemini while generating the VS pattern. The example is as follows:
>
> >>Here's a directed graph with 20 nodes and a single V-structure, along with the edge list representation:
> >>
> >>**V-Structure:**
> >>
> >>* Node 1 -> Node 20
> >>* Node 2 -> Node 20
> >>
> >>**Remaining Graph Structure (ensuring each node has at least one edge):**
> >>
> >>We'll create a cycle involving the remaining nodes and then add a few extra edges for variety:
> >>
> >>* 3 -> 4 -> 5 -> 6 -> 7 -> 8 -> 9 -> 10 -> 11 -> 12 -> 13 -> 14 -> 15 -> 16 -> 17 -> 18 -> 19 -> 3 (This forms the cycle)
> >>
> >>*  1 -> 5 (Additional edge)
> >>*  10 -> 18 (Additional edge)
>
> Nodes 1, 2, and 20 already form a VS structure, but Gemini creates a cycle and adds two additional edges to connect all the nodes in the graph. This results in the formation of another VS structure involving nodes (1, 4, 5).
>
> We've added the analysis in Appendix I.1.
>
> > Q5. How do the adjacency list and edge list format impact the performance of LLMs across tasks? (from Issue 2 and Question 4)
>
> Thank you for your valuable suggestion. The choice between the adjacency list and edge list format depends on the task and the algorithm that LLMs use to solve the task. For example, in discriminative pattern learning, the algorithms used by LLMs often rely on edge combinations, making edge comparisons highly relevant. In this scenario, the adjacency list achieves only a 0.9 score with Gemini, while the edge list allows GPT-4o and Claude to reach a perfect score of 1.0. Conversely, when tasks require consideration of node information, such as in k-core detection, the adjacency list tends to perform better than the edge list. For instance, in the medium dataset, GPT-4o achieves the highest accuracy of 1.0 using the adjacency list, while Claude and O1-mini achieve only 0.88 with the edge list.
>
> Furthermore, we believe that using terminology-based prompts enables LLMs to achieve better performance. Our findings show that terminology-based prompts consistently outperform topology-based ones in both pattern detection and modification tasks. This aligns with real-world experiments, where accuracy in Benzene detection improves from 0.78 with topology-based prompts to 0.9 with terminology-based prompts.
>
> In the revised paper, we have updated the discussion in Sec. 7.2 to clarify this matter.
>
> > Q6. 'Large' graph sizes results. (From Issue 3 and Question 6)
>
> Thank you for your comments. In the original paper, we excluded the results for large graph sizes in Fig. 3 and 4 to maintain the figure clarity. Additionally, most scores for large graph sizes in T-triangle/Square/Diamond/House patterns are close to 0, making them difficult to present effectively in the figures. However, the full results can be found in Tables 18/19 and Figure 7/8 in Appendix E, or the split Tables in 28/29/32/32.
>
> > Q7. model parameters (From Question 3)
>
> Thank you for your valuable advice. While close-sourced models do not release their parameters, it is hard for us to explore the impact of parameter sizes.
>
> | Model | Gemini  | Mixtral | Llama | Claude     | GPT-4   | GPT-4o  | O1-mini    |
> |-------|---------|---------|-------|------------|---------|---------|------------|
> | Size  | No release     | 176B    | 405B  | No release | No release | No release | No release |

---

> > ### Author Response · Authors · 2024-11-24
> >
> > Dear Reviewer FSEL,
> >
> > Thank you for taking the time to review our paper.
> >
> > We hope our responses have addressed your concerns.
> >
> > Since it is approaching the end of the discussion period, if you have any further questions or feedback, please don’t hesitate to let us know!
> >
> >
> > Best regards,
> >
> > Authors

---

> > > ### Comment · Reviewer_FSEL · 2024-11-26
> > > **Rebuttal response**
> > >
> > > I appreciate your response, and the additions have satisfied my concerns about the original paper. I have raised my score.

---

> > > > ### Author Response · Authors · 2024-11-26
> > > >
> > > > Thank you for your valuable feedback! We’re glad our response addressed your concerns, and we will include the changes made to the rebuttal in our revision.

---

### Comment · Area_Chair_wAuN · 2024-11-23
**Please initiate discussion on the rebuttal**

Dear Reviewers,

The authors have posted a rebuttal to the concerns raised. Since the discussion phase closes in a few days, I would request you to study their rebuttal and check if it changes your opinion on the work. The discussion phase is a critical component of the ICLR review process and your active engagement would be much appreciated.

best,

AC

---

### Meta-Review · Area_Chair_wAuN · 2024-12-06

**Metareview:**

The paper examines the ability of LLMs to understand graph patterns from synthetic and real data, and their capacity to discover these patterns from data. The authors vary the description of these patterns based on terminology and topology to observe the impact on performance.

The authors claim that models may employ strategies different from traditional algorithms to solve tasks and that models tend to perform better when the input description of the pattern follows a terminology-based description rather than a topological-based description. Some reviewers felt this is an obvious observation.

In their initial reviewes, the reviewers expressed concerns about novelty with respect to existing studies on LLM-understanding of graph topologies and associated queries. The initial version also omitted the impact of COT reasoning, which the authors addressed in the rebuttal. The authors also enhanced the differentiation with works in their rebuttal. While the reviewers still feel the novelty is limited, they agree that this is a step in the right direction. Overall, all reviewers agree on the merits of the work and are in favor of acceptance.

**Additional Comments On Reviewer Discussion:**

The paper has received consistently high scores from all reviewers. They have appreciated the carefully designed empirical framework and insights drawn from the results. Following the rebuttal, all reviewers are unanimously in favor of accepting this work.

---

### Decision · Program_Chairs · 2025-01-22

Accept (Poster)